# MindPilot: Closed-loop Visual Stimulation Optimization for Brain Modulation with EEG-guided Diffusion

**Dongyang Li**[1]* **Kunpeng Xie**[1]* **Mingyang Wu**[1] **Yiwei Kong**[1,2] **Jiahua Tang**[1,3]

**Haoyang Qin**[1] **Chen Wei**[1,4]† **Quanying Liu**[1,4,5]†

[1]Southern University of Science and Technology [2]University of Delaware

[3]PSL Research University [4]Omni-Intelligence [5]Shenzhen Loop Area Institute

https://github.com/ncclab-sustech/MindPilot

## Abstract

Whereas most brain–computer interface research has focused on decoding neural signals into behavior or intent, the reverse challenge—using controlled stimuli to steer brain activity—remains far less understood, particularly in the visual domain. However, designing images that *consistently elicit desired neural responses* is difficult: subjective states lack clear quantitative measures, and EEG feedback is both noisy and non-differentiable. We introduce **MindPilot**, the first closed-loop framework that uses **EEG signals as optimization feedback** to guide naturalistic image generation. Unlike prior work limited to invasive settings or low-level flicker stimuli, MindPilot leverages non-invasive EEG with natural images, treating the brain as a black-box function and employing a pseudo-model guidance mechanism to iteratively refine images without requiring explicit rewards or gradients. We validate MindPilot in both simulation and human experiments, demonstrating (i) efficient retrieval of semantic targets, (ii) closed-loop optimization of EEG features, and (iii) human-subject validations in mental matching and emotion regulation tasks. Our results establish the feasibility of EEG-guided image synthesis and open new avenues for non-invasive closed-loop brain modulation, bidirectional brain–computer interfaces, and neural signal–guided generative modeling.

## 1 Introduction

The ability to modulate brain activity with precisely designed visual stimuli could open new avenues for cognitive enhancement, neurorehabilitation, and bidirectional human–AI interaction. However, designing images that reliably steer neural responses remains largely unexplored. Such modulation can be understood as steering the brain toward specific internal states, as reflected in neural signals like EEG (Epstein & Kanwisher, 1998; Qiu et al., 2023). Conceptually, this task resembles steering a deep visual encoder, where the aim is to design images that elicit particular internal representations. Yet, unlike artificial networks, the human brain poses unique challenges: humans' subjective states lack clear quantitative measures, real EEG responses are noisy and variable, and the brain itself is fundamentally *non-differentiable*.

Recent advances in controllable generation, particularly text-conditioned diffusion models (Li et al., 2019; Rahmani et al., 2022; Epstein et al., 2023; Wei et al., 2024a), offer unprecedented flexibility in image synthesis. But these models are optimized for linguistic prompts, not neural feedback, and thus remain orthogonal to the challenge of brain-targeted generation. Prior efforts in closed-loop visual neuromodulation have shown that generative models can be guided by neuronal responses to synthesize activity-maximizing stimuli (Ponce et al., 2019; Walker et al., 2019; Bashivan et al., 2019; Minai et al., 2024). However, these approaches are typically invasive, relying on small-scale cortical recordings and targeting low-level neuronal activity rather than cognitive states. On the non-invasive

---

*Equal Contribution.

†Corresponding Author.

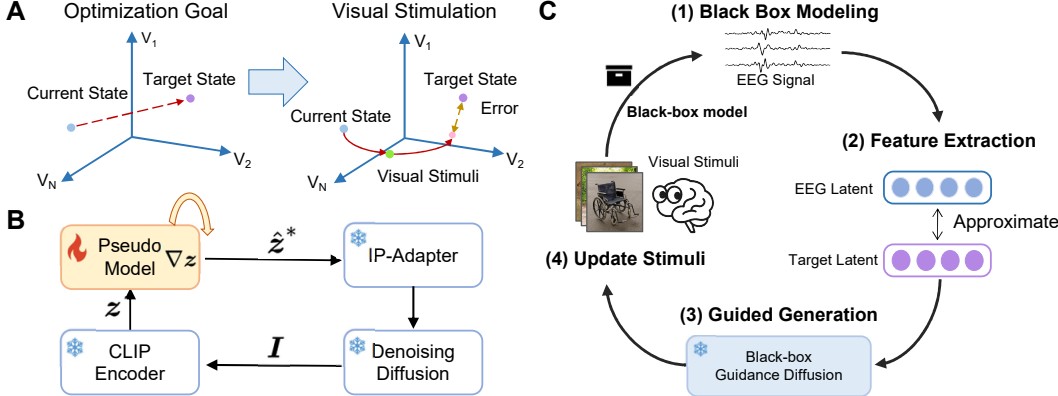

Figure 1: **Conceptualization of MindPilot. A**. The goal of MindPilot is to continuously optimize visual stimuli to drive the brain latent state to the target. **B**. A pseudo-model provides surrogate gradients to iteratively refine images with respect to neural targets (e.g., semantic feature, spectral feature). **C**. The closed-loop visual optimization.

side, EEG provides large-scale, distributed measures of brain activity that are more directly linked to cognition. For example, (Luo et al., 2024b) introduced the VEP Booster to modulate EEG with flickering visual stimuli. Yet such methods are constrained to low-level visual domains and cannot generate semantically rich, naturalistic images that engage higher-order brain representations.

We introduce **MindPilot**, a closed-loop framework that unifies black-box optimization with diffusion-based generation for EEG-guided image design (Fig. 1). MindPilot employs a *surrogate-guided strategy* that replaces explicit reward gradients with pseudo-model updates, enabling gradient-free optimization toward diverse neural targets such as semantic similarity or EEG spectral features. Through both surrogate simulations and human EEG experiments, MindPilot achieves efficient *convergence within limited iterations* while producing *interpretable, naturalistic images aligned with neural selectivity and perception*. Our main contributions are:

- **EEG-guided visual stimuli optimization:** a general framework that leverages the black-box proxy model to guide diffusion to synthesize stimuli capable of steering latent brain states.
- **Black-box guidance generation:** a novel pseudo-model strategy for flexible, gradient-free optimization of diverse neural targets.
- **Comprehensive validation**: extensive experiments in simulation and human EEG experiments, showing consistent feasibility in various neural targets, from EEG semantic features to EEG spectral features and human emotion ratings.

## 2 RELATED WORK

**Neural Selectivity and Invariance with EEG.** EEG captures distributed neural signatures reflecting both *selectivity* (e.g., the N170 ERP for faces (Eimer, 2011)) and *invariance* (different inputs yielding equivalent neural responses (Baroni et al., 2023)). Multivariate decoding methods have shown that EEG patterns can reliably differentiate object categories and semantic content (Holm et al., 2024). However, most work treats EEG as an offline readout rather than *an optimization signal for stimulus design*.

**Closed-loop Visual Neuromodulation.** Generative models have been used to design stimuli that maximize neural activity in invasive recordings (Bashivan et al., 2019; Walker et al., 2019; Pierzch-lewicz et al., 2023) and to refine low-level EEG biomarkers through flicker-based paradigms (Luo et al., 2024b). In the semantic domain, pioneering non-invasive works have explored feedback-driven stimulus selection (Grizou et al., 2025) or generation based on binary attributes (Davis et al., 2022). However, these approaches often operate on discrete choices or limited semantic subspaces, restricting the diversity of synthesized images. Similarly, methods relying on heuristic latent averaging (de la

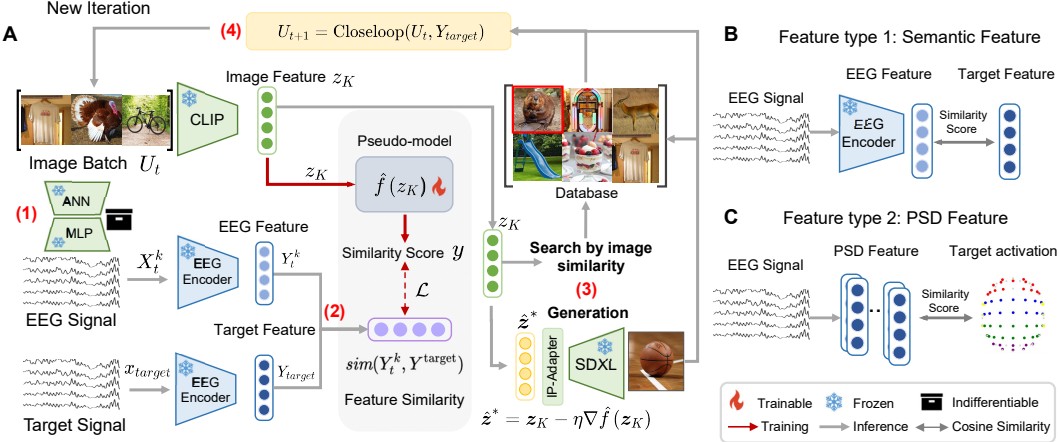

Figure 2: **Framework**. **A**. Each MindPilot iteration involves four steps: 1): The black-box proxy model $g$, which maps images to synthetic EEG, is designed as a black-box proxy model to predict the brain responses $X$. 2): The EEG Encoder $f$ identifies different kinds of features $Y$ from EEG. MindPilot calculates the similarity score $sim(f(g(u)), y_{\text{target}})$ as rewards based on EEG features. 3): Update the image embedding using gradient descent. 4): The image with a higher brain similarity score is selected and passed back to the image generator to optimize stimuli. **B**. Semantic feature from a pre-trained EEG encoder $f$, aligned with CLIP embedding. **C**. Brain energy feature using Power Spectral Density (PSD) features. For more details, refer to Section 3.2.

Torre-Ortiz et al., 2020) face challenges in sample efficiency during the search process. Consequently, **MindPilot** aims to bridge these gaps by establishing a framework for *efficient, continuous, and high-fidelity naturalistic image synthesis* driven directly by non-invasive EEG.

**Brain-conditioned Controllable Generation.** Text-conditioned diffusion has become the de facto standard for controllable image synthesis (Nichol et al., 2022; Ramesh et al., 2022; Ye et al., 2023; Podell et al., 2024). fMRI-conditioned image generation has been extensively developed (Wang et al., 2024; Xia et al., 2024; Scotti et al., 2024; Shen et al., 2024; Gong et al., 2025). Extensions to neural conditioning include gradient-based brain-guided synthesis (Luo et al., 2023; 2024a; Cerdas et al., 2025) and encoding-model alignment with fMRI (Gu et al., 2023; Bao et al., 2025). However, EEG-conditioned generation has mostly been explored for decoding or reconstruction (Bai et al., 2024; Li et al., 2024; Fu et al., 2025; Lopez et al., 2025; Guo et al., 2025), and has not yet been integrated into *closed-loop optimization frameworks*. Our work addresses this gap by introducing MindPilot, which directly optimizes naturalistic image generation guided by EEG responses.

## 3 METHODS

We introduce MindPilot, a closed-loop framework for generating the optimal visual stimuli to align with the target brain activity (Fig. 2A). At its core, MindPilot treats the brain (or its EEG readout) as a *black-box proxy model* and optimizes stimuli by iteratively maximizing the similarity between predicted neural responses and a target embedding. This formulation enables flexible objectives: e.g., aligning with *semantic features* in CLIP space (Fig. 2B) or *EEG spectral features* (Fig. 2C).

### 3.1 PROBLEM SETUP

Let $u$ be a visual stimulus (an image) and let $x \in \mathbb{R}^{C \times T}$ be the corresponding multi-channel EEG response it evokes, where $C$ is the number of channels and $T$ is the number of time points. We model the brain's visual processing as an unknown, non-differentiable forward process: $x = g(u)$, where the function $g$ can represent either the actual human brain or a pre-trained, black-box neural network serving as its proxy.

Our goal is to modulate the brain state towards a specific target. This target brain state is defined by a neural feature vector, $\boldsymbol{y}_{\text{target}}$, which is extracted from the specified target EEG signal $\boldsymbol{x}_{\text{target}}$ via a feature encoder, $f : \mathbb{R}^{C \times T} \to \mathbb{R}^F$. Thus, $\boldsymbol{y}_{\text{target}} = f(\boldsymbol{x}_{\text{target}})$. The objective is to find an optimal image, $\boldsymbol{u}^*$, that generates an EEG response whose features are maximally similar to the target features. This can be formulated as an optimization problem: $\boldsymbol{u}^* = \arg\max_{\boldsymbol{u}} \ sim(f(g(\boldsymbol{u})), \boldsymbol{y}_{\text{target}})$, where $sim(\cdot, \cdot)$ is the cosine similarity between the EEG representation evoked by visual stimuli $\boldsymbol{u}$ and the target brain state $\boldsymbol{y}_{\text{target}}$.

## 3.2 Closed-loop Optimization

MindPilot proceeds iteratively to find images matching a target feature $\boldsymbol{y}_{\text{target}}$. Starting from a uniform prior where every image has an equal selection probability $\frac{1}{N}$, MindPilot dynamically updates the score $\boldsymbol{S}_t(\boldsymbol{u})$ for each image in the database at iteration $t$. These scores define a non-uniform selection probability $P_t(\boldsymbol{u})$ that guides the sampling for the next round. Let $\mathcal{I}_{\text{best}}$ denote the set of indices corresponding to the top-$k$ images in step $t$ with the highest similarity scores (i.e., $|\mathcal{I}_{\text{best}}| = k$).

**Step A: Direct Reward Update.** First, we compute an intermediate score $\boldsymbol{S}'_t$ by applying a direct reward to all images whose indices are in the best set $\mathcal{I}_{\text{best}}$. The score for each of these top-$k$ images is updated via an Exponential Moving Average (EMA):

$$\boldsymbol{S}'_t(\boldsymbol{u}_i) := \begin{cases} (1-\alpha) \cdot \boldsymbol{S}_t(\boldsymbol{u}_i) + \alpha \cdot sim(f(g(\boldsymbol{u}_i)), \boldsymbol{y}_{\text{target}}) & \text{if } i \in \mathcal{I}_{\text{best}} \\ \boldsymbol{S}_t(\boldsymbol{u}_i) & \text{if } i \notin \mathcal{I}_{\text{best}}, \end{cases} \tag{1}$$

where $\alpha$ is the importance weight for the direct reward.

**Step B: Spreading Update.** Next, we "spread" the rewards from the top-$k$ images to other similar images in the database $\Omega = \{\boldsymbol{u}_1, \ldots, \boldsymbol{u}_N\}$. The updated score for the $j$-th image, $\boldsymbol{S}_{t+1}$, is calculated by adding an aggregated spreading term, which is the average of the rewards spread from each of the top-$k$ images, weighted by their CLIP embedding similarity $s$:

$$\boldsymbol{S}_{t+1}(\boldsymbol{u}_j) := (1-\beta)\boldsymbol{S}'_t(\boldsymbol{u}_j) + \frac{\beta}{|\mathcal{I}_{\text{best}}|} \sum_{i \in \mathcal{I}_{\text{best}}} \boldsymbol{S}'_t(\boldsymbol{u}_i) \frac{\exp(s(\boldsymbol{u}_i, \boldsymbol{u}_j))}{\sum_{l=1}^{N} \exp(s(\boldsymbol{u}_i, \boldsymbol{u}_l))}, \tag{2}$$

where $\beta$ is the hyperparameter. This ensures that an image $\boldsymbol{u}_i$ receives a stronger score boost if it is similar to multiple images in the high-scoring set $\mathcal{I}_{\text{best}}$. The reward being spread from each best item is proportional to its own updated score, $\boldsymbol{S}'_t(\boldsymbol{u}_i)$.

Having defined a valid score $\boldsymbol{S}_{t+1}(\boldsymbol{u}_j)$ for all candidate images $j \in \{1, \ldots, N\}$ with Eq. (1) and 2, the updated scores are converted into a probability distribution for the next selection round using the softmax function:

$$P_{t+1}(\boldsymbol{u}_j) := \frac{\exp(\boldsymbol{S}_{t+1}(\boldsymbol{u}_j))}{\sum_{l=1}^{N} \exp(\boldsymbol{S}_{t+1}(\boldsymbol{u}_l))}. \tag{3}$$

The framework then samples images based on this updated probability $P_{t+1}(\boldsymbol{u}_j)$ to continue the search and generation process in the subsequent iteration $t+1$.

## 3.3 Black-box Proxy Model

Direct real-time EEG acquisition is costly and impractical for large-scale closed-loop experiments. To address this, MindPilot is designed to interface flexibly with an EEG black-box proxy model $g$, serving as a black-box proxy of the human brain, which is treated as a non-differentiable black-box in optimization. The proxy model combines a pre-trained backbone with a regression head trained to minimize Mean Square Error (MSE) against real EEG recordings. Specifically, we replace the classification layer of the backbone with a $C \times T$ regression layer, where each unit corresponds to a flattened EEG channel–timepoint signal.

To demonstrate the framework's flexibility, we instantiated a diverse set of black-box proxies, ranging from classic CNNs (AlexNet (Krizhevsky et al., 2012), ResNet50 (He et al., 2016), CORnet-S (Kubilius et al., 2019)) to recent self-supervised and vision-transformer architectures (MoCo (He et al., 2020), ViT-B-32 (Dosovitskiy et al., 2021), OpenCLIP-ViT-B-32 (Cherti et al., 2023), DINO2-ViT-B-14 (Oquab et al., 2024), DINO-ViT-B-16 (Caron et al., 2021), SYNCLR-ViT-B-16 (Sundaram et al., 2024)), as reported in Tab. 1.

Table 1: **Evaluation of black-box proxy models.** We compared the predicted visual-evoked EEG response against the real EEG response over the [60, 500] ms post-stimulus window. We report the **averaged** Pearson's correlation coefficient $R$ and the Noise Ceiling. *Note:* This metric aggregates performance across a broad window, which inherently dilutes peak correlations. Time-resolved analysis reveals significantly stronger predictive power at key latencies (reaching $r \approx 0.6$ at $\sim$100ms, see **Appendix Fig. A.8**).

| Metrics(%) | AlexNet | ResNet50 | CORnet-S | MoCo | ViT | OpenCLIP | DINO | DINO2 | SYNCLR | Noise Ceiling |
|---|---|---|---|---|---|---|---|---|---|---|
| $R \uparrow$ | 16.41 | **16.89** | 8.74 | 16.26 | 13.05 | 12.78 | 13.55 | 13.84 | 12.89 | 18.18 |

## 3.4 INTERACTIVE SEARCH

To bootstrap optimization with an unknown target image, drawing inspiration from interactive retrieval methods (Ferecatu & Geman, 2007), we propose a similarity-weighted sampling approach. Starting from random candidates, MindPilot updates the sampling distribution using a roulette-wheel selection weighted by similarity score. Over iterations, the distribution sharpens around stimuli eliciting neural activity closer to the target (Algorithm 1).

The process begins with a randomly selected set of images $U_0$, without prior knowledge of the specific features of the target image. We use the roulette wheel selection algorithm to choose from current images based on the similarity score $sim(f(g(\boldsymbol{u}_+^k)), \boldsymbol{y}_{\text{target}})$. The system updates the probability $\boldsymbol{P}_t(\boldsymbol{u}_+^k)$ for each image in the database belonging to the target class, based on the response model's prediction $\boldsymbol{Y}_t = f(g(\boldsymbol{U}_t)) \in \mathbb{R}^{N \times F}$. Subsequently, MindPilot calculates the similarity score between the target and the feature predicted by the image selected. See more implementation details in Appendix A.1.3. Once an image is identified as the best in a iteration, the likelihood of similar images in the search space belonging to the target class is increased.

## 3.5 HEURISTIC GENERATION

Searching in a fixed image pool is limiting. We therefore integrate EEG-guided diffusion to generate new stimuli. Specifically, we employ the pretrained guided diffusion model SDXL-Lightning (Lin et al., 2024) with black-box guidance: a Gaussian Process (GP) surrogate predicts reward gradients in latent space for EEG-guided image generation.

**EEG-driven Black-box Guidance**  Recently, black-box guidance diffusion has proved to be successful at generating high-quality images and drug discovery (Fan et al., 2023; Black et al., 2024; Tan et al., 2025). To replace directly calculating the gradient on the black-box encoding model, we use a pseudo target embedding $\hat{z}^*$ by the pseudo target model with input $z_K$. For the Gaussian Process (GP) update case, we set the pseudo target embedding $\hat{z}^*$ as the one gradient step update using the gradient of the mean prediction of the GP surrogate model as follows:

$$\hat{z}^* = \boldsymbol{z}_K - \eta \nabla \hat{f}(\boldsymbol{z}_K; \boldsymbol{Z}^n), \tag{4}$$

where $K$ is the number of diffusion sampling steps, $\eta$ is the step size that decays linearly, $\boldsymbol{z}_K$ denotes the CLIP embedding from generated latents and $\boldsymbol{Z}^n = [\boldsymbol{z}^1, \cdots, \boldsymbol{z}^n]$. The $\hat{f}(\boldsymbol{z}_K; \boldsymbol{Z}^n)$ denotes the prediction of the GP surrogate model (Seeger, 2004) evaluated at $z_K$, which has closed-form as below:

$$\hat{f}(\boldsymbol{z}_K; \boldsymbol{Z}^n) = \boldsymbol{k}(\boldsymbol{z}_K, \boldsymbol{Z}^n)^T \left( \mathcal{K}(\boldsymbol{Z}^n, \boldsymbol{Z}^n) + \lambda \boldsymbol{I} \right)^{-1} y \tag{5}$$

where $\boldsymbol{Z}^n = [\boldsymbol{z}^1, \cdots, \boldsymbol{z}^n]$ and $\boldsymbol{y} = [y^1, \cdots, y^n]$ denotes the collected data from the previous $t - 1$ step and its corresponding rewards in each sampled set $U_t$, respectively. We define each reward $y^i$ by the scaling factor $\gamma$ as $y^i = sim(f(g(\boldsymbol{u}_i)), \boldsymbol{y}_{\text{target}}) \times \gamma$.

**Online Iteration Algorithm**  We integrate an EEG-guided genetic algorithm to optimize stimuli towards the evolution direction of the target neural activity. The specific procedural steps of our algorithm are outlined in Algorithm 2. Unlike the interactive searching process in Algorithm 1, after sampling the stimulus image in each step $t$, we perform crossover and "mutation" on the image embeddings and then sample new images from the image space. See Appendix A.1.4 for additional details on the evolution process. Throughout this process, the relative order of the original CLIP

features is preserved within each dimension to ensure that the mutated images remain semantically coherent and interpretable by humans.

Figure 3: **EEG-guided Interactive Search. A**. Similarity score between MindPilot's neural representations (steps 1, 2, last, and best-step) and the target, versus random stimuli. **B**. Cross-modal correlation between image and EEG embedding similarity ($R = 0.23$, $P < 0.01$). **C**. Similarity score improvement across all subjects using semantic features. **D**. The correlation between image embedding similarity and EEG semantic feature similarity across all subjects. The vertical axis represents the similarity score between the EEG features at the current step and the target.

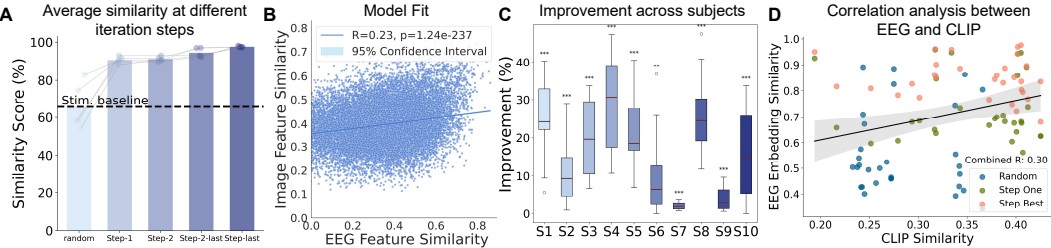

# 4 EXPERIMENTS

## 4.1 SETUP

**EEG readout from proxy models**   We trained a family of predictors to approximate EEG responses evoked by visual stimuli using the THINGS-EEG2 dataset (Gifford et al., 2022; Grootswagers et al., 2022). Each model regressed 17-channel × 250 timepoint EEG signals from pre-trained visual features. Nine variants were evaluated across participants, with performance quantified by Pearson's $r$ and noise ceiling (Tab. 1). Training used all repetitions (four trials per image) and was computationally efficient, fitting within 48 GB memory on a single NVIDIA L40 GPU (see Appendix A.1.1 for details). As shown in Tab. 1, even relatively simple CNN-based proxies (e.g., AlexNet, ResNet50) achieved competitive prediction accuracy. These results confirm that **MindPilot can operate effectively with a wide spectrum of black-box proxies**, highlighting the framework's generality: it is not tied to a particular backbone, but instead offers a plug-and-play recipe for closed-loop optimization with any image-to-EEG predictor.

**Target Features of the evoked EEG**   We considered two neural targets: (i) semantic embeddings, extracted from ATM-S (Li et al., 2024) and aligned with CLIP ViT-H-14 features, and (ii) spectral signatures, derived from EEG power spectral density (PSD). Semantic targets were used in the retrieval task, while both semantic and PSD targets were used in the generation task, ensuring that optimization was cognitively meaningful. Moreover, to test whether MindPilot framework can be extended to the subjective state modulation even without an explicit EEG feature as the target, we considered emotion regulation under visual stimulation using self-reported scores in real human experiment in Appendix 4.4.1.

## 4.2 EEG-GUIDED INTERACTIVE SEARCH

We first tested whether MindPilot can retrieve a target stimulus from the THINGS-EEG2 search space (50 categories × 12 conditions = 600 images). Starting from 10 random images, the model iteratively updated a sampling distribution based on EEG–CLIP similarity. We set slippage factor ($\alpha = 0.1$) and reward propagation factor ($\beta = 0.1$). Although the set hyperparameters were not subjected to a thorough search, they were sufficient to bring about an increase in iterations.. Results are summarized in Fig. 3. **Convergence**: Similarity scores improved consistently across iterations, surpassing random sampling baselines (Fig. 3A). **Alignment**: EEG embeddings exhibited significant correlations with CLIP representations across participants, confirming CLIP similarity as a valid proxy for neural alignment (Fig. 3B). **Subject-level robustness**: Per-subject performance (averaged across five seeds and three target images) showed significant improvements, confirmed by paired

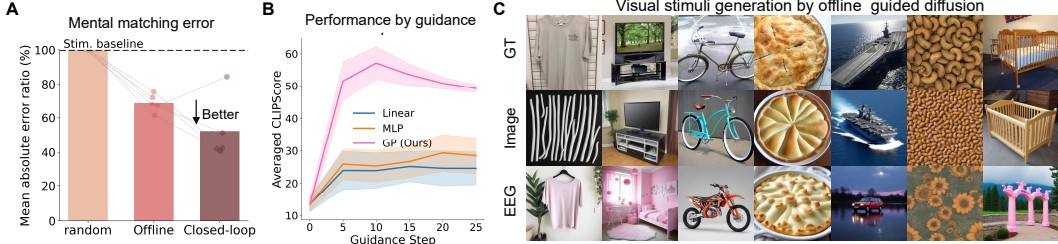

Figure 4: **EEG semantic feature-guided visual stimuli generation. A**. Mean L1 error reduction relative to the random stimuli baseline and offline EEG-guided generation, averaged over five closed-loop experimental targets. **B**. CLIP similarity scores for semantic reconstruction across three different estimators in the pseudo-model setup. **C**. Ablation study on optimization guidance modalities. Top: Original ground-truth target images. Middle (Upper Bound): Generation results optimized directly via target image features (CLIP). Bottom (Ours): Generation results optimized via EEG features (MindPilot), demonstrating the model's capability to bridge the modality gap under real-world noisy conditions.

$t$-tests (Fig. 3C). **Dynamics**: Iterative updates progressively aligned EEG and CLIP embeddings (Fig. 3D). These results establish MindPilot as an effective closed-loop retrieval system under realistic EEG constraints.

## 4.3 EEG-GUIDED STIMULUS GENERATION

We next evaluated whether MindPilot can generate optimized stimuli rather than search in a fixed pool. Four candidates were selected for the next round via roulette sampling. We conducted a closed-loop stimulus optimization experiment within the 600-image space (50 categories × 12 conditions) of THINGS-EEG2. Starting from 10 randomly selected images (out of 200), we used Stable Diffusion XL-Lightning (Lin et al., 2024) with IP-Adapter (Ye et al., 2023) to generate 2 new images per iteration. From the resulting image feature space, 4 candidates were selected via roulette sampling for the next round of stimulation. Tab. 3 reports different feature-based Semantic Similarity Score (SS) and Intensity Similarity Score (IS) scores, showing that our method effectively identifies target stimulus within limited iterations. Additional quantitative results are in Appendix A.2. These results highlight the efficiency and generalizability of MindPilot for brain modulation tasks.

Table 2: **Performance of MindPilot in EEG semantic-driven image generation.** We compare MindPilot with two high-performance EEG-to-image methods (i.e., ATM-S (Li et al., 2024), Cong-Capturer (Zhang et al., 2025)), where ATM-S serves as the upper bound. The chance-level baseline for Subject-01 is also shown. Bold indicates the best performance (optimal model). Underlined indicates the second-best performance (suboptimal model).

| Type | Method | Low-level | | High-level | | | | |
|------|--------|-----------|------|-----------|------------|-----------|--------|--------|
| | | PixCorr ↑ | SSIM ↑ | AlexNet(2) ↑ | AlexNet(5) ↑ | Inception ↑ | CLIP ↑ | SwAV ↓ |
| EEG-to-image | ATM-S (Upper bound) | 0.14 | 0.32 | **0.80** | **0.85** | **0.72** | **0.76** | **0.58** |
| | CongCapturer | **0.15** | **0.33** | 0.73 | 0.81 | 0.65 | 0.68 | 0.59 |
| Modulation | Chance-level | 0.05 | 0.28 | 0.49 | 0.49 | 0.50 | 0.48 | 0.69 |
| | MindPilot (Ours) | **0.09** | **0.35** | **0.70** | **0.73** | **0.58** | **0.67** | **0.60** |

**Semantic-driven generation** In a 10-round EEG semantic-guided visual stimuli generation experiment, MindPilot was compared against random selection and an offline black-box diffusion baseline. Across five target images, closed-loop optimization consistently reduced L1 error relative to baselines (Fig. 4A). Generated images preserved semantic fidelity, as illustrated by comparisons with offline GP-based estimators (Fig. 4B–C). In Fig. 4, the middle row serves as a conceptual ceiling, confirming that our proxy-guided optimization algorithm is highly effective when provided with a noiseless target signal. Importantly, Tab. 2 benchmarks MindPilot against specialized EEG-to-image decoders (ATM-S (Li et al., 2024), CongCapturer (Zhang et al., 2025)). *Note:* These reconstruction models serve as a **theoretical upper bound** because they map ground-truth EEG directly to images, whereas

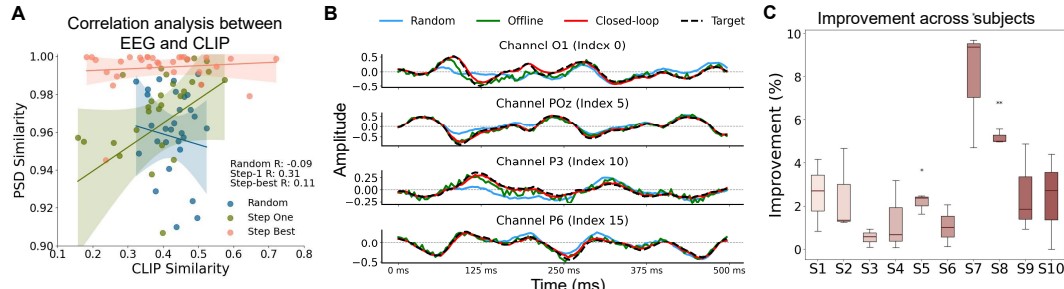

Figure 5: **EEG PSD-driven generation. A**. Neural-visual correlation analysis. We report the correlation coefficients between EEG PSD feature similarity and image CLIP feature similarity across 10 subjects in the THINGS-EEG2 dataset. **B**. Examples of heuristic generation guided by PSD feature across EEG channels. **C**. Improvement in feature similarity scores across all 10 subjects, with statistical significance determined by paired t-tests.

MindPilot must iteratively search for the target without seeing the ground truth. Despite this more challenging setting, MindPilot **substantially outperforms chance-level baselines** (e.g., SSIM: 0.35 vs. 0.28) and achieves high-level semantic alignment comparable to the upper bound (e.g., CLIP-2 way: 0.67 vs. 0.76 for ATM-S), validating the efficacy of closed-loop modulation.

These results highlight that *closed-loop optimization in MindPilot alone can effectively steer image generation toward neural targets with competitive semantic fidelity*, achieving performance surprisingly close to specialized decoders, even though our framework is designed for general-purpose closed-loop brain–stimulus optimization rather than task-specific semantic decoding.

Table 3: **Performance of closed-loop iteration using semantic embedding**. We used the pretrained AlexNet as the backbone of the black-box proxy model. We evaluate the performance using two metrics: Semantic Similarity Score (SS) and Intensity Similarity Score (IS). We reported per-subject SS and IS values, from initial stimuli (Random) and from optimized stimuli (Step-1 and Step-Best). Ratios of improvements by the closed-loop framework (i.e. ΔSS, ΔIS) are reported.

| Subject | Random SS | Random IS | Step-1 SS | Step-1 IS | Step-Best SS | Step-Best IS | Improved by MindPilot ΔSS (%) | ΔIS (%) |
|---|---|---|---|---|---|---|---|---|
| 1 | 0.5174 | 0.9632 | 0.6686 | 0.9729 | **0.8375** | **0.9976** | 16.8859 | 2.4790 |
| 2 | 0.5197 | 0.9678 | 0.6675 | 0.9764 | **0.7372** | **0.9998** | 6.9701 | 2.3406 |
| 3 | 0.5113 | 0.9883 | 0.6597 | 0.9927 | **0.7871** | **0.9980** | 12.7402 | 0.5306 |
| 4 | 0.5065 | 0.9650 | 0.6498 | 0.9836 | **0.8299** | **0.9963** | 18.0136 | 1.2690 |
| 5 | 0.5315 | 0.9788 | 0.6937 | 0.9768 | **0.8418** | **0.9979** | 14.8151 | 2.1055 |
| 6 | 0.6747 | 0.9836 | 0.8099 | 0.9856 | **0.8826** | **0.9961** | 7.2634 | 1.0461 |
| 7 | 0.8838 | 0.8955 | 0.9410 | 0.9033 | **0.9500** | **0.9742** | 1.8237 | 7.0879 |
| 8 | 0.5077 | 0.8344 | 0.6838 | 0.9435 | **0.8568** | **0.9925** | 17.3066 | 4.8947 |
| 9 | 0.8465 | 0.9602 | 0.9251 | 0.9751 | **0.9597** | **0.9997** | 3.4662 | 2.4597 |
| 10 | 0.5128 | 0.8172 | 0.6707 | 0.9705 | **0.7687** | **0.9934** | 9.8032 | 2.2849 |
| Average | 0.6012 | 0.9354 | 0.7370 | 0.9680 | **0.8451** | **0.9946** | 10.9088 | 2.6498 |

**PSD-driven generation** MindPilot also optimized toward PSD features, directly modulating neural spectral patterns. Correlation analysis confirmed progressive alignment between image features and EEG PSD (Fig. 5A). The comparisons across the random baseline, initial step (step-1), and optimized step (step-best) demonstrate that our iteration effectively aligns neural spectral patterns with visual semantics. Neural alignment was particularly evident in the early post-stimulus window (0–500 ms), where closed-loop images evoked EEG responses closely matching the target (Fig. 5B). Across subjects and seeds, PSD-guided generation significantly improved similarity scores (Fig. 5C).

Together, these findings demonstrate that MindPilot extends beyond search to actively design stimuli that align with both semantic and spectral targets.

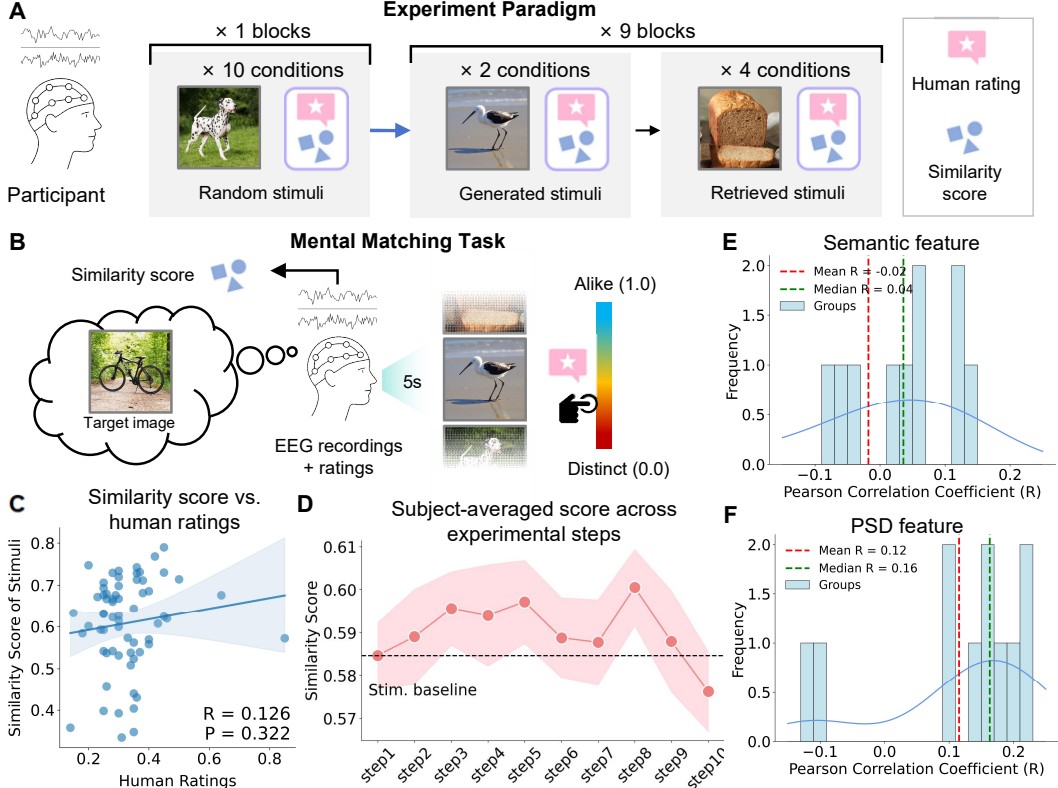

Figure 6: **Real-time closed-loop human experiments validate the control effects of MindPilot. A**. Overview of the experimental paradigm, involving sequential image presentation and rating blocks. **B**. Target-guided similarity judgment: participants (N=10) first memorized a target image, then rated subsequent images on their similarity score to the target, ranging from 0 [least similar] to 1 [highest similar]. **C**. Example correlation between model-derived similarity scores and human ratings for a single target from one participant (Participant, Target 1). **D**. The mean and std. of similarity score from step-1 to step-10 stimuli generated by MindPilot across all 10 participants. **E-F**. Correlation between the similarity score and human ratings.

## 4.4 REAL-TIME CLOSED-LOOP HUMAN EXPERIMENTS

To test generalization beyond proxy models, we conducted behavioral experiments with 10 participants (approved by the institutional ethics committee). All participants provided written informed consent. In the mental matching task, participants memorized a target image, then rated optimized images on a [0,1] similarity scale (Fig. 6A,B). These ratings served as ground-truth labels for assessing MindPilot's ability to steer latent brain states without explicit supervision. EEG was recorded concurrently to provide real-time feedback for adaptive stimulus optimization. More experimental details are provided in Appendix A.1.2.

Results are shown in Fig. 6C-F. Specifically, model-derived similarity scores from EEG responses strongly correlated with human judgments (Fig. 6C). Across iterations, participant ratings increased steadily, consistent with model predictions (Fig. 6D). These effects held across both semantic and PSD-driven targets (Fig. 6E,F). Crucially, the "mental matching" task (Fig. 7C-E) was deployed as an ambitious "stress test" to probe the limits of fine-grained semantic decoding. The moderate performance reflects the intrinsic "sim-to-real" semantic gap and the physiological constraints of non-invasive EEG, establishing a realistic baseline for this frontier challenge. Importantly, when applied to a task with a clearer ground truth—emotion regulation (see Fig. 4.4.1)—MindPilot achieved highly significant modulation. This contrast confirms that the closed-loop optimization framework is robust, effectively steering brain–stimulus alignment whenever the neural target is definable.

### 4.4.1 HUMAN EMOTION REGULATION EXPERIMENTS

**Emotion regulation experiment**. Participants rated each image based on the evoked emotional valence, using a continuous scale from 0 (negative: sad, unpleasant) to 1 (positive: happy, pleasant). This score served as a proxy similarity signal to guide the closed-loop optimization process, enabling MindPilot to modulate emotional states without requiring explicit target EEG features. The control goal was to steer the mind states toward positive valence, as reflected by ratings. Fig. 7A is a schematic diagram of the emotion regulation experiment driven by ratings. The results in Figure 7B show that the rewards estimated by the pseudo model are significantly correlated with the ratings given by human participants (R = 0.714, P ⩽ 0.001), and the regulatory effect on the group is very obvious in Figure 7C (0.45→0.60).

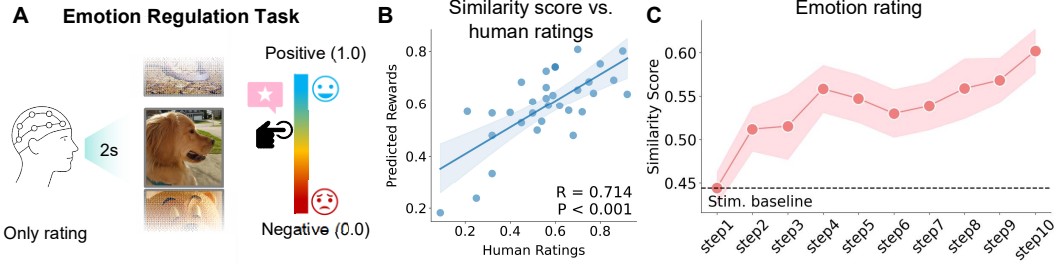

Figure 7: **A**. Affective evaluation: participants rated along an affective dimension, ranging from 0 [sad] to 1 [happy]. **B**. Correlation between the predicted rewards from the estimator and human ratings in the emotion regulation task (i.e., Participant 4, Target 1). **C**. The mean and std. of similarity score from step-1 to step-10 stimuli generated by MindPilot across all 10 participants.

## 5 DISCUSSION AND CONCLUSION

We presented **MindPilot**, a closed-loop framework for EEG-guided visual optimization that synthesizes naturalistic images to modulate brain activity. Across proxy readouts, simulation-based optimization, and human validation, MindPilot consistently converged toward neural targets while maintaining semantic interpretability and high visual fidelity. These results establish the feasibility of treating the brain as a black-box function and position MindPilot as a general approach for brain–stimulus co-optimization. This work represents a pioneering proof-of-concept, demonstrating—for the first time—that closed-loop generation is viable even under the rigorous constraints of non-invasive EEG and naturalistic visual stimuli, expanding the frontier of non-invasive neuro-AI interfaces.

**Technical Impact**    MindPilot introduces a surrogate-guided optimization strategy that enables gradient-free guidance of diffusion models toward EEG-derived objectives. This design not only preserves image quality but also demonstrates that semantically meaningful stimuli can steer neural representations associated with cognition and affect. By explicitly aligning visual features with EEG embeddings, our framework provides a functional link between perception and brain states, opening new avenues for brain–computer interfaces (BCIs), cognitive neuroscience, and neuromodulation therapies (Jang et al., 2021; Alamia et al., 2023).

**Limitations and future directions**    While our results are promising, several challenges remain. First, multiple distinct stimuli can elicit similar EEG responses, echoing Metamers (Feather et al., 2023). Future work should disentangle the representational factors underlying such degeneracy of EEG-conditioned generation. Second, current models do not explicitly address inter-subject variability. Incorporating subject-adaptive or transfer learning strategies could improve robustness and individual control accuracy (Alamia et al., 2023). Moreover, our current experiments validate feasibility of control but are not optimized for latency. Extending MindPilot to real-time operation would enable continuous, adaptive interaction between human and AI systems. Lastly, it is important to extend MindPilot to other modalities to provide richer neural feedback and support higher-level control objectives (e.g., attention, memory, or affective modulation).

## ETHICS STATEMENT

All human-subject experiments were conducted in accordance with the Declaration of Helsinki. The experimental protocol was reviewed and approved by the institutional ethics committee at a local university. All 10 participants were informed of the experimental procedures and goals before the study and provided written informed consent prior to their participation. The experiments involved non-invasive EEG recordings while participants performed visual perception tasks, including mental matching and emotion regulation, by viewing images and providing subjective ratings. Participants were compensated for their time. In addition to the data collected for this study, our research also utilized the publicly available THINGS-EEG2 dataset, which was collected and shared under its own established ethical guidelines.

## ACKNOWLEDGEMENT

This work was supported by the National Natural Science Foundation of China (62472206), National Key R&D Program of China (2025YFC3410000), Shenzhen Science and Technology Innovation Committee (RCYX20231211090405003, JCYJ20220818100213029), GuangDong Basic and Applied Basic Research Foundation (2025A1515011645 to ZC.L.), Shenzhen Doctoral Startup Project (RCBS20231211090748082 to XK.S.), Guangdong Provincial Key Laboratory of Advanced Biomaterials (2022B1212010003), and the open research fund of the Guangdong Provincial Key Laboratory of Mathematical and Neural Dynamical Systems, the Center for Computational Science and Engineering at Southern University of Science and Technology.

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

SUPPLEMENTARY MATERIAL:

MINDPILOT: CLOSED-LOOP VISUAL STIMULATION OPTIMIZATION FOR BRAIN MODULATION WITH EEG-GUIDED DIFFUSION

## A   APPENDIX

Appendix includes the following sections.

- **Sec. A.1:** More Implementation Details
- **Sec. A.2:** Additional Experimental Results
- **Sec. A.3:** EEG black-box Proxy model Experiment Details
- **Sec. A.4:** Additional Interactive Searching Examples
- **Sec. A.5:** Additional Heuristic Generation Examples

### A.1   MORE IMPLEMENTATION DETAILS

#### A.1.1   PUBLIC DATASETS

We conducted our experiments using the training set of the THINGS-EEG2 dataset (Gifford et al., 2022; Grootswagers et al., 2022), which consists of a large EEG corpus from 10 human subjects performing a visual task. The experiments used the Rapid Serial Visual Presentation (RSVP) paradigm for orthogonal target detection tasks to ensure participants' attention to the visual stimuli. All 10 participants underwent 4 equivalent experiments, resulting in 10 datasets with 16,540 unique training image conditions, each repeated 4 times, and 200 unique testing image conditions, each repeated 80 times. In total, this yielded (16,540 training image conditions × 4 repetitions) + (200 testing image conditions × 80 repetitions) = 82,160 image trials. The original data were recorded using a 64-channel EEG system with a 1000 Hz sampling rate. For preprocessing, the data were first downsampled to 250 Hz, and 17 channels were selected from the occipital and parietal regions, which are closely related to the visual system. These channels are O1, Oz, O2, PO7, PO3, POz, PO4, PO8, P7, P5, P3, P1, Pz, P2, P4, P6, and P8. The EEG data were then segmented into trials, spanning from 0 to 1000 ms post-stimulus onset, with baseline correction applied using the mean of the 200 ms pre-stimulus period. Multivariate noise normalization was applied to the training data (Guggenmos et al., 2018).

#### A.1.2   HUMAN EXPERIMENTAL DATA ACQUISITION

**Participants**   The Medical Ethics Committee of SOUTHERN UNIVERSITY OF SCIENCE AND TECHNOLOGY approved the experimental protocol, and participation in the neurophysiological experiment was advertised to the university population. The mean age of the participants was $21 \pm 3$ years. All participants were cisgender (6 males and 4 females). All participants were informed of the study's purpose and provided written consent before participation. Upon completion of the experiment, participants received compensation corresponding to the duration of their involvement.

**Mental matching task**   The "mental matching" is a common psychological task (Geman, 2006): participants memorize the target image beforehand, then view new images interactively, and their EEG is recorded while they are viewing the new images. Based on the feedback from the EEG signals, the image in the participant's mind can be restored. In our experiment, only the judgments from participants are known, both the target EEG signal and the target image are unknown. MindPilot aims to regulate brain states to target states, rather than recovering the image. We need to optimize the visual stimuli based on the target EEG features (known priors) and the feedback from each participant. Eventually, MindPilot achieves the goal of regulation.

The mental matching task integrated real-time EEG acquisition with an image-stimulation paradigm. At the beginning of each trial, the server randomly selected a target image from the THINGS-EEG2 dataset (comprising 200 images). Participants were instructed to view and mentally encode the

target image for a duration of one minute. Subsequently, they were presented with a sequence of server-generated or retrieved images, during which their EEG signals were continuously recorded. Each image was shown for 5 seconds, and a trigger signal was sent to the EEG acquisition device at image onset to accurately mark the temporal window corresponding to image perception.

**Emotion regulation task** The emotion regulation task involved an online image-stimulation and self-assessment system designed to probe participants' affective responses. Participants viewed a series of images presented by the server, either retrieved from an image pool or generated dynamically. Each image was displayed for 2 seconds, after which participants rated their emotional response using a scale ranging from 0.00 to 1.00, with higher scores indicating stronger positive affect. This setup allowed for the real-time capture of subjective emotional evaluations in response to visual stimuli. Participants were instructed as follows: "If the image evokes positive, warm, or happy feelings, please assign a higher score. If the image evokes negative, sad, or disgusting feelings, please assign a lower score. Try to quantify your feelings as precisely as possible and avoid extreme ratings unless strongly warranted."

**Image Set** The emotion regulation task utilized a curated pool of 127 images. Of these, 105 were sourced from a validated Picture Database for Discrete Emotions, constructed using images drawn from established affective image sets including the International Affective Picture System (IAPS) (Lang et al., 1997), ArtPhoto (Machajdik & Hanbury, 2010), and the Geneva Affective Picture Database (GAPED) (Dan-Glauser & Scherer, 2011), as well as selected online platforms. All images in this database were validated through large-scale subjective ratings. An additional 22 images were selected from EmoSet, a large-scale dataset designed for affective computing research (Li et al., 2022).

**EEG recording** The participants were seated in front of a 24" LCD screen, with a resolution of 1920 x 1080 pixels and the refresh rate is 60 Hz. EEG data were recorded using the Neuracle NeuSen W4 wireless EEG acquisition system. A total of 59 scalp electrodes were used in accordance with the international 10–20 system, providing dense spatial coverage of frontal, central, parietal, occipital, and temporal regions. Independent reference and ground electrodes were included in the system. The EEG signals were sampled at a frequency of 250 Hz, sufficient for capturing the temporal dynamics of cognitive and affective processing.

The full set of recording sites included: Fpz, Fp1, Fp2, AF3, AF4, AF7, AF8, Fz, F1, F2, F3, F4, F5, F6, F7, F8, FCz, FC1, FC2, FC3, FC4, FC5, FC6, FT7, FT8, Cz, C1, C2, C3, C4, C5, C6, T7, T8, CP1, CP2, CP3, CP4, CP5, CP6, TP7, TP8, Pz, P3, P4, P5, P6, P7, P8, POz, PO3, PO4, PO5, PO6, PO7, PO8, Oz, O1, and O2.

**EEG preprocessing** Following each image stimulus, EEG signals recorded by the acquisition device were immediately transmitted to the server. The server identified event markers (triggers) within the data stream and segmented the EEG signals accordingly, aligning each segment precisely with its corresponding image presentation. Each EEG epoch, corresponding to a single image, was then subjected to real-time preprocessing to ensure low-latency signal analysis.

To accommodate the constraints of real-time processing, we did not apply computationally intensive artifact removal techniques such as Independent Component Analysis (ICA) for eliminating eye movement or muscle artifacts. Instead, a lightweight preprocessing pipeline was implemented using Python's SciPy library, including a notch filter centered at 50 Hz to remove power line interference, a 1–100 Hz band-pass filter to retain relevant neural activity, and optional signal resampling. Additionally, baseline correction was applied using the final 250 ms of the pre-stimulus period to minimize low-frequency drifts and slow fluctuations.

A.1.3 INTERACTIVE SEARCHING ALGORITHM PIPELINE

We provide a more detailed description of algorithm 1. The algorithm begins by initializing equal selection probabilities for each image in the candidate set, denoted as $P_0(u_j) = \frac{1}{N}$, where $N$ is the total number of images in the retrieval set. This initialization with equal probabilities reflects the absence of prior information, serving as an exploratory phase. In each iteration, a subset of images

---

**Algorithm 1** Closed-loop Interactive Search Algorithm

---

1: **Initialize:** Database $\Omega = \{u_1, ..., u_N\}$. Scores $S_0(u_j)$ for all $j \in \{1, ..., N\}$.
2: Set uniform selection probability as the initial scores $S_0(u_j) = P_0(u_j) = \frac{1}{N}$. Set hyperparameters $\alpha, \beta$.
3: Set the number of best items to consider, $k \geqslant 1$.
4: **repeat**
5:     **Action Selection:** Sample a subset of images $U_t \subset \Omega$ based on probabilities $P_t$.
6:     **Reward Calculation:** For each image $u_i \in U_t$, calculate its similarity score:
7:     $r_i \leftarrow \text{sim}(f(g(u_i)), y_{\text{target}})$
8:     **Find Top-k Best:** Identify the set of indices $I_{\text{best}}$ for the top $k$ images in $U_t$ with the highest scores.
9:     **Step A: Direct Reward Update**
10:     Compute intermediate scores $S'_t$ for all $u_i \in U_t$:
11:     $S'_t(u_i) \leftarrow \begin{cases} (1 - \alpha) \cdot S_t(u_i) + \alpha \cdot r_i & \text{if } i \in I_{\text{best}} \\ S_t(u_i) & \text{if } i \notin I_{\text{best}} \end{cases}$
12:     **Step B: Spreading Update (Aggregated from Top-k)**
13:     Compute updated scores $S_{t+1}$ for all $u_j \in \Omega$ by averaging the spread from all top-k images:
14:     $S_{t+1}(u_j) \leftarrow (1 - \beta) \cdot S'_t(u_j) + \beta \cdot \frac{1}{k} \sum_{i \in \mathcal{I}_{\text{best}}} \left( S'_t(\boldsymbol{u}_i) \cdot \frac{\exp(s(\boldsymbol{u}_i, \boldsymbol{u}_j))}{\sum_{l=1}^{N} \exp(s(\boldsymbol{u}_i, \boldsymbol{u}_l))} \right)$     $\triangleright$ $s$ is CLIP
    embedding similarity.
15:     **Update Action Probabilities**
16:     Update the probability $P_{t+1}$ using softmax for the next iteration:
17:     $P_{t+1}(u_j) \leftarrow \frac{\exp(S_{t+1}(u_j))}{\sum_{l=1}^{N} \exp(S_{t+1}(u_l))}$
18:     $t \leftarrow t + 1$
19: **until** convergence criteria met
20: **Return:** $\arg\max_{u_j \in \Omega} S_t(u_j)$

---

$U_t = \{u_1, u_2, \ldots, u_N\}$ is selected from the candidate images space $\Omega$ based on the current selection probabilities $P_t(u)$.

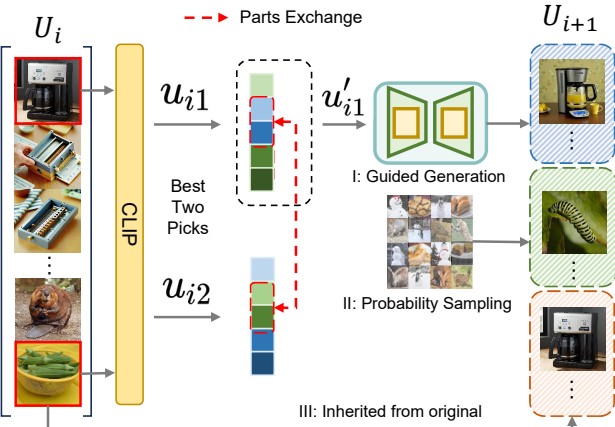

Figure A.1: Generating subsequent images based on the current round is achieved through crossover, mutation, and a guided diffusion model. Both crossover and mutation operations preserve the relative ordering of CLIP features, thereby maintaining their semantic coherence.

For each image $u_+^k$ in the subset $U_t$ the algorithm computes a similarity score $sim\langle u_+^k, u_{\text{target}}\rangle$ by comparing the image's representation with the target. This similarity score acts as an immediate **reward** within the framework. The maximum similarity score among the subset is identified as a measure of the effectiveness of the current action. If $sim_{\max}$ does not meet a predefined $threshold$, the reward is considered insufficient, and the algorithm returns to the image selection step, effectively

---

**Algorithm 2** Closed-loop Heuristic Generation Algorithm

---

1: **Initialize:** Database $\Omega_0$, scores $S_0(u_i)$ for all $i \in \{1, ..., N\}$, probabilities $P_0(u_i) = S_0(u_i) = \frac{1}{N}$.

2: Parameters: parent population size $N_p$, top-$k$ offspring count $k$, hyperparameters $\alpha, \beta$.

3: Operators: parent selector SelectParents$(\cdot)$, embedding crossover Crossover$(\cdot, \cdot)$, image generator Generate$(\cdot)$.

4: **repeat**

5:     $U_{\text{parents}} \leftarrow \text{Sample}(\Omega_t, P_t, N_p)$         ▷ Select parents based on global probabilities

6:     $U_{\text{offspring}} \leftarrow \{\text{Generate}(\text{Crossover}(F(u_a), F(u_b))) \mid (u_a, u_b) \sim \text{SelectParents}(U_{\text{parents}}, S_t)\}_{i=1}^{N_p}$

7:     Calculate rewards $\{r_j \mid u_j \in U_{\text{offspring}}\}$ where $r_j \leftarrow sim(f(g(u_j)), y_{\text{target}})$.

8:     $\Omega_{t+1} \leftarrow \Omega_t \cup U_{\text{offspring}}$         ▷ Integrate new offspring into the database

9:     Extend $S_t$ to cover $\Omega_{t+1}$ with initial scores for new images.

10:     $\mathcal{I}_{\text{best}} \leftarrow \arg \text{topk}_j\{r_j \mid u_j \in U_{\text{offspring}}\}$         ▷ Identify best new offspring

11:     Compute intermediate scores $S'_t$ for all $u_i \in \Omega_{t+1}$:

12:     $S'_t(u_i) \leftarrow \begin{cases} (1-\alpha) \cdot S_t(u_i) + \alpha \cdot r_i & \text{if } i \in \mathcal{I}_{\text{best}} \\ S_t(u_i) & \text{if } i \notin \mathcal{I}_{\text{best}} \end{cases}$

13:     Compute final scores $S_{t+1}$ for all $u_j \in \Omega_{t+1}$:

14:     $S_{t+1}(u_j) \leftarrow (1-\beta)S'_t(u_j) + \frac{\beta}{k} \sum_{j \in \mathcal{I}_{\text{best}}} \left( S'_t(u_i) \cdot \frac{\exp(s(u_i, u_j))}{\sum_{l \in \Omega_{t+1}} \exp(s(u_i, u_l))} \right)$

15:     Update global probabilities $P_{t+1}$ for all $u_j \in \Omega_{t+1}$:

16:     $P_{t+1}(u_j) \leftarrow \frac{\exp(S_{t+1}(u_j))}{\sum_{l \in \Omega_{t+1}} \exp(S_{t+1}(u_l))}$

17:     $t \leftarrow t + 1$

18: **until** convergence criteria met

19: **Return:** $\arg \max_{u_j \in \Omega_t} S_t(u_j)$

---

trying a new action within the same state. If $sim_{\max}$ meets or exceeds the $threshold$, the algorithm proceeds to identify the images with the highest similarity scores.

As for each image $u_+^k$ in $U_t$, its selection probability $P_{t+1}(u_+^k)$ is updated by multiplying with a constant factor, representing a policy improvement step that prioritizes images likely to yield higher rewards. After updating, a Softmax function is applied to normalize the probabilities, focusing selection weight on images more similar to the target. This normalization step reflects the transition to a new state with an updated policy. The iteration continues, with the algorithm transitioning through states by selecting new subsets based on the refined probabilities, until $sim_{\max}$ reaches $threshold$ or reach the upper limit of the number of iteration rounds. At this point, the loop terminates, as the algorithm has successfully identified an optimized subset of images that maximizes the similarity reward to the target.

### A.1.4 HEURISTIC GENERATION ALGORITHM PIPELINE

We provide a more detailed description of algorithm 2. As illustrated in Fig. A.1, each image set consists of three parts:

- **Latent Space Crossover and Generation:** This phase integrates evolutionary operators with the diffusion generation process. Two parent instances, denoted as $u_a$ and $u_b$, are selected via roulette wheel sampling based on their fitness. We apply a randomized crossover operation to their latent embeddings, employing a stochastic starting index to maximize variability. The resulting hybrid embedding serves as the conditioning input for the diffusion model. This approach effectively expands the search space (exploration) while preserving the high-fidelity semantic structures of the parent nodes (exploitation).

- **Novelty Injection:** To mitigate mode collapse and maintain population diversity, we introduce an exploratory sampling step. New candidates are drawn from the original dataset, strictly excluding instances utilized in prior iterations. This ensures the continuous introduction of novel semantic elements, preventing the optimization process from stalling in local optima.

- **Elitist Preservation:** We employ an elitist strategy by directly propagating the selected parent instances $u_a$ and $u_b$ to the subsequent generation. This preservation mechanism ensures monotonic improvement of the population quality and stabilizes the evolutionary trajectory.

By combining these three parts, we obtain a new image set for the next iteration.

## A.2 ADDITIONAL EXPERIMENTAL RESULTS

### A.2.1 JUSTIFICATION FOR THE THEORETICAL UPPER BOUND

To rigorously evaluate MindPilot, we defined a upper bound based on the direct reconstruction performance of ATM-S. It is crucial to distinguish the roles of these two methods:

**MindPilot (Our Method):** Operates via iterative, closed-loop feedback. It searches for an image that matches a target brain state without ever accessing the ground-truth EEG signal corresponding to the target image.

**Upper Bound:** Represents a non-iterative, ideal scenario. It utilizes the ground-truth EEG signal (corresponding to the known target image) to directly reconstruct the visual content. This score represents the best possible performance achievable if one already possessed the perfect target brainwave, bypassing the iterative search challenge.

We selected ATM-S as this upper bound specifically because it represents the current state-of-the-art in EEG-to-Image decoding. To validate this choice, we benchmarked numerous recent encoders on the THINGS-EEG2 retrieval task. As shown in Table A.1, ATM-S consistently and significantly outperformed all other methods (including NICE, CogCap, and MindEyeV2) across all metrics. Its superior ability to map EEG to visual semantic features makes it the most appropriate and rigorous ceiling for benchmarking our framework.

Table A.1: **EEG-to-Image Encoder performance on THINGS-EEG2 (in-subject).** We compared ATM-S against other leading decoding baselines. ATM-S achieves the highest retrieval accuracy across all $k$-way metrics, justifying its selection as the theoretical upper bound for our experiments.

| Model | 2-way Top-1 | 4-way Top-1 | 10-way Top-1 | 200-way Top-1 | 200-way Top-5 |
|---|---|---|---|---|---|
| NICE(Song et al., 2024) | 93.23 | 83.93 | 69.22 | 21.67 | 51.34 |
| EEGNetV4(Lawhern et al., 2018) | 91.42 | 80.21 | 63.37 | 16.84 | 42.58 |
| CogCap(Zhang et al., 2025) | 93.15 | 82.85 | 69.35 | 22.05 | 51.60 |
| MB2C(Wei et al., 2024b) | 78.40 | 62.25 | 43.75 | 8.85 | 25.20 |
| MindEyeV2(Scotti et al., 2024) | 92.50 | 82.80 | 66.10 | 23.80 | 50.25 |
| **ATM-S(Li et al., 2024)** | **94.70** | **86.73** | **74.00** | **26.85** | **57.21** |

### A.2.2 HYPERPARAMETER SENSITIVITY ANALYSIS

In the Interactive Searching experiments, we utilized a fixed setting for the hyperparameters $\alpha$ (moving factor) and $\beta$ (reward propagation factor) based on initial pilot observations. To systematically evaluate the robustness of MindPilot and identify the optimal configuration, we performed a comprehensive grid search on the validation set. We explored the range of $[0.1, 0.9]$ for both $\alpha$ and $\beta$. The results are summarized in Table A.2.

The best performance is achieved when $\alpha = 0.1$ and $\beta = 0.1$, yielding a similarity score of $0.6586 \pm 0.0879$. The model maintains competitive performance across a wide range of configurations. However, we observe a general trend where lower values of $\beta$ (controlling reward propagation spread) tend to yield higher scores, suggesting that more localized reward feedback is beneficial for this specific task. The grid search reveals that the results currently reported in the main paper (which used suboptimal initialization) represent a *conservative lower bound* of MindPilot's capabilities. The method effectively "survived" the suboptimal initialization and demonstrates a much higher performance ceiling with proper tuning ($\alpha = 0.1, \beta = 0.1$).

In the Tab.A.2, we found that the $\beta$ parameter has a significant impact on the results. This might be because when we implemented the codes, we included a standardized penalty term for direct rewards,

Table A.2: **Ablation study on hyperparameters $\alpha$ and $\beta$.** The performance is evaluated on the validation set. The values represent the target score (mean $\pm$ std). The best performance is achieved at $\alpha = 0.1, \beta = 0.1$, indicating that finer-grained updates and localized reward propagation favor optimization.

| $\beta$ \ $\alpha$ | 0.1 | 0.3 | 0.5 | 0.7 | 0.9 |
|---|---|---|---|---|---|
| **0.1** | **0.6586 $\pm$ 0.0879** | 0.6336 $\pm$ 0.0584 | 0.6284 $\pm$ 0.0644 | 0.6167 $\pm$ 0.0658 | 0.6224 $\pm$ 0.0600 |
| **0.3** | 0.6586 $\pm$ 0.0879 | 0.6336 $\pm$ 0.0584 | 0.6284 $\pm$ 0.0644 | 0.6203 $\pm$ 0.0709 | 0.6224 $\pm$ 0.0600 |
| **0.5** | 0.6586 $\pm$ 0.0879 | 0.6336 $\pm$ 0.0584 | 0.6264 $\pm$ 0.0667 | 0.6203 $\pm$ 0.0709 | 0.6224 $\pm$ 0.0600 |
| **0.7** | 0.6586 $\pm$ 0.0879 | 0.6292 $\pm$ 0.0592 | 0.6264 $\pm$ 0.0667 | 0.6203 $\pm$ 0.0709 | 0.6224 $\pm$ 0.0600 |
| **0.9** | 0.6586 $\pm$ 0.0879 | 0.6298 $\pm$ 0.0599 | 0.6264 $\pm$ 0.0667 | 0.6217 $\pm$ 0.0700 | 0.6224 $\pm$ 0.0600 |

in order to ensure a stable acquisition of rewards, which made finding the distribution of potential target rewards more important than a single reward.

### A.2.3 NECESSITY OF EEG-DRIVEN OPTIMIZATION

To conclusively demonstrate that EEG feedback provides essential and unique information driving the optimization process, we conducted a rigorous control experiment comparing MindPilot against theoretical lower and upper bounds. We define the Brain Modulation task under the strict constraint that the target image is unknown to the system, and only the target brain feature is the known goal. Based on this, we situate our method's performance within three setups:

1. **Random Image Guidance (Lower Bound/Null):** No optimization is performed. Removing the EEG renders the system without any optimization direction. This establishes the baseline performance.

2. **EEG Feature Guidance (MindPilot):** Optimization is driven solely by EEG feedback, where the target is the specific brain state.

3. **Target Image Guidance (Upper Bound/Ceiling):** Optimization is driven directly by the ground-truth target image's CLIP features. We frame this as a theoretical *ceiling* (or Oracle) rather than a baseline, because it utilizes the exact answer (the target image), which is precisely the information unavailable in our task.

The quantitative results are presented in Tab. A.3. We observe that MindPilot significantly outperforms the Random baseline across all metrics. For instance, in the closed-loop setting, MindPilot achieves a EEG score of $0.7065 \pm 0.0537$ compared to the random baseline of $0.5746 \pm 0.0206$. This statistically significant improvement proves that the EEG signal provides essential, unique information to steer the generation. While the Target Image Guidance (Ceiling) represents the theoretical optimum, MindPilot effectively bridges the gap toward this performance level using only noisy neural feedback.

Table A.3: **Control experiments with different guiding targets.** Comparison of MindPilot (EEG guidance) against the theoretical ceiling (Target image guidance) and the lower bound (Random guidance). MindPilot significantly outperforms the random baseline, confirming the necessity of EEG-driven optimization.

| Type | EEG feature guidance (MindPilot) | | Target image guidance (Ceiling) | | Random image guidance (Null) | |
|---|---|---|---|---|---|---|
| | EEG score | CLIP score | EEG score | CLIP score | EEG score | CLIP score |
| Offline (200 pairs) | 0.5369 $\pm$ 0.0160 | 0.6580 $\pm$ 0.0492 | **0.5452 $\pm$ 0.0122** | **0.8464 $\pm$ 0.0553** | 0.5223 $\pm$ 0.0160 | **0.5849 $\pm$ 0.0286** |
| Closed-loop (10 loops) | **0.5461 $\pm$ 0.0141** | **0.7065 $\pm$ 0.0537** | 0.5440 $\pm$ 0.0121 | 0.7838 $\pm$ 0.0699 | **0.5236 $\pm$ 0.0132** | 0.5746 $\pm$ 0.0206 |

### A.2.4 COMPARISON WITH BLACK-BOX OPTIMIZERS AND EFFICIENCY ANALYSIS

In this section, we provide a detailed theoretical justification for choosing a GP-based "Surrogate Gradient" approach (MindPilot) over standard Bayesian Optimization (BO) and present empirical comparisons regarding convergence speed and query efficiency.

**Pseudo Model vs. BO** We explicitly chose a GP-based "Surrogate Gradient" approach rather than standard BO for two primary theoretical reasons, particularly given the high-dimensional nature of the generative latent space:

**The Curse of Dimensionality:** Standard BO relies on optimizing an acquisition function (e.g., Upper Confidence Bound (UCB) or Expected Improvement (EI)) over the input space. This process becomes computationally intractable and sample-inefficient in the high-dimensional latent space of diffusion models (e.g., $64 \times 64 \times 4 \approx 16,384$ dimensions). Even when optimizing within the same CLIP embedding space (with $1024$ dimensions) of Mindpilot, convergence is still extremely difficult. MindPilot circumvents this by using the proxy to guide gradient estimation rather than performing global optimization over the raw input space.

**Local vs. Global Search:** Standard BO aims for global optimization, which requires extensive exploration and is expensive in terms of query budget. In contrast, MindPilot uses GP to model the *local* landscape around the current generation to estimate an update direction. This allows us to leverage the efficiency of gradient-based guidance (similar to RL) while utilizing GP's superior ability to model EEG uncertainty and noise.

To further demonstrate the performance-speed trade-off, we conducted a comprehensive benchmark comparing MindPilot (Pseudo-model) against state-of-the-art Reinforcement Learning (RL) methods (DDPO, DPOK, D3PO) and traditional black-box optimizers (Standard BO, CMA-ES). We evaluated all methods using different query budgets (5, 10, 50, and 200 samples). For the assessment stage, we selected the minimum of 5 or (num_budget // 2) images for the evaluation of EEG scores. Running time represents the wall-clock time required to complete the optimization process. As shown in Tab. A.4, MindPilot (Pseudo model Offline) achieves a superior trade-off between speed and performance:

1. **vs. RL Baselines (DDPO/DPOK/D3PO):** MindPilot converges orders of magnitude faster. For example, at 200 samples, MindPilot (Offline) takes $\approx$ 73s, whereas DDPO takes $\approx$ 1279s and D3PO takes $\approx$ 1614s.

2. **vs. Black-box Optimizers (BO/CMA-ES):** While BO is fast at low sample counts, its computational cost scales poorly with history size due to matrix inversion (cubic complexity). At 200 samples, MindPilot (Offline) is not only faster ($\approx$ 73s vs. 229s for BO) but also achieves a higher EEG score (0.5384 vs. 0.5242), validating its effectiveness in high-dimensional optimization.

Table A.4: **Efficiency and Performance Benchmark.** Comparison of EEG Scores (Higher is better) and Running Time (Lower is better) across different optimization methods. MindPilot demonstrates trade-off between efficiency and scalability, especially at larger sample sizes.

| Method | 5 Samples | | 10 Samples | | 50 Samples | | 200 Samples | |
|---|---|---|---|---|---|---|---|---|
| | EEG Score ↑ | Time (s) ↓ | EEG Score ↑ | Time (s) ↓ | EEG Score ↑ | Time (s) ↓ | EEG Score ↑ | Time (s) ↓ |
| DDPO(Black et al., 2024) | 0.5125 ± 0.0139 | 107.82 ± 19.42 | 0.5095 ± 0.0097 | 220.27 ± 17.29 | 0.5154 ± 0.0126 | 683.42 ± 38.87 | 0.5125 ± 0.0136 | 1279.59 ± 87.56 |
| DPOK(Fan et al., 2023) | 0.5093 ± 0.0121 | 116.61 ± 21.75 | 0.5138 ± 0.0124 | 221.89 ± 12.14 | 0.5108 ± 0.0154 | 692.70 ± 40.37 | 0.5101 ± 0.0174 | 1332.15 ± 164.56 |
| D3PO(Yang et al., 2024) | 0.5138 ± 0.0162 | 117.70 ± 9.93 | 0.5113 ± 0.0154 | 285.38 ± 6.56 | **0.5500 ± 0.0156** | 486.90 ± 109.96 | 0.5192 ± 0.0104 | 1614.16 ± 128.63 |
| BO(Bashashati et al., 2016) | **0.5228 ± 0.0096** | 5.35 ± 0.34 | 0.5247 ± 0.0058 | 11.11 ± 0.45 | 0.5222 ± 0.0087 | 53.05 ± 3.57 | 0.5242 ± 0.0111 | 229.06 ± 9.57 |
| CMA-ES(Xu et al., 2019) | 0.5224 ± 0.0076 | 5.40 ± 0.42 | 0.5209 ± 0.0086 | **9.26 ± 0.54** | 0.5195 ± 0.0087 | 49.76 ± 2.92 | 0.5227 ± 0.0096 | 239.26 ± 14.03 |
| MindPilot (Offline) | 0.5222 ± 0.0067 | 17.94 ± 1.61 | **0.5264 ± 0.0077** | 20.56 ± 2.87 | 0.5291 ± 0.0058 | **27.75 ± 4.89** | **0.5384 ± 0.0114** | 73.62 ± 19.02 |
| MindPilot (Closed-loop) | 0.5208 ± 0.0107 | 39.63 ± 7.37 | 0.5208 ± 0.0107 | 42.87 ± 9.71 | 0.5327 ± 0.0091 | 64.67 ± 10.92 | 0.5341 ± 0.0071 | 219.78 ± 40.33 |

### A.2.5 ADDITIONAL RECONSTRUCTED IMAGES RESULTS

Fig. A.2 visualizes the best, medium, and worst examples of our image generation results. To obtain these results, we used EEG data from Subject-01 viewing 200 test images. Each image in the test set was sequentially used as a target for an iterative optimization process. We then evaluated the generation quality by calculating the cosine similarity of CLIP embeddings between the generated and original images, selecting the 12 best, 12 medium, and 12 worst-performing examples.

In the best group, the generated images are highly consistent with the original images in both high-level semantics and low-level visual features. In the medium group, the generated images successfully capture the core semantics, but some low-level visual details are distorted or lost. In the worst group, the generated images fail to preserve either the semantic or the low-level features of the original images.

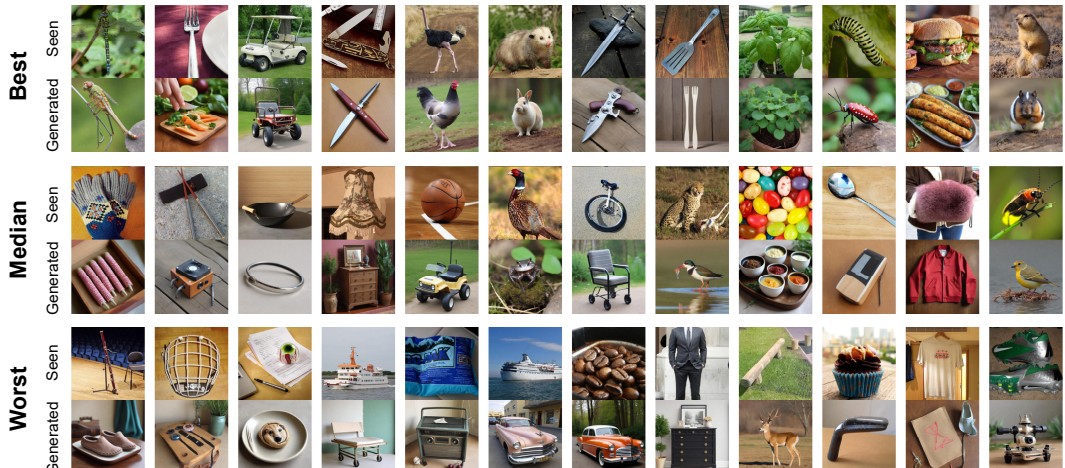

Figure A.2: **Examples of EEG semantic feature-guided visual reconstruction**. From top to bottom, we exhibit the best, median, and worst 12 generated images from Subject-01, respectively. We show the images subjects had seen and the generated images.

### A.2.6 HUMAN EXPERIMENTAL CORRELATION ANALYSIS

We recruited 10 participants (6 males, 4 females) for the experiment. Due to poor data quality, we excluded 4 participants (3 males, 1 female), retaining 6 (3 males, 3 females) for analysis. Our analysis in Fig. 6 (D) and Fig. 7 (C) is derived from all 10 participants. In the correlation analysis of Fig. 6 (E-F), only the selected 6 participants were used.

### A.2.7 ITERATION IMPROVEMENT FROM DIFFERENT SUBJECTS

Based on the conclusions drawn from Fig. A.6, we employ the pre-trained AlexNet end-to-end model as the black-box proxy model and use ATM-S, which is based on semantic similarity score (both the training and testing signals are synthesized), to obtain semantic representations aligned with 1×1024 CLIP image features. The experimental design involves randomly selecting 50 categories, resulting in a searching space of $50 \times 12 = 600$ images. Specifically, we present the iterative performance improvements for three different targets randomly selected from the test set, with results reported for Subjects 1, 7, 8, and 10. As shown in Fig. A.5, we calculate the EEG feature similarity of Subject 1, 7, 8, and 10 at random, step-1, and step-best in the iterative process respectively.

### A.2.8 PERFORMANCE OF DIFFERENT TARGET IMAGES ACROSS SUBJECTS

We report the results of iterative optimization using different targets in two different cases. The results for each subject are shown, along with the average percentage improvement across 5 random seeds. For the EEG semantic feature case, we determined that training and testing with synthetic EEG yielded the highest accuracy based on the retrieval performance shown in Fig. A.6. For the PSD feature, we selected 3 images using the method described in Section 4 and supplemented the iterative improvement performance.

### A.3 VALIDITY VERIFICATION OF SYNTHETIC EEG

To evaluate the performance of our EEG encoding models, we compare the synthetic EEG signals generated by two deep neural networks (DNNs)—AlexNet and CORnet-S—with real EEG data. Here's a step-by-step breakdown of how we processed and compared the data.

We selected 17 specific channels from the original 63-channel EEG dataset, focusing on those most relevant to visual processing. It ensured that we focused on neural regions most directly involved in responding to the visual stimuli. For each stimulus, we averaged the EEG signals across all trials, resulting in a representative dataset for each stimulus. This reduced the dimensionality of the data, making it easier to compare with synthetic data. We used a pretrained end-to-end encoding model to

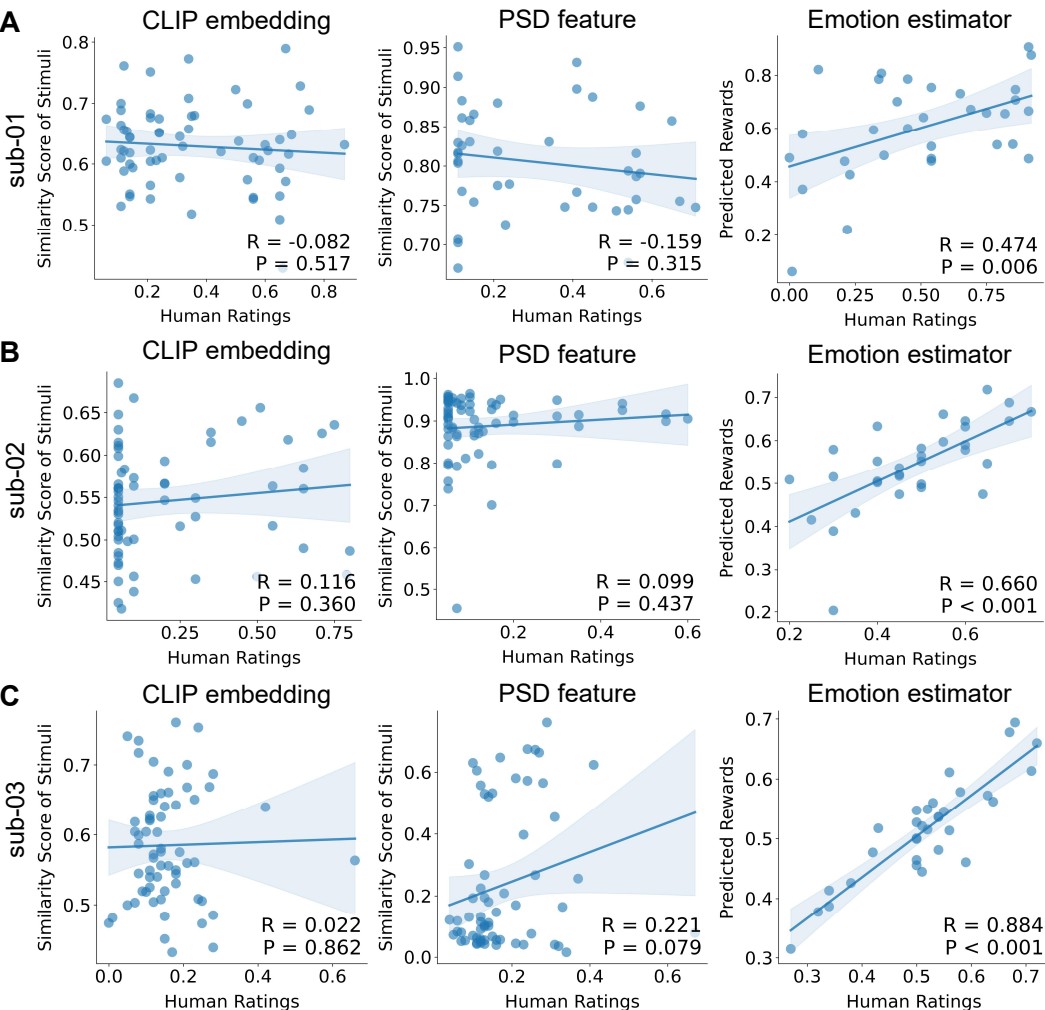

Figure A.3: **Correlations analysis for S1-S3.** (Left and Middle) Correlations between the similarity score of stimulus and human rating in the mental matching task in Target 1. (Right) Correlations between the predicted rewards from the estimator and human ratings in the emotion regulation task in Target 1.

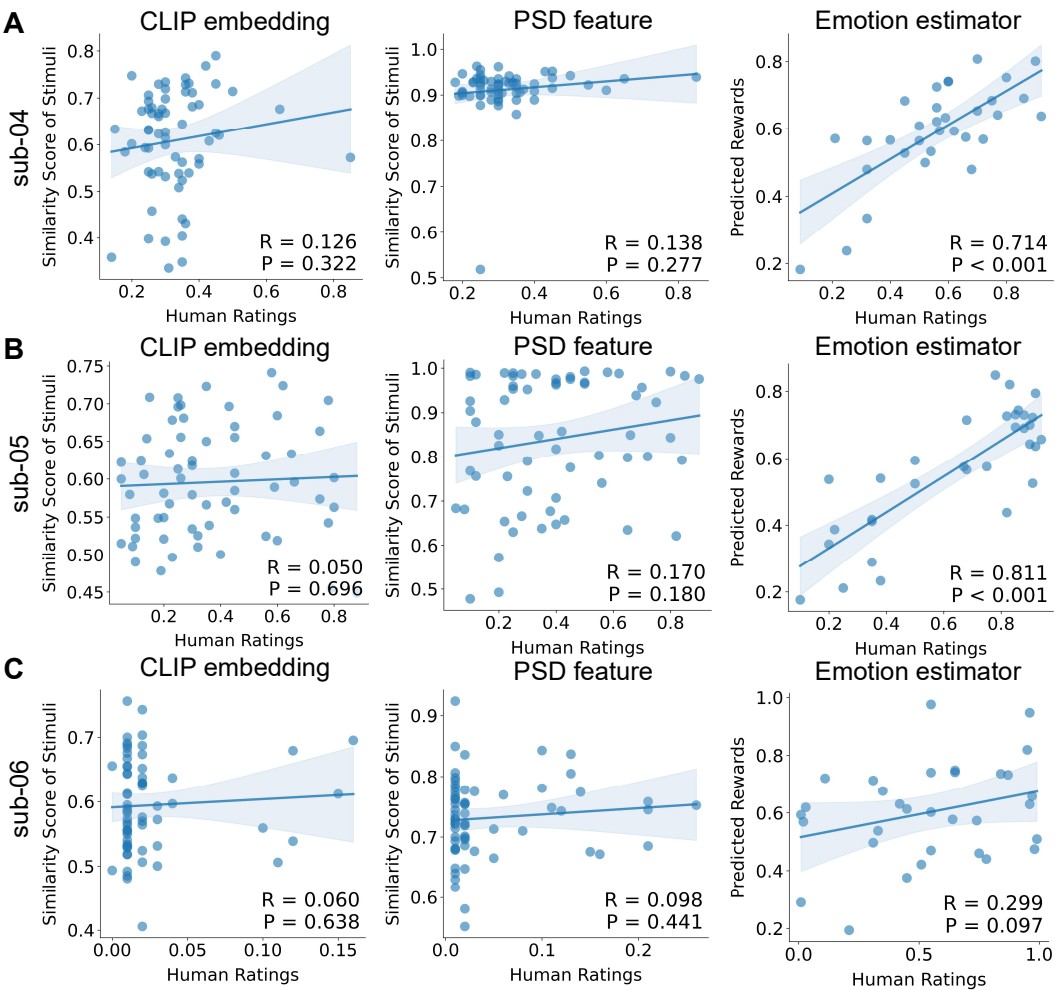

Figure A.4: **Correlations analysis for S4-S6.** (Left and Middle) Correlations between the similarity score of stimulus and human rating in the mental matching task in Target 1. (Right) Correlations between the predicted rewards from the estimator and human ratings in the emotion regulation task in Target 1.

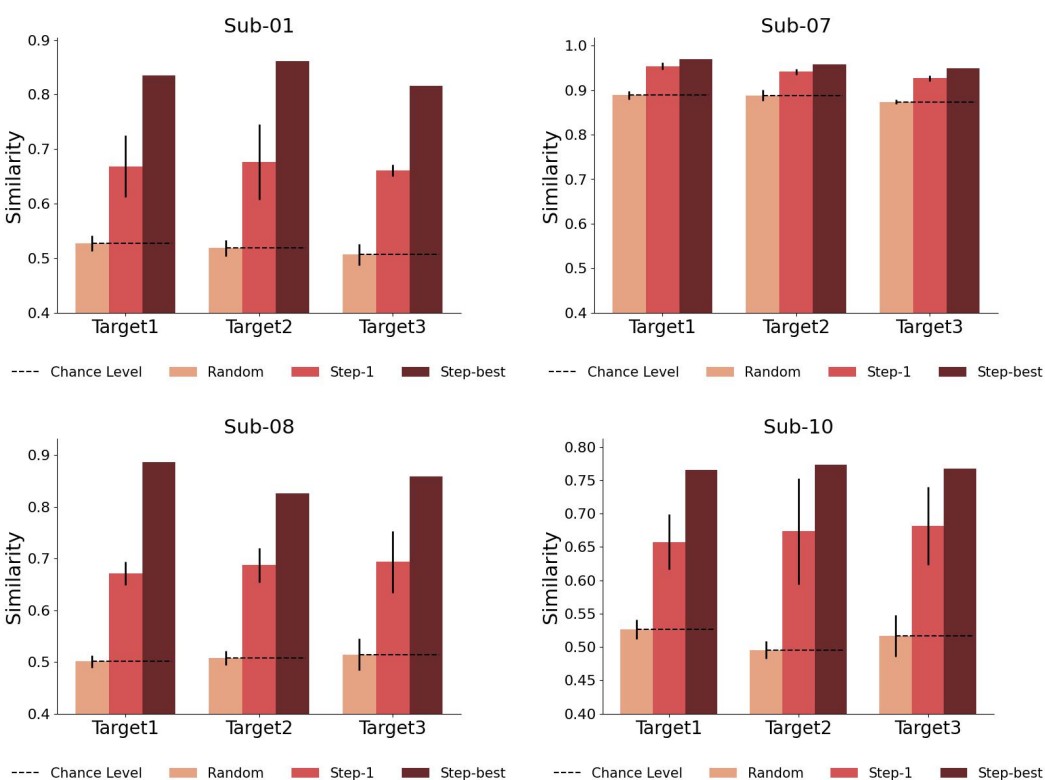

Figure A.5: **Comparison of improved performance by different targets.** We present the similarity scores of EEG features generated by random stimulation, open-loop stimulation (step 1), and step-best stimulation, in comparison to the target features. Each subject randomly selected 3 images from the searching space as target images.

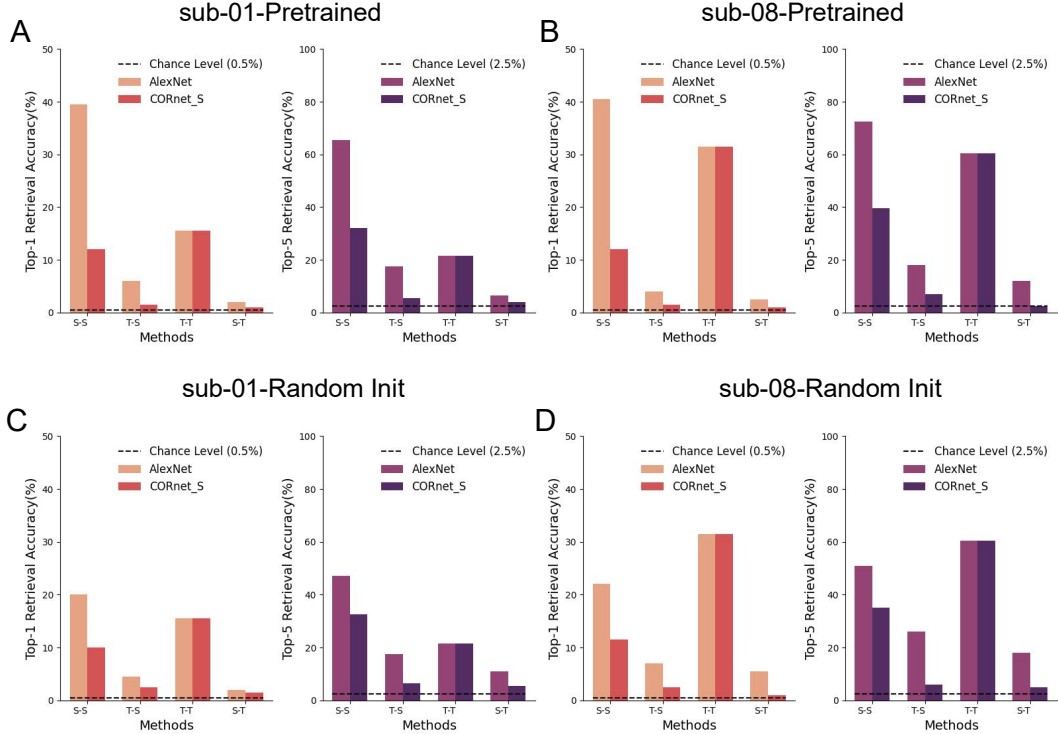

Figure A.6: Retrieval accuracy under different training and test datasets. Zero-shot retrieval performance of EEG data from different sources in Subject 1 and Subject 8 using ATM-S in different Settings. AlexNet and CORnet-S used in the first row were both pre-trained end-to-end models, and the second row was randomly initialized end-to-end.

generate synthetic EEG signals based on the visual stimuli. The model captures the mapping between the visual input and the resulting EEG signals using deep neural networks. These synthetic signals represent the neural responses predicted by the model in response to the stimuli.

Table A.5: MSE Values for synthesized EEG

| Subject | Pretrained | | Random Init | | Average |
|---------|---------|----------|---------|----------|---------|
| | AlexNet | CORnet-S | AlexNet | CORnet-S | |
| Sub-01 | 0.1095 | 0.1126 | 0.1161 | 0.0994 | 0.1094 |
| Sub-02 | 0.0764 | 0.0788 | 0.0840 | 0.0994 | 0.0847 |
| Sub-03 | 0.0787 | 0.0806 | 0.0816 | 0.0910 | 0.0830 |
| Sub-04 | 0.0652 | 0.0664 | 0.0662 | 0.1011 | 0.0747 |
| Sub-05 | 0.0493 | 0.0515 | 0.0704 | 0.0975 | 0.0672 |
| Sub-06 | 0.0690 | 0.0719 | 0.0498 | 0.0966 | 0.0718 |
| Sub-07 | 0.1267 | 0.1300 | 0.0914 | 0.1312 | 0.1198 |
| Sub-08 | 0.0718 | 0.0727 | 0.1038 | 0.1165 | 0.0912 |
| Sub-09 | 0.0529 | 0.0563 | 0.0781 | 0.0756 | 0.0657 |
| Sub-10 | 0.1122 | 0.1151 | 0.0961 | 0.1149 | 0.1096 |
| **Average** | 0.0810 | 0.0832 | 0.0838 | 0.1023 | 0.0876 |

Tab. A.5 presents the mean squared error (MSE) between the synthetic EEG signals generated by AlexNet and CORnet-S, and the real EEG signals for 10 subjects in THINGS-EEG2. The MSE was computed for each individual test sample and then averaged across the entire test set. Lower MSE values indicate better alignment between the synthetic and real EEG signals.

From the comparison shown in the Fig. A.6, the retrieval accuracy for S-S (both training and testing sets consist of generated signals) is significantly higher than other categories, including T-T (both

training and testing sets consist of real signals), T-S (training set consists of real signals, testing set consists of generated signals), and S-T (training set consists of generated signals, testing set consists of real signals), under both AlexNet and CORnet-S models. This indicates:

**Advantages of generated signals** Supported by black-box ANN models (e.g., AlexNet and CORnet-S), generated signals perform significantly better in retrieval tasks compared to real signals. In particular, the highest retrieval accuracy for S-S demonstrates the consistency and model adaptability of generated signals in this retrieval task. Different ANN models (e.g., AlexNet and CORnet-S) show consistent superiority in the retrieval tasks for generated signals, indicating that generated signals are more easily captured and distinguished by black-box proxy models.

**Correlation analysis** In Fig. A.7, for each time point, we compute the Pearson correlation between the real EEG signal and the synthetic signals. This analysis enables us to visualize how well each model replicates the temporal structure of real neural responses to visual stimuli, with shaded regions representing the standard deviation across samples. Notably, the results reveal that supervised models (e.g., ResNet) frequently outperform representation learning models (e.g., CLIP) in capturing these temporal dynamics. This contrasts with fMRI studies prioritizing semantic alignment (Wang et al., 2023), likely due to the high temporal resolution of non-invasive EEG, which is inherently more sensitive to transient, low-to-mid-level visual features (e.g., edges, textures) during early processing stages (Groen et al., 2017). ResNet, optimized for hierarchical feature extraction, aligns better with these "shallower" visual dynamics than the abstract semantic representations of CLIP. Furthermore, the dimensionality reduction (PCA) inherent to the encoding pipeline may favor the robust hierarchical features of ResNet over the complex joint-space representations of contrastive models for the specific task of Image-to-EEG synthesis.

**Variance analysis** In Fig. A.8, we compute the variance across all samples and time points for each channel, providing a measure of the overall variability of the EEG signals in response to different visual stimuli and their temporal dynamics. This variance can help identify channels with the highest variability, which may be useful for selecting specific channels for further analysis or modulation. In Fig. A.9, we show the variance and standard deviation of the EEG signals computed across samples for each time point, and then averaged across channels. This analysis allows us to assess how signal variability evolves over time. By comparing the real EEG data with synthetic data generated by AlexNet and CORnet-S, we can evaluate how well each model captures the temporal variability present in the real EEG signals.

**Correlation distribution analysis** In Fig. A.10, we compute the pearson correlation coefficient between the averaged real EEG data and the synthetic data for each stimulus, measuring how well the synthetic data matches the real EEG on a per-sample basis. The histogram shows the distribution of correlation coefficients across all samples for both AlexNet and CORnet-S. A higher concentration of peaks near higher Pearson coefficients indicates better alignment between the synthetic data and the real EEG, reflecting superior model performance.

These findings highlight the robustness of our EEG encoding models, demonstrating their ability that not only mimic the structural features of real EEG data but also capture the realistic variability seen in neural responses to visual stimuli. This suggests that our models are effective in approximating the neural representations underlying visual processing.

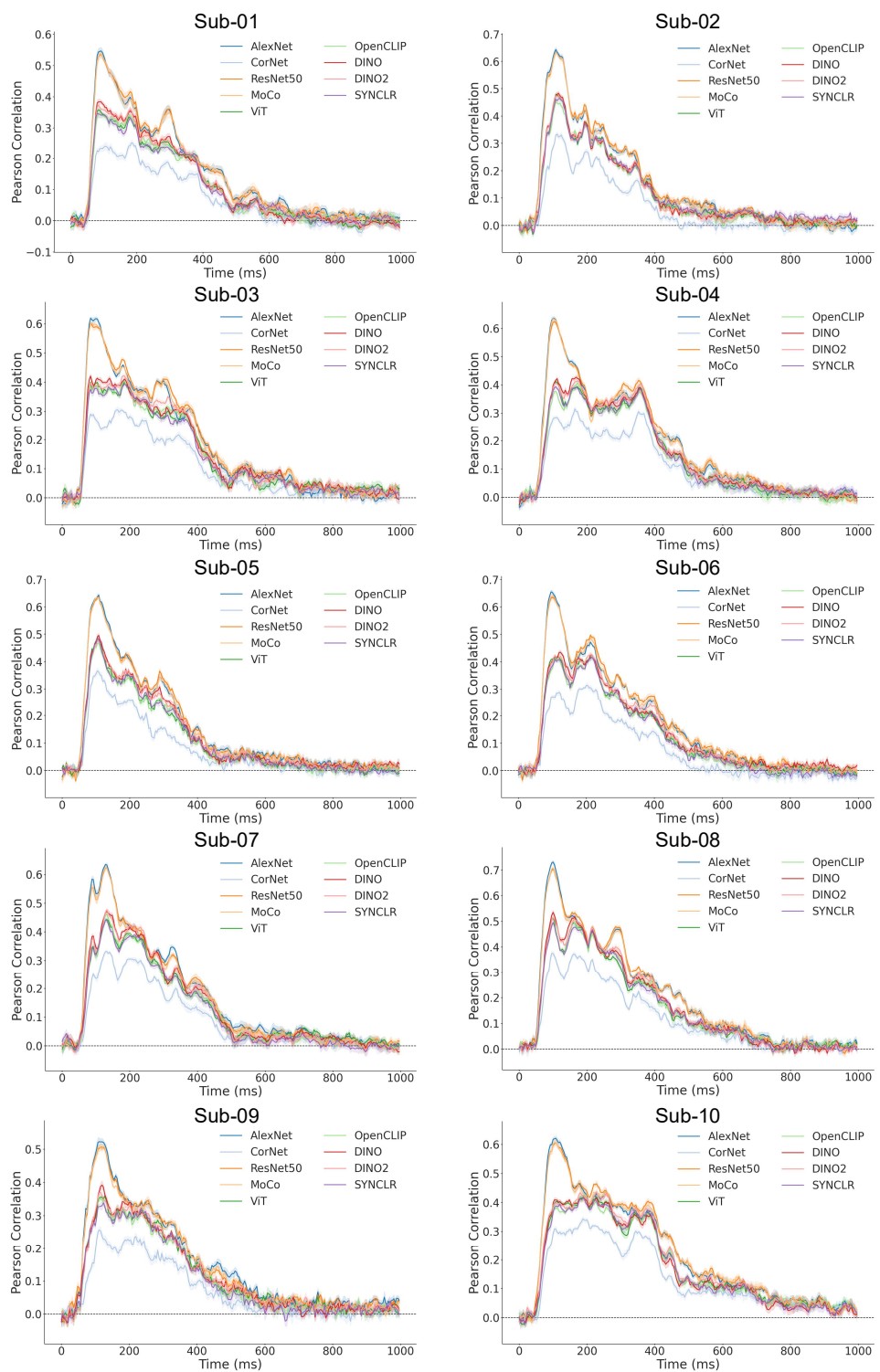

Figure A.7: Time-resolved pearson correlation between ground truth and synthetic EEG signals. These synthetic signals were generated using a linearizing model fitted on feature maps from a range of pre-trained deep neural networks (AlexNet, CORnet-S, ResNet50, MoCo, ViT-B-32, OpenCLIP-ViT-B-32, DINO2-ViT-B-14, DINO-ViT-B-16, and SYNCLR-ViT-B-16).

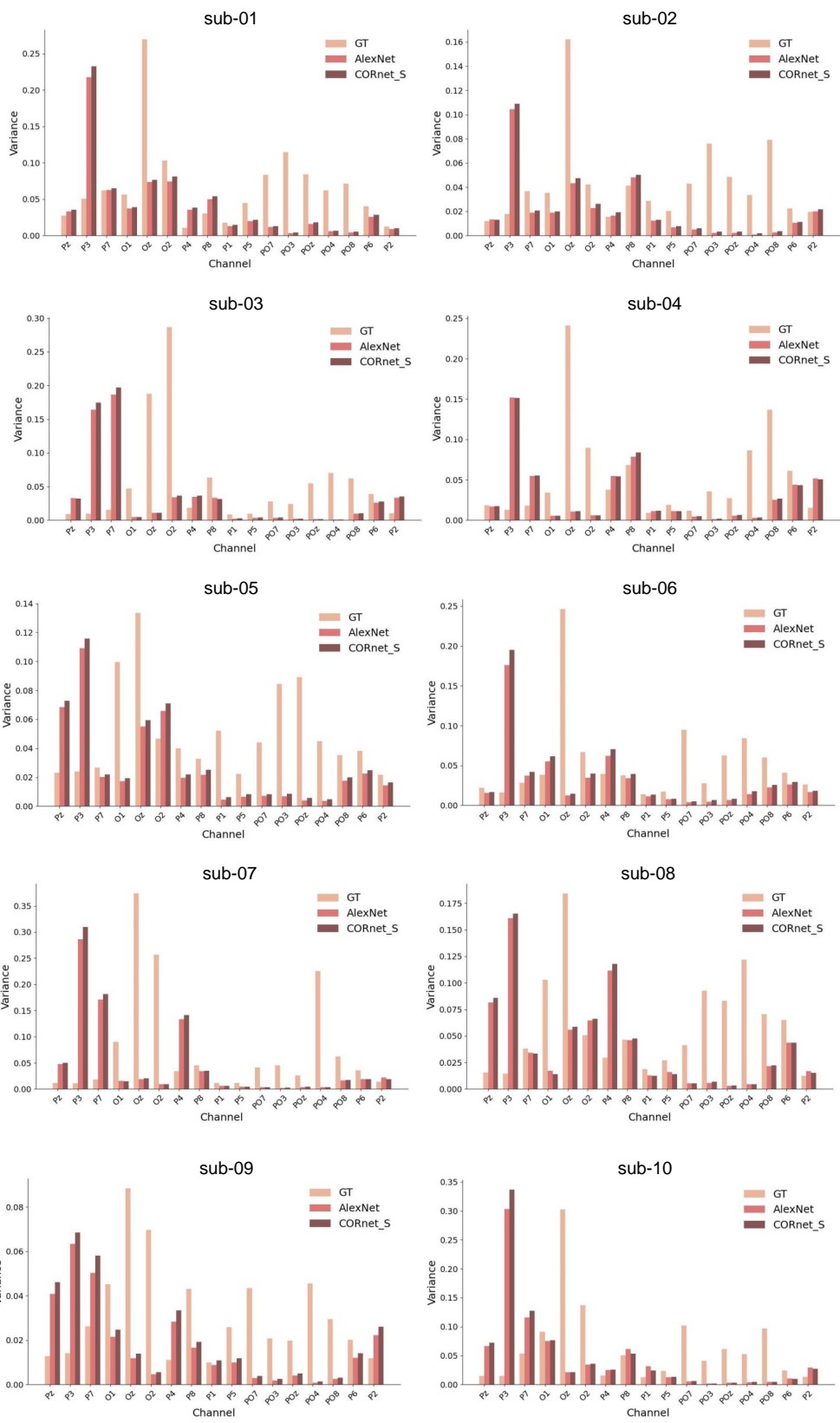

Figure A.8: Variance across different channels for different visual stimulus and temporal dynamics

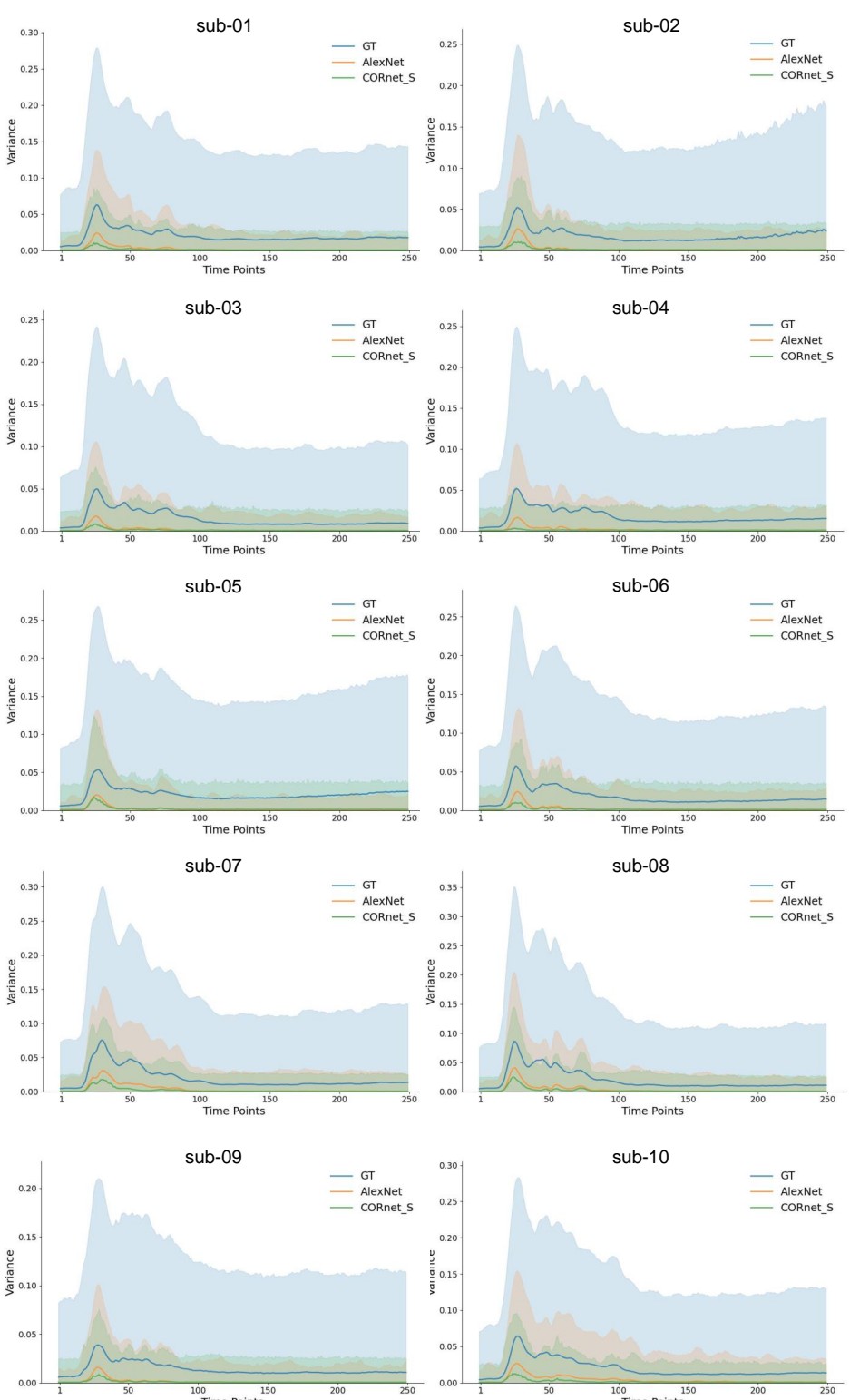

Figure A.9: Variance across different time points for different visual stimuli and channels.

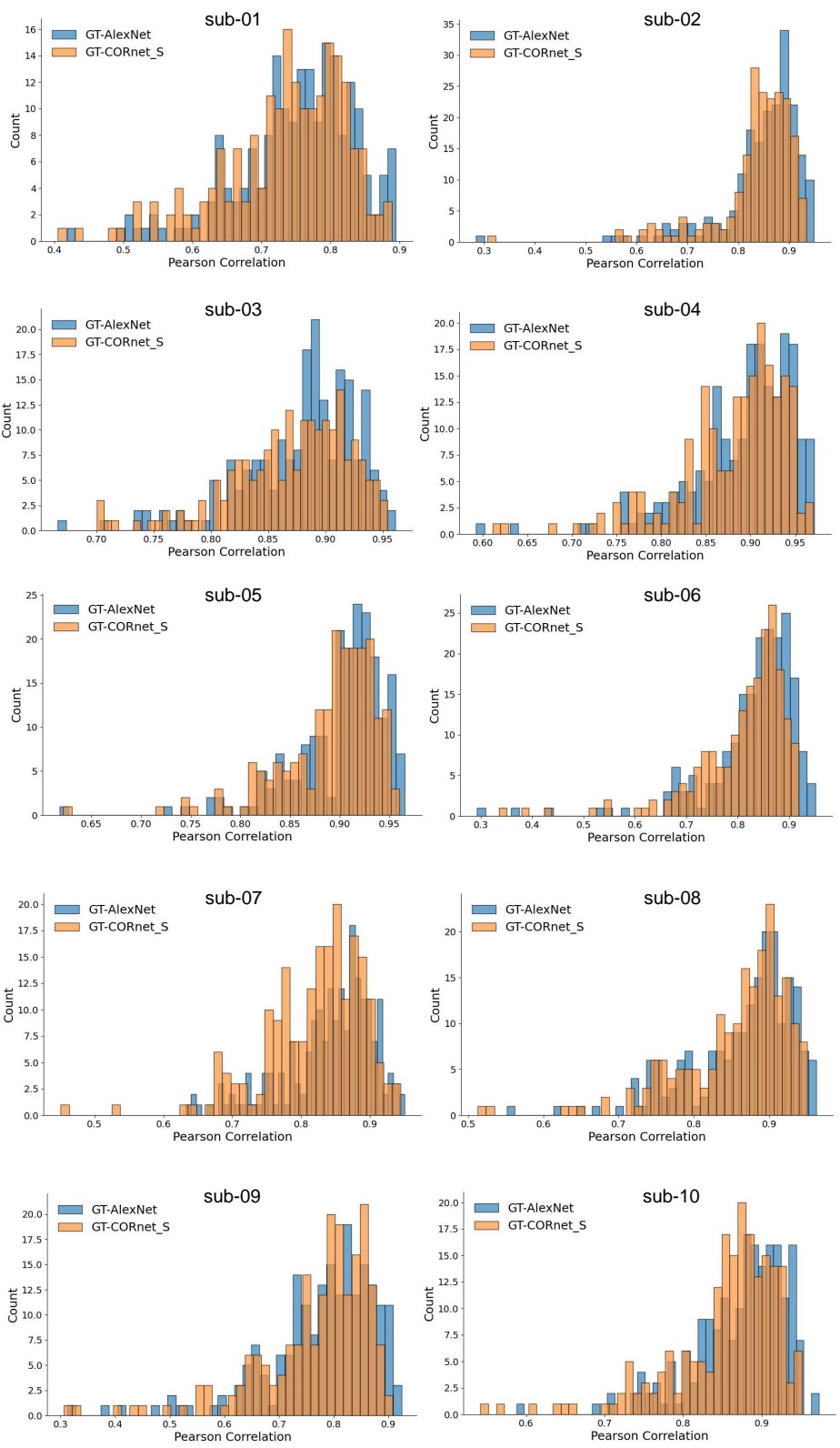

Figure A.10: Distribution of pearson correlation coefficients across all sample pairs.

## A.4 ADDITIONAL SEARCHING EXAMPLES OF SEMANTIC REPRESENTATION

### A.4.1 MORE EXAMPLES OF INTERACTIVE SEARCHING

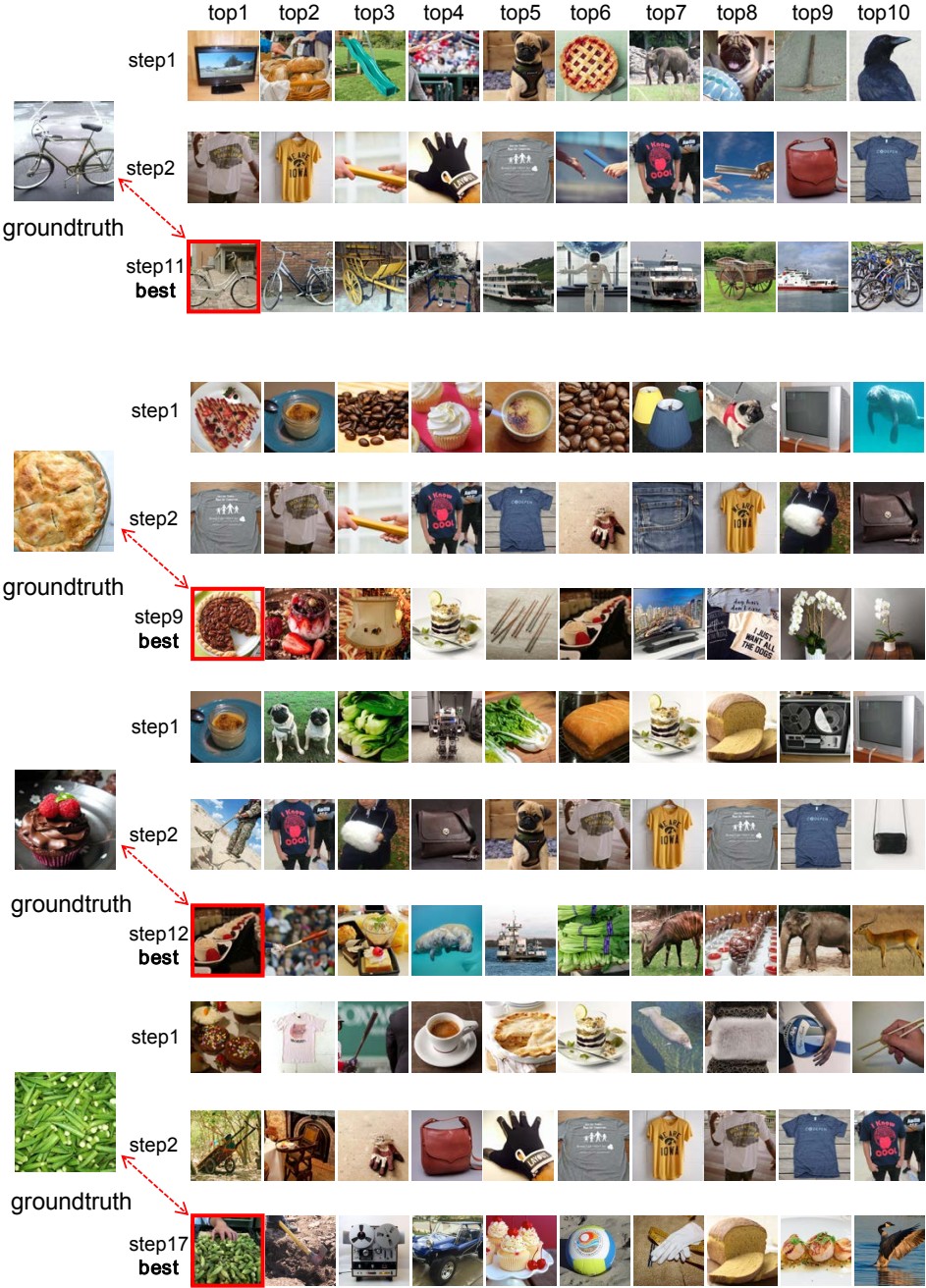

Figure A.11: **Some searching examples of Subject 8, 4, 4, and 1.** By setting different targets, we present examples where the stimulus retrieved at the end of the iterative optimization process increasingly approximates the true category.

### A.4.2 SOME FAILURE EXAMPLES OF INTERACTIVE SEARCHING

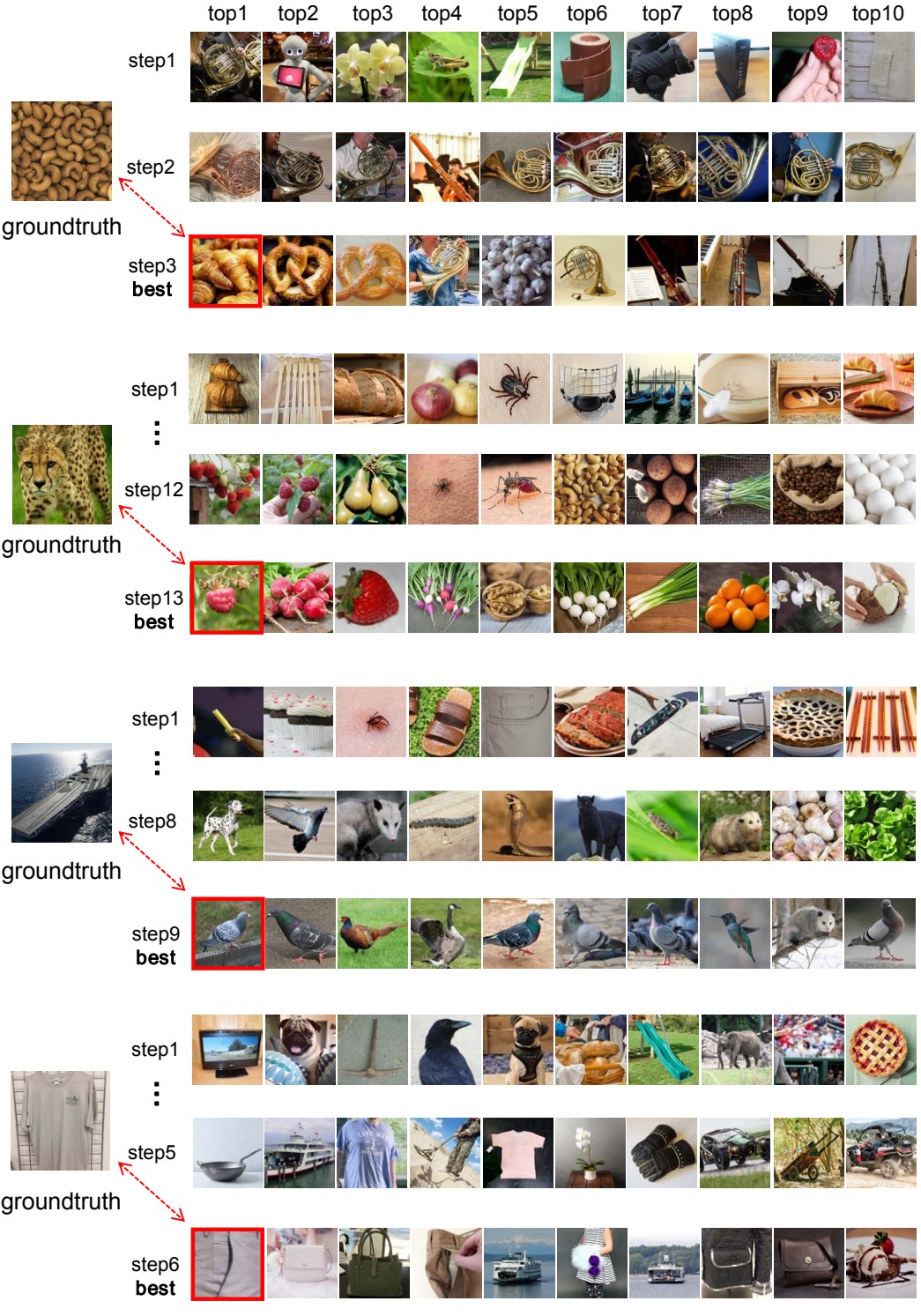

Figure A.12: **Some failure examples from Subject 8.** By setting different targets, we show examples where the stimulus retrieved at the end of the iteration is far from the true category. In these examples, the final retrieved stimulus exhibits varying degrees of similarity to the target image.

## A.5 ADDITIONAL CONTROLLABLE GENERATION EXAMPLES OF PSD FEATURE

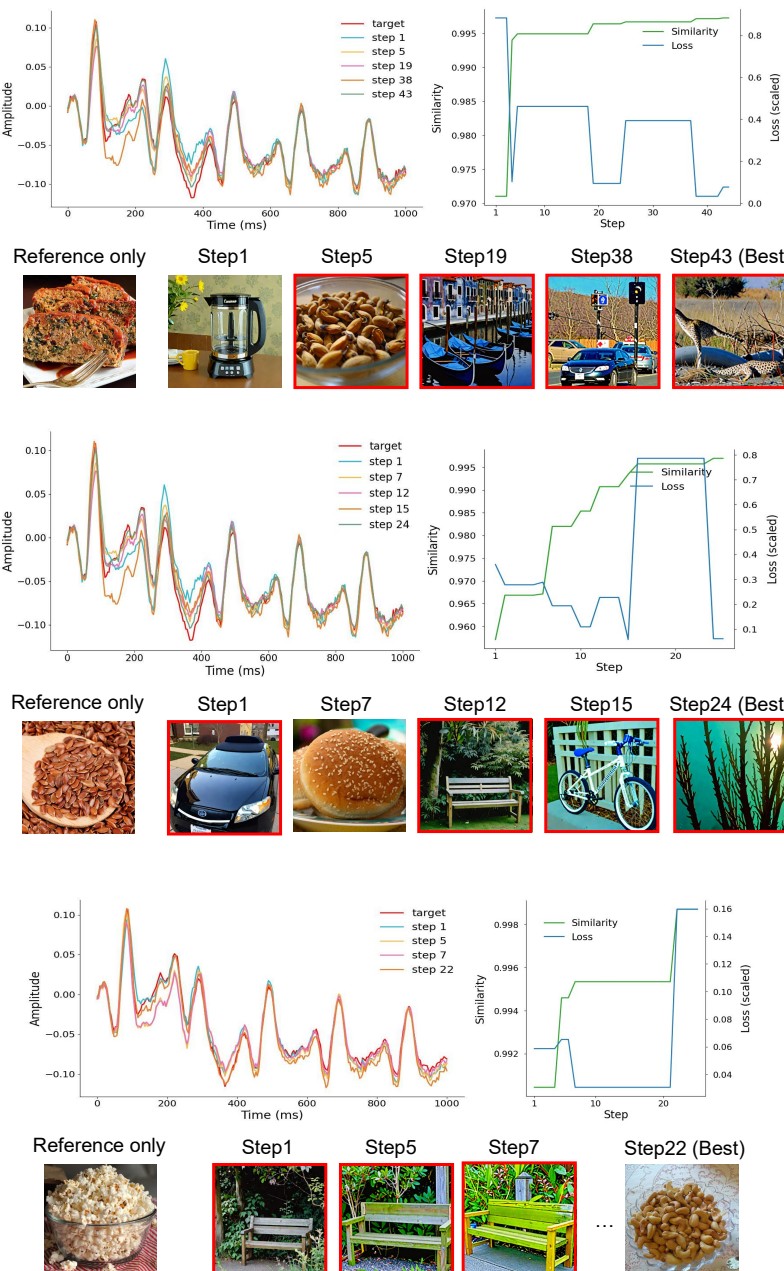

Figure A.13: **Illustration of the closed-loop iterative process for Subject 1.** Three distinct visual targets were presented, each based on a specific similarity measure (details in Target Features of EEG, Section 4.1), with new visual stimuli iteratively generated for each target. The left panel illustrates the time-domain evolution of neural responses across iterations. The right panel depicts the changes in similarity (green curve) and loss (blue curve, scaled) between the current stage features and the target features.

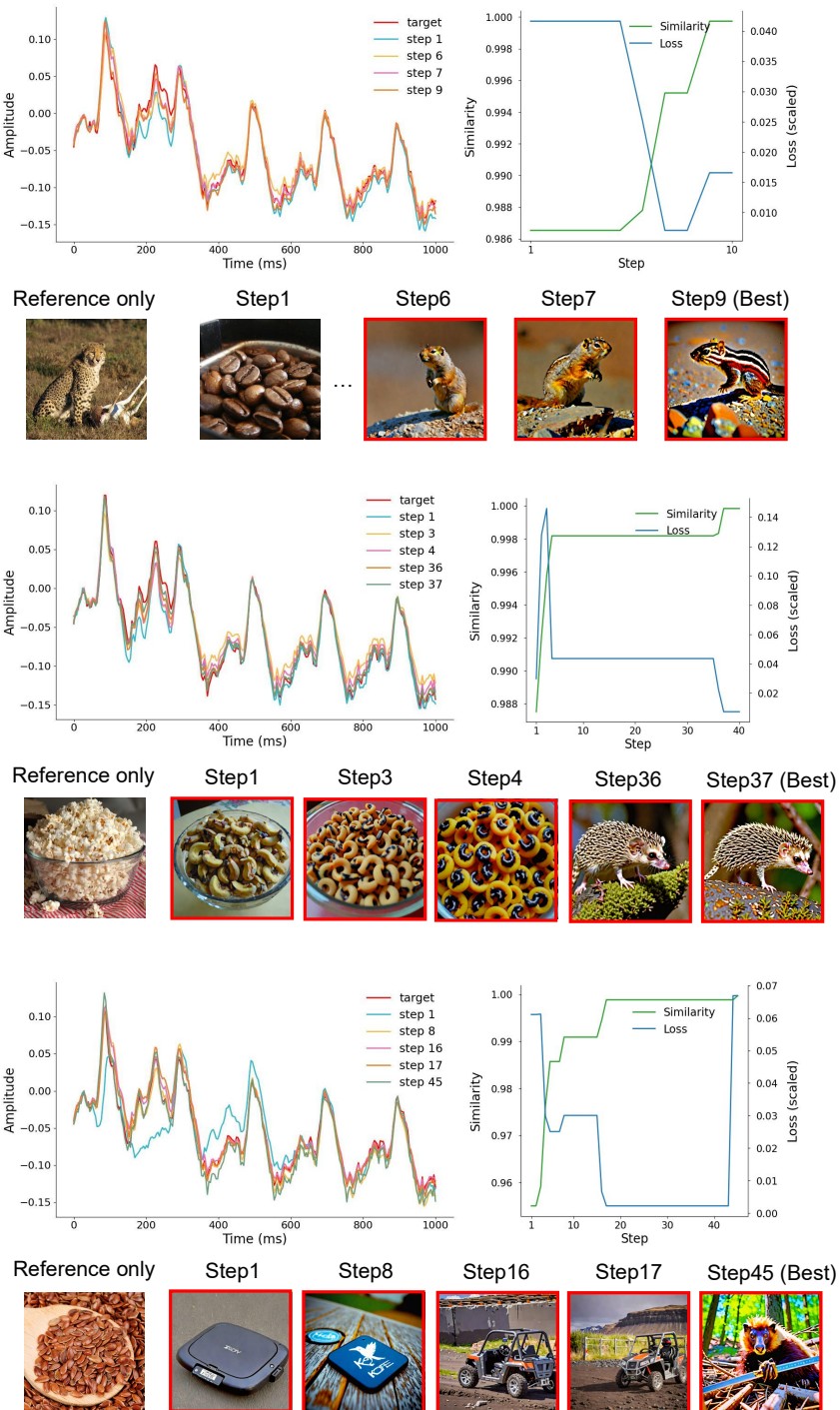

Figure A.14: **Illustration of the closed-loop iterative process for Subject 2.** Three distinct visual targets were presented, each based on a specific similarity measure (details in Target Features of EEG, Section 4.1), with new visual stimuli iteratively generated for each target. The left panel illustrates the time-domain evolution of neural responses across iterations. The right panel depicts the changes in similarity (green curve) and loss (blue curve, scaled) between the current stage features and the target features.

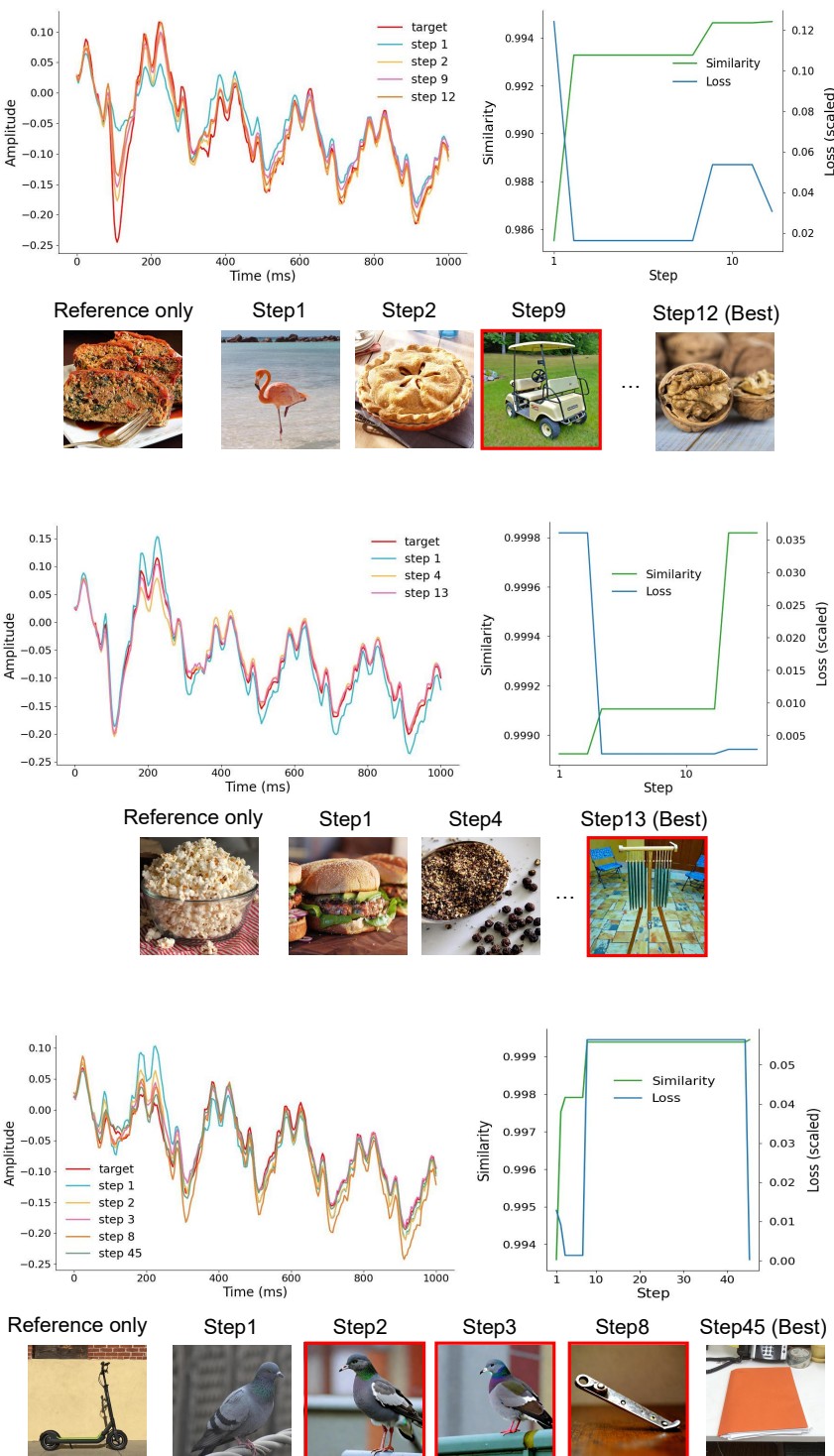

Figure A.15: **Illustration of the closed-loop iterative process for Subject 3.** Three distinct visual targets were presented, each based on a specific similarity measure (details in Target Features of EEG, Section 4.1), with new visual stimuli iteratively generated for each target. The left panel illustrates the time-domain evolution of neural responses across iterations. The right panel depicts the changes in similarity (green curve) and loss (blue curve, scaled) between the current stage features and the target features.

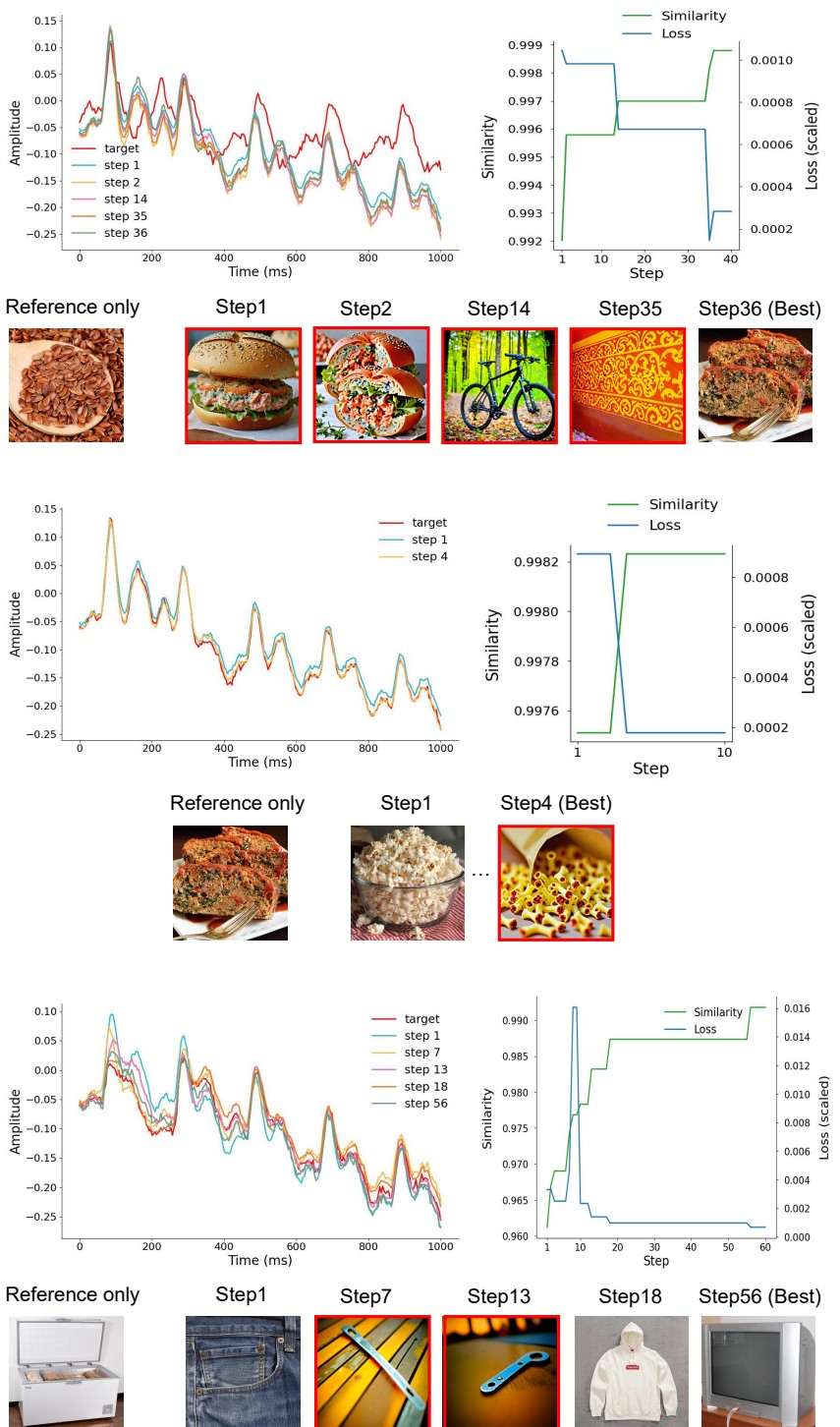

Figure A.16: **Illustration of the closed-loop iterative process for Subject 4.** Three distinct visual targets were presented, each based on a specific similarity measure (details in Target Features of EEG, Section 4.1), with new visual stimuli iteratively generated for each target. The left panel illustrates the time-domain evolution of neural responses across iterations. The right panel depicts the changes in similarity (green curve) and loss (blue curve, scaled) between the current stage features and the target features.

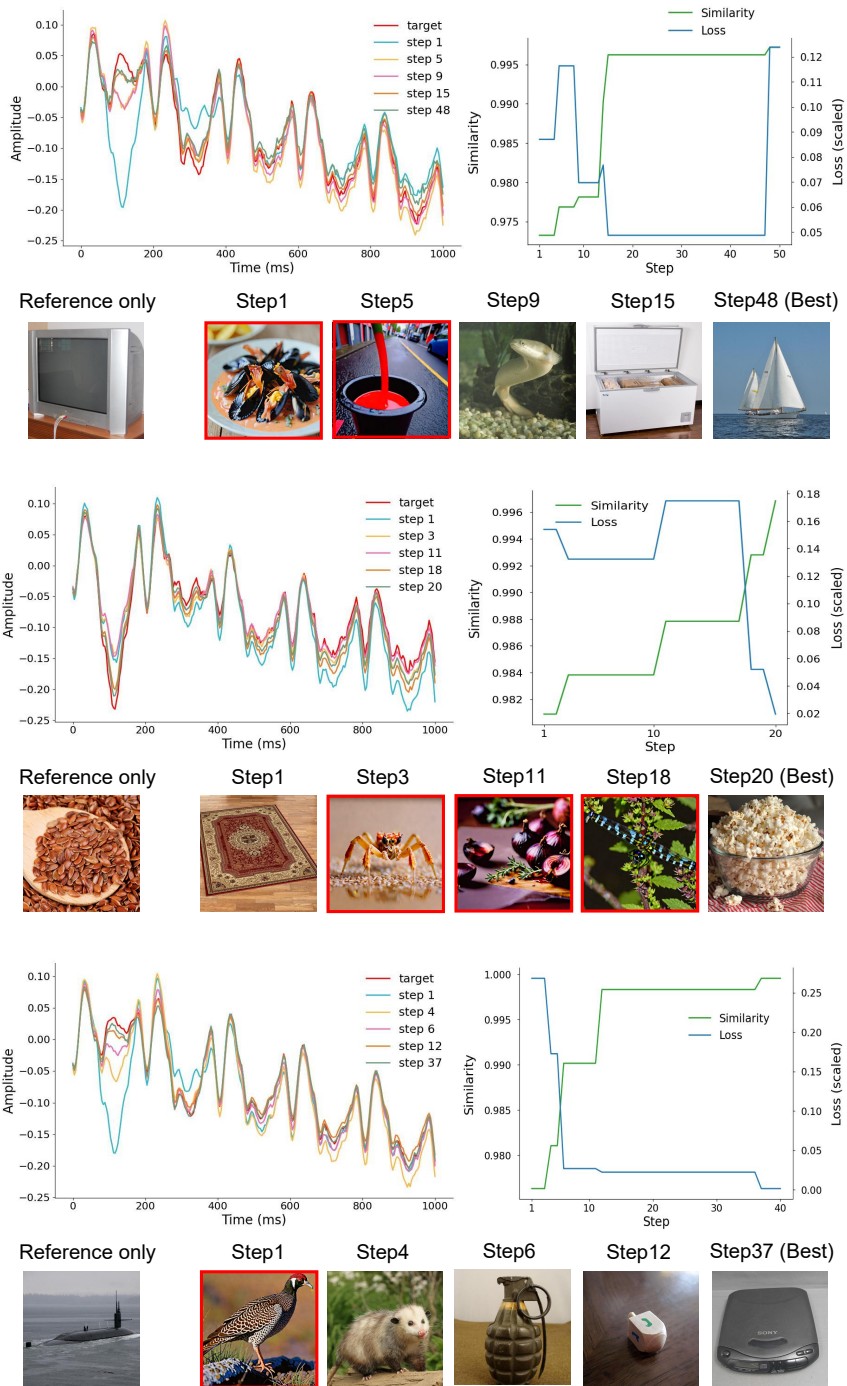

Figure A.17: **Illustration of the closed-loop iterative process for Subject 5.** Three distinct visual targets were presented, each based on a specific similarity measure (details in Target Features of EEG, Section 4.1), with new visual stimuli iteratively generated for each target. The left panel illustrates the time-domain evolution of neural responses across iterations. The right panel depicts the changes in similarity (green curve) and loss (blue curve, scaled) between the current stage features and the target features.

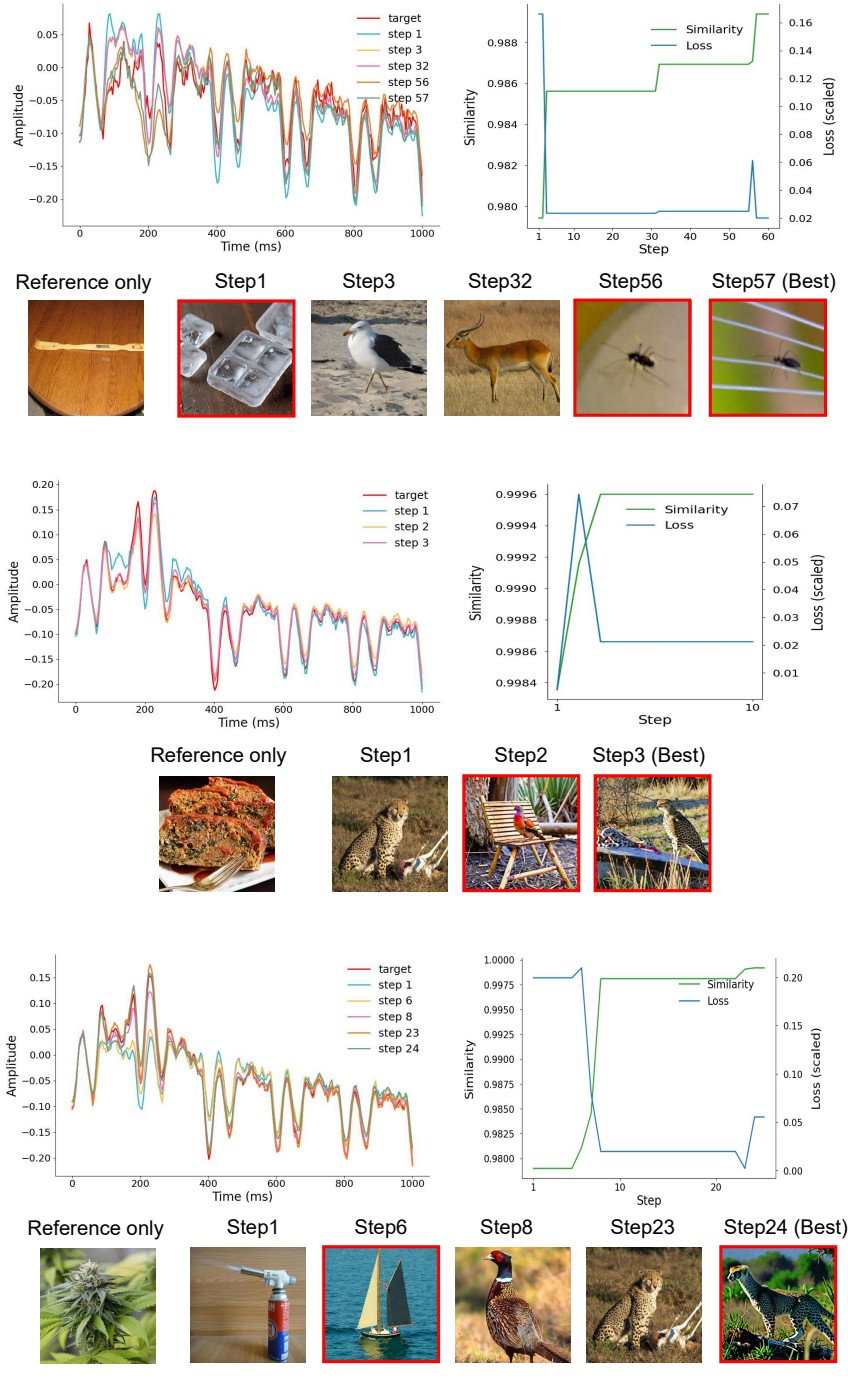

Figure A.18: **Illustration of the closed-loop iterative process for Subject 6.** Three distinct visual targets were presented, each based on a specific similarity measure (details in Target Features of EEG, Section 4.1), with new visual stimuli iteratively generated for each target. The left panel illustrates the time-domain evolution of neural responses across iterations. The right panel depicts the changes in similarity (green curve) and loss (blue curve, scaled) between the current stage features and the target features.

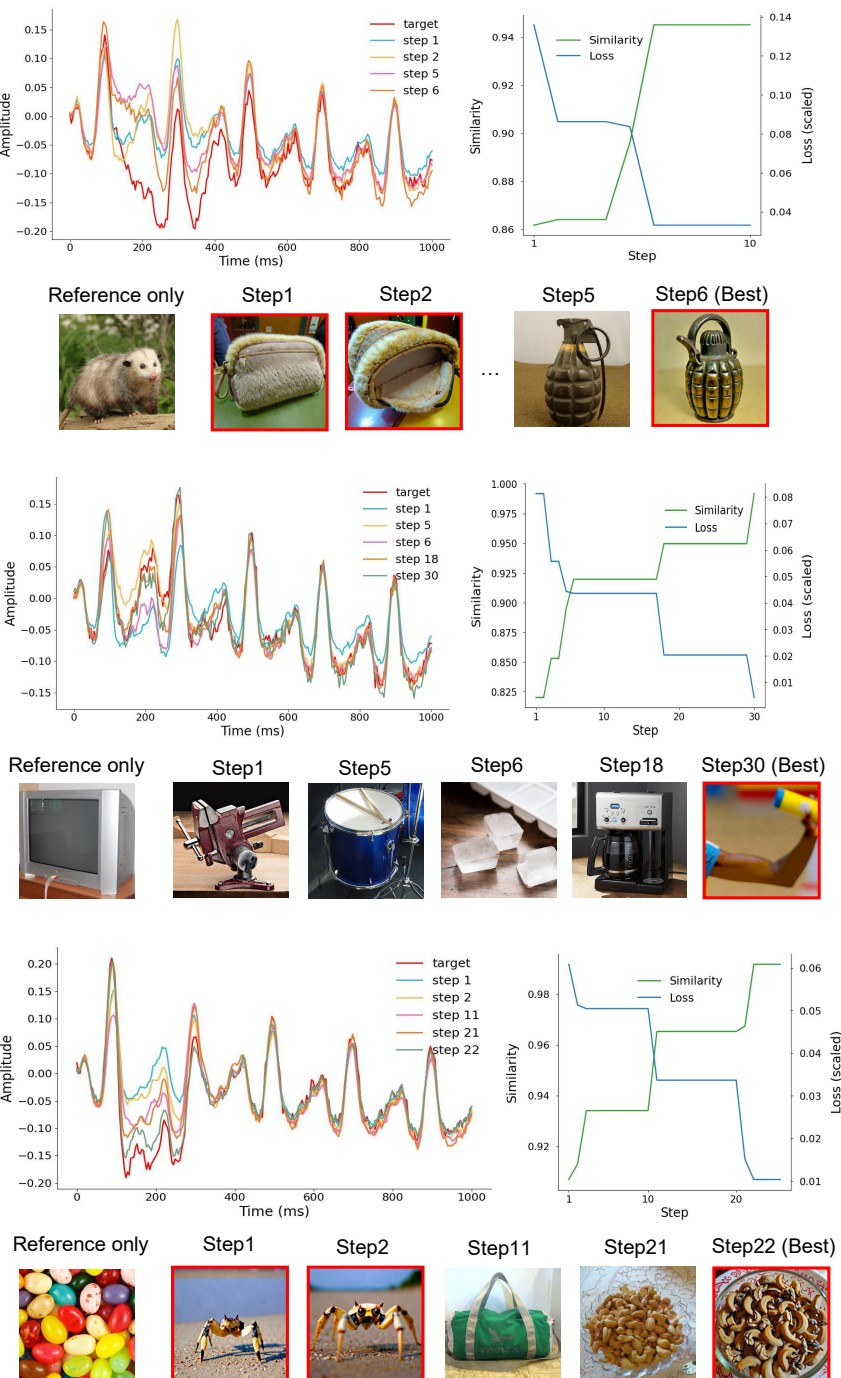

Figure A.19: **Illustration of the closed-loop iterative process for Subject 7.** Three distinct visual targets were presented, each based on a specific similarity measure (details in Target Features of EEG, Section 4.1), with new visual stimuli iteratively generated for each target. The left panel illustrates the time-domain evolution of neural responses across iterations. The right panel depicts the changes in similarity (green curve) and loss (blue curve, scaled) between the current stage features and the target features.

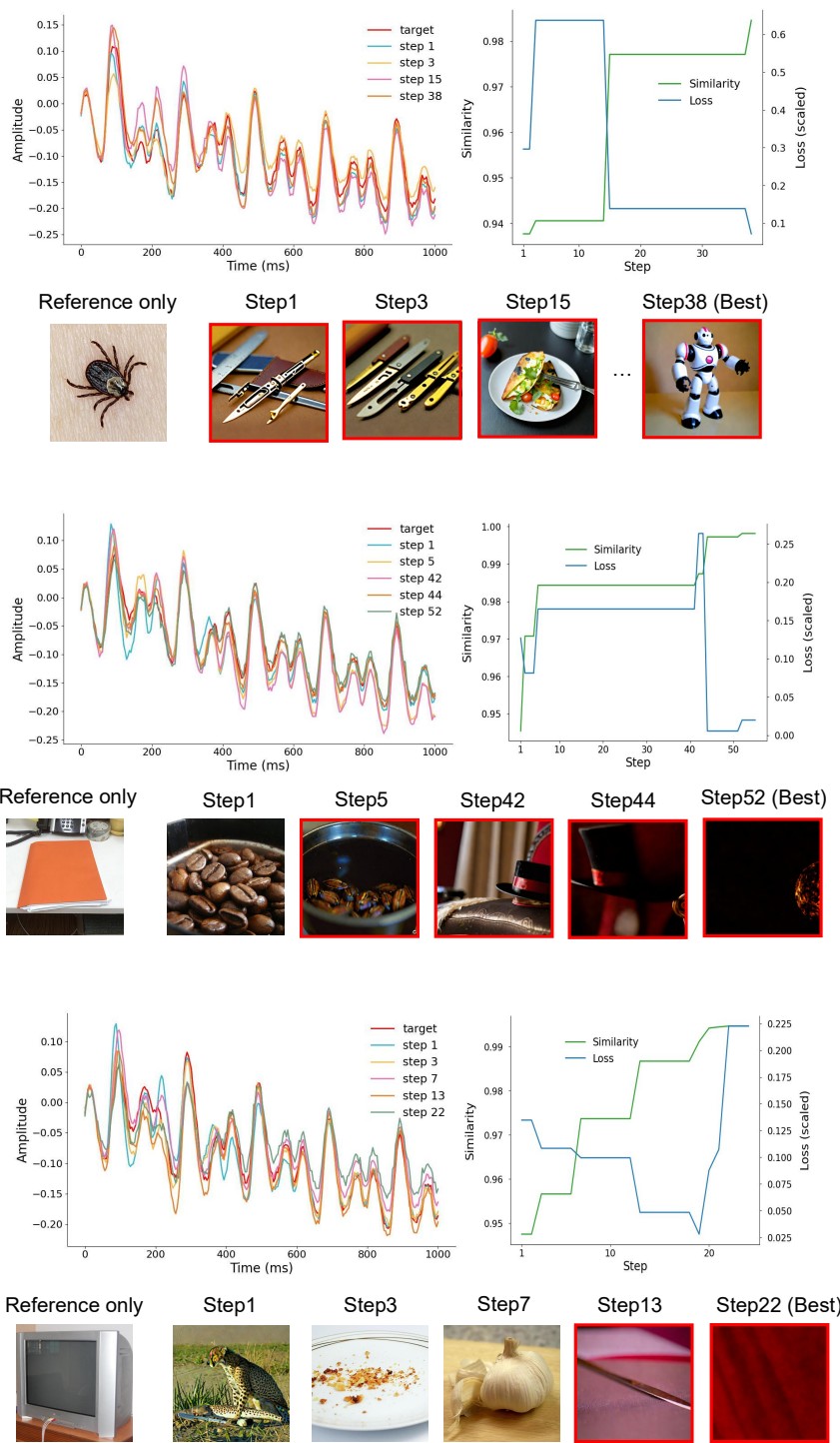

Figure A.20: **Illustration of the closed-loop iterative process for Subject 8.** Three distinct visual targets were presented, each based on a specific similarity measure (details in Target Features of EEG, Section 4.1), with new visual stimuli iteratively generated for each target. The left panel illustrates the time-domain evolution of neural responses across iterations. The right panel depicts the changes in similarity (green curve) and loss (blue curve, scaled) between the current stage features and the target features.

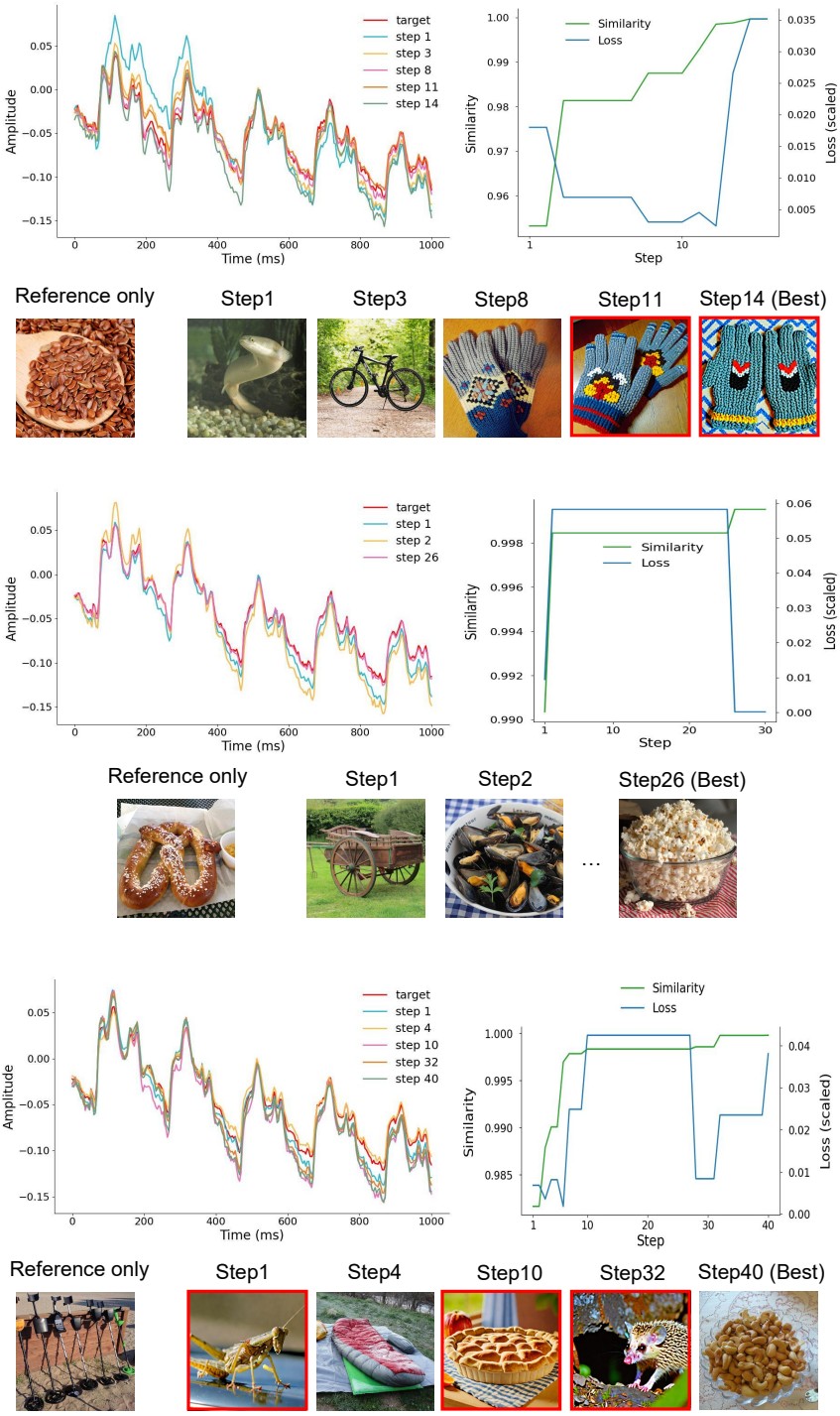

Figure A.21: **Illustration of the closed-loop iterative process for Subject 9.** Three distinct visual targets were presented, each based on a specific similarity measure (details in Target Features of EEG, Section 4.1), with new visual stimuli iteratively generated for each target. The left panel illustrates the time-domain evolution of neural responses across iterations. The right panel depicts the changes in similarity (green curve) and loss (blue curve, scaled) between the current stage features and the target features.

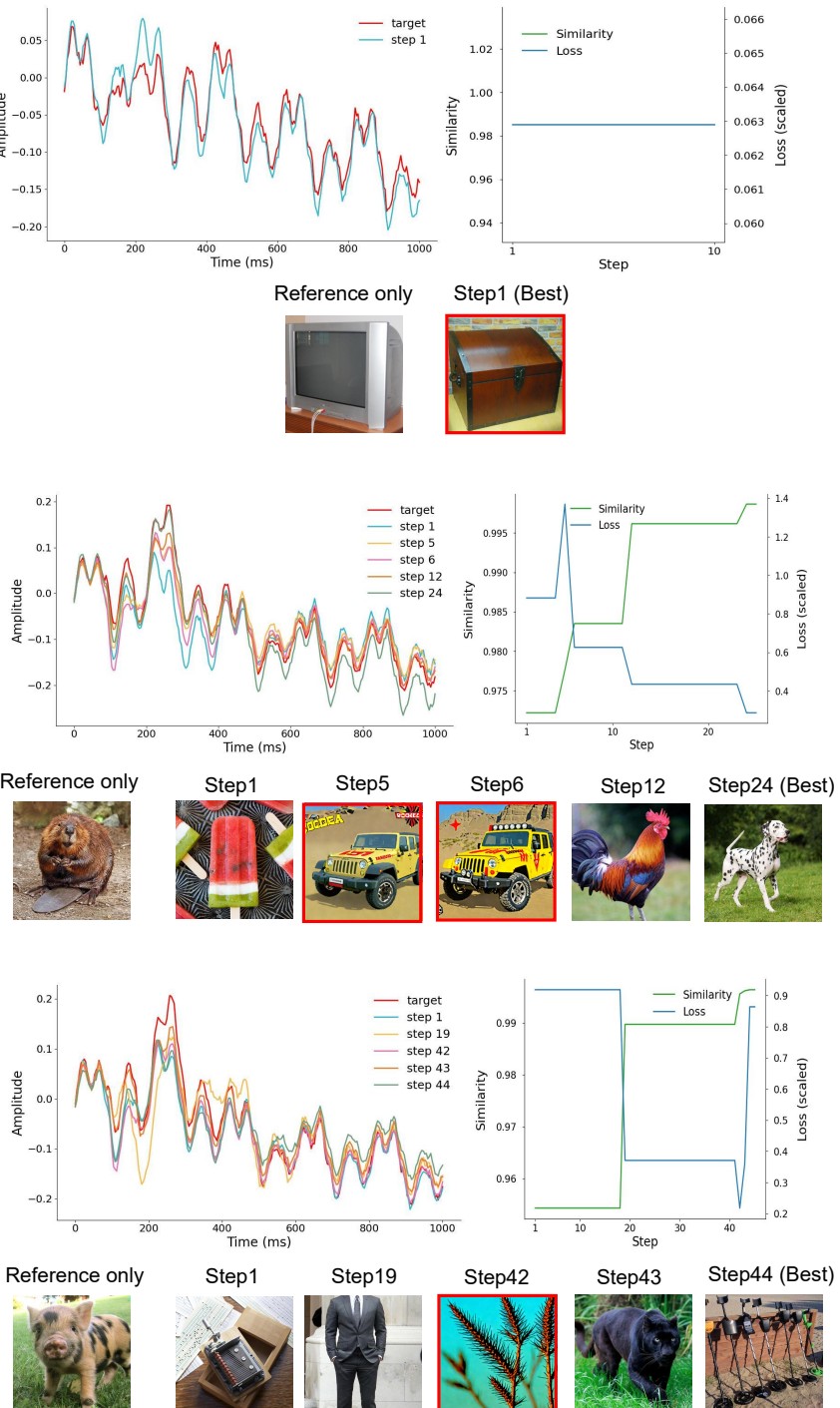

Figure A.22: **Illustration of the closed-loop iterative process for Subject 10.** Three distinct visual targets were presented, each based on a specific similarity measure (details in Target Features of EEG, Section 4.1), with new visual stimuli iteratively generated for each target. The left panel illustrates the time-domain evolution of neural responses across iterations. The right panel depicts the changes in similarity (green curve) and loss (blue curve, scaled) between the current stage features and the target features.

