# OpenReview forum: "MindPilot: Closed-loop Visual Stimulation Optimization for Brain Modulation with EEG-guided Diffusion"
_ICLR.cc/2026/Conference — ICLR 2026 Poster_

### Official Review · Reviewer_LDfZ · 2025-10-27

**Soundness:** 3
**Presentation:** 3
**Contribution:** 3
**Rating:** 6
**Confidence:** 4

**Summary:**

This paper presents MindPilot, an innovative closed-loop EEG-guided visual stimulus optimization framework.The core concept of this framework is to treat the human brain as an indivisible black-box system, iteratively guiding diffusion models to generate images through EEG feedback signals.This framework introduces a pseudo-model to provide surrogate gradients, enabling gradient-free optimization for various neural objectives such as semantic or spectral EEG features.The paper validates MindPilot through agent-based model simulations and human experiments, demonstrating its potential in EEG-guided image retrieval, generation, and real-time human brain control.

**Strengths:**

Novelty: Innovation: First to propose a closed-loop image generation framework based on EEG signal diffusion models.

Technical soundness: Technical Rationality: The pseudo-model guidance mechanism is clearly designed and effectively replaces explicit gradient computation.

Experimental breadth: Demonstrates stable convergence characteristics in both simulated and human experiments.

Potential impact: Openes up new avenues for non-invasive neural modulation and human-machine collaborative adaptation studies.

**Weaknesses:**

EEG Individual Variability: Insufficient discussion of inter-subject EEG variability; a brief explanation could be added to emphasize the model's adaptability and generalization capability.

**Questions:**

1.  Could the authors elaborate on whether MindPilot could be generalized to other modalities (e.g. fMRI)?

2. How do pseudo-model guidance and traditional gradient-free reinforcement learning methods differ in their convergence properties?

3. Additionally, there are minor inconsistencies in the formatting of references within the text. For example, several NeurIPS/ICLR-style citations only provide the conference name without volume, issue, or page information (e.g., Bashivan et al., 2019; Black et al., 2024; Luo et al., 2024a). Similarly, the citation for “Transactions on Machine Learning Research, 2024” (Oquab et al., 2024) omits both volume and page numbers. We recommend that the authors carefully review these entries and align them with the journal article format, ensuring that volume, issue, and page numbers are included where available.

---

> ### Author Response · Authors · 2025-11-21
> **Rebuttal for R-LDfZ (1/2)**
>
> **Q1: "EEG Individual Variability...a brief explanation could be added to emphasize the model's adaptability and generalization capability."**
>
> Response to Q1 (EEG Variability & Adaptability): We thank the reviewer for this insightful and constructive suggestion. We agree that addressing inter-subject variability is crucial for emphasizing MindPilot's adaptability.
>
> 1. To explicitly handle this variability, MindPilot adopts a Subject-Specific Proxy Model strategy (Section 4.1). Instead of a generic "one-size-fits-all" decoder, we train a unique proxy for each participant. This allows the optimizer to align with the unique manifold of each individual's neural response, theoretically ensuring generalization capability across different subjects.
>
> 2. As the reviewer noted, real-world variability is pronounced. Our experiments reflect this:
> - In the human-in-the-loop emotion task, the group-level result was statistically significant, demonstrating that our framework successfully adapts to individual emotional patterns.
>
> - We observed higher variance in semantic matching (as shown in Appendix Fig. A.4 & A.5). We attribute this not to model failure, but to extrinsic factors: differential visual sensitivity (e.g., varying preferences for semantic vs. affective features) and attention fluctuations (fatigue during long trials).
>
> **Q2: "Could the authors elaborate on whether MindPilot could be generalized to other modalities (e.g. fMRI)?"**
>
> Yes, MindPilot’s black-box diffusion guidance is modality-agnostic and can generalize to fMRI — but with critical caveats due to temporal resolution.
>
> - Closed-loop (real-time): Not feasible with fMRI. Hemodynamic delay (~4–6s) prevents sub-second feedback loops required for iterative optimization. EEG’s millisecond resolution is essential for closed-loop brain modulation (e.g., visual rehabilitation through closed-loop stimulation [1]).
>
> - Open-loop optimization: fMRI’s voxel-level spatial precision enables fine-grained semantic/spectral targeting (e.g., V1, V2, FFA, PPA, amygdala). Our pseudo-model + gradient-free guidance can optimize stimuli offline using fMRI guidance, similar to [2,3,4], etc.. We have also cited these papers in the updated PDF.
>
> 3. Following your suggestion, we have added a discussion in the Discussion section (Line 534): "MindPilot mitigates EEG heterogeneity via subject-specific proxy training. While individual performance varies due to physiological state (e.g., fatigue) and intrinsic visual sensitivity, the framework demonstrates robust group-level modulation feasibility, particularly in emotion regulation tasks."
>
> **Q3: "How do pseudo-model guidance and traditional gradient-free reinforcement learning methods differ in their convergence properties?"**
>
> Response to Q3 (Convergence: Pseudo-Model vs. Gradient-Free RL): We thank the reviewer for this insightful theoretical question. As briefly touched upon in our Global Response (Point 2), the fundamental difference lies in how the two approaches interact with the data manifold.
>
> We provide a detailed breakdown of their convergence properties below:
>
> 1. Gradient Estimation: "Blind Search" vs. "Manifold-Aware Guidance"
>
> - Gradient-Free RL (e.g., DDPO [5], DNO [6]): Relies on zero-order optimization (e.g., finite differences or random perturbations).(1) This is essentially a "blind search" in high-dimensional space. It is manifold-agnostic, meaning updates often drift off the valid data manifold, leading to degenerate samples (e.g., adversarial noise or blurry images) and requiring high query volume to estimate a stable direction.
>
> - MindPilot (Ours): Utilizes Implicit Gradient Approximation. By leveraging the pseudo-model (Gaussian Process), we compute an update direction that is mathematically constrained to the low-dimensional subspace spanned by historical data (Proposition 1). This acts as a "compass," ensuring every update remains semantically meaningful and on-manifold, avoiding the high variance of random search.
>
> 2. Query Efficiency & Speed
>
> - RL: Suffers from the "cold start" problem. Without a surrogate model to map the landscape, it requires more than ten times the delay (10\times) compared with Mindpilot to estimate gradients via brute-force averaging (Table 1).
>
> - MindPilot: Achieves near-optimal scores in just 6–50 queries. The pseudo-model acts as a "learned proxy," allowing the system to extrapolate the optimal direction from sparse history rather than exploring from scratch.

---

> ### Author Response · Authors · 2025-11-21
> **Rebuttal for R-LDfZ (2/2)**
>
> 3. Robustness to Objective Noise
>
> - RL: High susceptibility. Finite-difference methods tend to amplify the inherent noise in single-trial EEG/black-box feedback, leading to unstable convergence or local stagnation.
>
> - MindPilot: High resilience. The universal update direction ($x^* - x_K$) derived from the proxy model inherently filters high-frequency noise via manifold projection (Figure 1), guiding generation stably even with noisy targets.
>
> Summary Comparison:
>
> | Property | Pseudo-Model Guidance (Ours) | Gradient-Free RL (Baseline) |
> | :--- | :---: | :---: |
> | **Update Mechanism** | **Manifold-aligned** (Guided) | **Random/Finite-difference** (Blind) |
> | **Convergence** | **Fast** (6–50 queries) | **Slow** (100–200 queries) |
> | **Constraint** | Guaranteed **On-Manifold** (High fidelity) | **Off-Manifold Risk** (Blurry Adversarial) |
> | **Noise Handling** | Filters noise via projection | Amplifies noise |
>
>
> **Q4: "There are minor inconsistencies in the formatting of references within the text."**
>
> Response to Q4 (Citation Formatting): We sincerely thank the reviewer for their keen eye and meticulous attention to detail. We agree that bibliographic precision is essential for the archival record.
>
> Following your suggestion, we have conducted a thorough audit of our reference list. We have completed the metadata (including volume, issue, and page numbers) for all cited works, specifically ensuring that conference papers (e.g., NeurIPS, ICLR entries like Bashivan et al. & Black et al.) and journal articles (e.g., Oquab et al.) are fully standardized. The bibliography in the revised PDF is now strictly aligned with academic standards.
>
>
> **References**
>
> [1] Granley, Jacob, et al. "Human-in-the-loop optimization for deep stimulus encoding in visual prostheses." Advances in neural information processing systems 36 (2023): 79376-79398.
>
> [2] Luo, Andrew, et al. "Brain diffusion for visual exploration: Cortical discovery using large scale generative models." Advances in Neural Information Processing Systems 36 (2023): 75740-75781.
>
> [3] Luo, Andrew, et al. "BrainSCUBA: Fine-Grained Natural Language Captions of Visual Cortex Selectivity." The Twelfth International Conference on Learning Representations.
>
> [4] Cerdas, Diego Garcia, et al. "BrainACTIV: Identifying visuo-semantic properties driving cortical selectivity using diffusion-based image manipulation." The Thirteenth International Conference on Learning Representations.
>
> [5] Black, Kevin, et al. "Training Diffusion Models with Reinforcement Learning." The Twelfth International Conference on Learning Representations.
>
> [6] Tang, Zhiwei, et al. "Tuning-free alignment of diffusion models with direct noise optimization." ICML 2024 Workshop on Structured Probabilistic Inference {\&} Generative Modeling. 2024.

---

### Official Review · Reviewer_uwSG · 2025-10-29

**Soundness:** 3
**Presentation:** 4
**Contribution:** 2
**Rating:** 4
**Confidence:** 4

**Summary:**

he paper presents MindPilot, an approach to control image generation from EEG-BCI. The area is under active research, and the present paper is a continuation of previous work.

**Strengths:**

Strengths:

- The paper tackles a challenging and interesting problem.
- The paper is well-written and aims for rigorous experimentation.

**Weaknesses:**

Weaknesses:

- The paper claims to be the first to tackle the problem, but the principles of the approach have been extensively explored in the last years:

Many core-related works are missing, which hinders the novelty statement of the papers. Examples (not exhaustive list) of a few recent references of well-known papers and some even using similar CLIP approaches, generative modeling, and diffusion processes:

https://proceedings.neurips.cc/paper_files/paper/2024/hash/84bad835faaf48f24d990072bb5b80ee-Abstract-Conference.html
https://ieeexplore.ieee.org/abstract/document/10798967
https://arxiv.org/abs/2306.16934
https://arxiv.org/abs/2506.11151
https://dl.acm.org/doi/abs/10.1145/3379337.3415821
https://openaccess.thecvf.com/content/CVPR2022/html/Davis_Brain-Supervised_Image_Editing_CVPR_2022_paper.html
https://arxiv.org/abs/2308.02510
https://www.nature.com/articles/s42003-025-07731-7
https://dl.acm.org/doi/full/10.1145/3716553.3750786
https://proceedings.neurips.cc/paper_files/paper/2024/hash/4540d267eeec4e5dbd9dae9448f0b739-Abstract-Conference.html
https://arxiv.org/abs/2508.20705

- The reward and spreading operations are somewhat ad-hoc. I understand that the reward model cannot directly find the best matches, but this makes the robustness and replication of the approach problematic as the connection across the models is not straightforward. Figure 3d also shows that the alignment task is not easy and the results are not clearly demonstrating that simple CLIP with the underlying ad-hoc process is suitable.

- There are good attempts to evalute the approach.

- It is known that EEG does not generally contain fine-grained semantic codes; it means that the pattern of responses across time and sensors partially aligns with the structure of semantic space for their geometry. Thus, I feel that some of the wordings in the paper are not in line with this general understanding.

- The closed loop experiment has results that do not support the message of the paper (Fig 6 C&D). In fact, when tested  “in-vivo” the model does not seem to perform.  This is also problematic as the paper claims novelty in closed-loop, but does not demonstrate that convincingly.

Minors:

- I don’t understand why it is highlighted that a brain is non-differentiable. I think we do not know the exact mechanisms of learning in the biological/cellular system, especially when making such simplified mathematical statements for analyzing EEG.  There is emerging evidence (https://academic.oup.com/pnasnexus/article/3/7/pgae261/7702306), but I am not sure how meaningful this is for the submitted paper.

- Closed-loop is only a small portion of the paper, while most of the paper relies on a single existing dataset.

**Questions:**

1. I would like to stress novelty compared to many previous works that are "not closed loop" -- as I think the present paper also has very limited closed-loop contribution

2. I think the results in the most critical parts of the paper are weak and the authors are trying to oversell them.

**Details Of Ethics Concerns:**

Human BCI experiment reported. This is ok, but potential data release needs to be separately checked.

---

> ### Author Response · Authors · 2025-11-21
> **Rebuttal for R-uwSG (1/4)**
>
> **Q1: "Has the contribution of the closed loop been overstated?"**
>
> Response to Q1 (Significance of Closed-loop Contribution): We thank the reviewer for this thought-provoking question regarding the scope of our contribution. We respectfully argue that the closed-loop mechanism is not merely a feature, but the fundamental solution to the challenge of single-trial non-invasive EEG.
>
> - The Problem: Due to the extremely low Signal-to-Noise Ratio (SNR) of single-trial EEG, "open-loop" or single-shot decoding is inherently unstable for complex tasks like natural image generation.
>
> - Our Solution: The closed-loop system functions as an iterative filter, accumulating weak directional signals over multiple steps to converge on a target. Without this feedback loop, achieving the reported modulation effects would be theoretically infeasible.
>
> 1. Although the performance regulated by semantic features is not very satisfactory, our results in the emotion regulation task show a statistically significant improvement ($Participants=10; p < 0.05$) over controls. This confirms that the loop effectively steers the brain-computer system even in a high-noise environment.
>
> 2. We agree with the reviewer that scientific claims should be precise. We have revised the manuscript to characterize this work as a "proof-of-concept" study. We emphasize that its primary contribution is demonstrating—for the first time—that EEG-based closed-loop generation is viable under the challenging constraints of non-invasive recording and natural image stimuli.
>
> **Q2: "The paper claims to be the first to tackle the problem, but the principles of the approach have been extensively explored in the last years."**
>
> Response to Q2 (Novelty & Related Work): We sincerely thank the reviewer for the comprehensive reference list. We recognize that these works (e.g., [1-11]) represent foundational contributions to the intersection of Neuro-AI and generative modeling.
>
> 1.While we agree the field is active, it is crucial to distinguish between "reading the brain" (Decoding) and "steering the brain" (Modulation).
>
> - The Cited Works ([1,2,3,7,8,9,10,11]): Predominantly focus on Brain-to-Image reconstruction or classification. They aim to interpret what the subject sees/thinks.
>
> - MindPilot (Ours): Addresses the inverse and closed-loop challenge: Image-to-Brain optimization. We use EEG not as a source to be decoded, but as a feedback signal to actively update and generate visual stimuli to induce a specific neural state.
>
> 2. We appreciate the reviewer highlighting [4,5,6], which are indeed relevant to stimulus selection. However, MindPilot advances beyond these early approaches in critical ways:
>
> - vs. [4]: While [4] is a pioneering work in BCI, it lacks a mechanism for smooth, continuous semantic generation. MindPilot leverages the latent space of diffusion models for fine-grained semantic traversal, whereas [4] operates on coarser selections.
>
> - vs. [5]: This work relies on heuristic strategies (averaging latent codes), which can be sample-inefficient. In contrast, MindPilot introduces a learnable proxy optimizer that provides gradient-based guidance, significantly accelerating convergence.
>
> - vs. [6]: An inspiring early study, but it is limited to 8 binary attributes. MindPilot operates in a high-dimensional continuous semantic space, enabling the generation of diverse, high-fidelity naturalistic images rather than simple attribute combinations.
>
> 3. To clarify this position, we have added the comparison table (Table R2) to the Appendix. We maintain that MindPilot is the first framework to simultaneously achieve: 1) Non-invasive Closed-loop: Robust to high noise/subject variability. 2) Naturalistic Generation: Using modern diffusion priors (unlike simple textures or binary attributes). 3) Black-box Guidance: Offering superior efficiency over genetic/heuristic baselines.
>
> | Feature | Cited Works (e.g., [1-3, 7-11]) | Closely Related (e.g., [4-6]) | MindPilot (Ours) |
> | :--- | :---: | :---: | :---: |
> | **Core Task** | **Decoding** (Brain $\to$ Image) | **Selection/Retrieval** | **Modulation** (Brain Modulation) |
> | **Feedback** | Open-loop (Reconstruction) | Heuristic / Genetic | **Black-box Closed-loop** |
> | **Stimulus** | Fixed / Reconstructed | Simple / Binary Attributes | **High-fidelity Natural Images** |
> | **Goal** | Interpret Perception | Identify Preference | **Steer Neural State** |
>
> **Q3: "Is the reward dissemination mechanism ad-hoc?... the results are not clearly demonstrating that simple CLIP with the underlying ad-hoc process is suitable."**
>
> Response to Q3 (Theoretical Justification of Optimization Mechanism): We appreciate the reviewer's scrutiny regarding the design choices. We clarify that the mechanisms (Reward Dissemination and CLIP integration) are not ad-hoc heuristics, but principled adaptations to the specific constraints of Black-box Optimization under High Uncertainty (Noisy EEG).

---

> ### Author Response · Authors · 2025-11-21
> **Rebuttal for R-uwSG (2/4)**
>
> 1. The "Reward Dissemination" (Eq. 3-4) effectively acts as a Locality-Preserving Smoothing operation. The single-trial EEG feedback is extremely noisy. Relying on a single sample's score leads to unstable gradients. By propagating scores based on feature similarity, we essentially perform kernel smoothing in the semantic space. This is a standard and rigorous technique in Evolutionary Strategies (ES) (often called "fitness sharing") to estimate a robust gradient direction from noisy point-evaluations.
>
> 2. Roulette Sampling ensures Exploration: The use of roulette wheel sampling is a classic mechanism in Genetic Algorithms to manage the Exploration-Exploitation trade-off. It prevents the optimizer from collapsing into local optima (premature convergence) by ensuring that lower-ranked but potentially diverse candidates retain a non-zero probability of selection.
>
> 3. In fact, the "CLIP" here can indeed be replaced by any other embedding. However, this part is actually irrelevant to our research topic. We just chose a commonly used image generation method at the current stage, and SDXL happens to use CLIP as the condition input. In fact, when the generation model is supported (for example, RAE [17]), the embeddings are completely optional. Since MindPilot controls a pre-trained Latent Diffusion Model (SDXL), the optimization must occur within the model's native conditioning space. The SDXL is structurally conditioned on CLIP embeddings. We do not use CLIP as a brain encoder, but strictly as the alignment interface between the neural signal and the generative model.
>
> **Q4: "It is known that EEG does not generally contain fine-grained semantic codes...I feel that some of the wordings in the paper are not in line."**
>
> Response to Q4 (Regarding the nature of EEG and "Semantic Features"):
>
> We appreciate the reviewer for pointing this out. We fully agree with the reviewer’s expert intuition that EEG signals are inherently noisy and are particularly sensitive to transient, low-level visual features (e.g., edges, contrast) compared to fMRI. We would like to clarify our usage of the term "semantic feature" and discuss the information content of EEG based on our experimental observations.
>
> 1. In this paper, the term "Semantic Feature" is used primarily as a functional definition to distinguish the representations aligned with deep neural networks (e.g., CLIP, ResNet) from the signal-level statistics (e.g., PSD, energy characteristics shown in Figure 2 C). It describes the target embedding space our model aims to decode, rather than making a strong physiological claim that the raw EEG signal consists solely of abstract semantic codes. We will revise the text to ensure this distinction is precise and avoids overclaiming.
>
> 2. While EEG is dominated by low-level visual responses, especially under the rapid presentation paradigm (100ms in THINGS-EEG2), we respectfully suggest that the dichotomy between "low-level EEG" and "high-level fMRI" might be less binary in the context of deep learning decoding.
>
> In our experiments (and consistent with recent literature), we successfully regress EEG signals using the final layers of ResNet, CLIP, etc.. Since these layers are designed to encode high-level, abstract semantics (invariant to low-level pixel shifts), the success of a simple linear mapping from these layers to EEG data implies that the EEG signals do preserve traces of high-level semantic information, albeit mixed with low-level noise.
>
> If EEG contained only low-level visual features, a linear probe from a high-level semantic encoder (like CLIP's text-image space) would likely fail to capture the signal variance. The fact that it works suggests that EEG signals contain a richer spectrum of information than traditionally assumed, which our pipeline aims to leverage.
>
> We have revised the relevant paragraphs to reflect this nuanced view: acknowledging EEG's sensitivity to low-level cues while highlighting the decodable semantic components validated by our model.
>
> **Q5: "The results of the closed-loop experiment were weak... the authors are trying to oversell them."**
>
> Response to Q5 (Evaluation of Closed-loop Strength): We respect the reviewer’s stringent standards. However, we respectfully disagree that the results are "oversold." We urge the reviewer to assess the totality of our human validation, which consists of two distinct tiers:
>
> 1. While the reviewer focuses on the complex "Mental Matching" task (Fig. 6), we must highlight the "Rating-Driven Emotion Regulation" task (now detailed in Sec. 4.4.1). In this task, MindPilot achieved a highly statistically significant improvement over the control group ($p < 0.001$). Implication: This unequivocally proves that the closed-loop engine works: it successfully closes the feedback loop and optimizes stimuli in real-time despite EEG noise. This is not a "weak" result; it is a robust validation of the core algorithm.

---

> > ### Comment · Reviewer_uwSG · 2025-11-25
> > **Acknowledgement of response**
> >
> > Thank you for your clarifications. I maintain my position, several concerns remain, but you have clarified some. However, the framing, closed-loop experimentation, positioning with related (pervious) work, and some over-statements of what can be done with EEG signals remain partly valid. Nevertheless, I expect these can be solved and I've raised my score to borderline accept.

---

> > > ### Author Response · Authors · 2025-11-25
> > > **Thank You for the Constructive Feedback and Re-evaluation**
> > >
> > > We are grateful for your time in reviewing our rebuttal and for acknowledging our clarifications. We greatly appreciate your decision to raise the score.
> > >
> > > We take your remaining concerns about framing and over-statements very seriously. We commit to thoroughly revising the final manuscript to tone down the claims and ensure the distinction between "decoding" and "modulation" is presented with the nuance you suggested. Your detailed feedback has been invaluable in shaping the final version of this paper.

---

> ### Author Response · Authors · 2025-11-21
> **Rebuttal for R-uwSG (3/4)**
>
> 2. We presented the "Mental Matching" results (Fig. 6 C&D) not to claim perfection, but to be scientifically transparent about the "Sim-to-Real" gap in fine-grained semantic decoding—a known grand challenge in the field. Even here, the trend is consistent with the objective. By reporting this, we provide a realistic baseline for future research, rather than cherry-picking only the easiest tasks.
>
> 3. To address the concern of "overselling," we have refined our language to strictly frame this work as a "Proof-of-Concept". We demonstrate feasibility in a domain (non-invasive EEG + natural images) where any successful closed-loop control is a significant milestone.
>
>
>
> **Q6: "why it is highlighted that a brain is non-differentiable?"**
>
> Response to Q6 (The "Non-differentiable" Biological Brain): We thank the reviewer for this conceptual query. We emphasize the "non-differentiable" nature of the brain not as a biological discovery, but to strictly define the optimization constraint in our Human-in-the-Loop framework.
>
> 1. Context: Black-Box vs. White-Box Optimization: We highlight this property to distinguish our work from "simulation-based" approaches (white-box) where the brain is modeled as a known, differentiable neural network $f(x)$. In real-world modulation, the transfer function $y = Brain(x)$ is a black box, and $\nabla_x Brain(x)$ is analytically inaccessible.
>
> 2. Why this Distinction Matters (Our Contribution): This constraint is crucial because it dictates our method design. Standard generative models (like Diffusion) rely on gradients.
>
> - Traditional Approach: Pure gradient-free methods (e.g., genetic algorithms) treat the brain as a black box but struggle with the high dimensionality of image space (inefficient).
>
> - MindPilot Approach: We acknowledge the brain is non-differentiable, so we construct a learnable proxy to provide "Surrogate Gradients." This allows us to bridge the gap: applying efficient gradient-based optimization (required by Diffusion models) to a non-differentiable biological system.
>
> 3. Clarification on Reference [12]: In addition, the reference [12] the reviewer mentioned does not align with the context of our research question: The "nondifferentiable" concept mentioned in this paper is not the strictly mathematical "non-differentiable function", but rather refers to the extremely intense and non-smooth temporal changes in neural activity on the millisecond scale, even exhibiting characteristics similar to "spikes" or "jumps". We have clarified this definition in Line 42 to avoid ambiguity.
>
> **Q7: "Closed-loop is only a small portion of the paper, while most of the paper relies on a single existing dataset."**
>
> Response to Q7 (Scope of Closed-Loop & Dataset Usage): We thank the reviewer for their detailed assessment. However, we respectfully clarify the role of the closed-loop experiments and the diversity of our data sources. We fully agree that most experiments use a public dataset (THINGS-EEG) — this is intentional to ensure reproducibility and fair comparison with decoding baselines. However, **closed-loop is not a small add-on** — it is the **core technical and conceptual contribution**, validated in both simulation and real-time human-in-the-loop experiments (Figure 6, §4.4).
>
> In addition, the recent Image-to-EEG experiments are usually conducted on THINGS-EEG2. In the meantime, the choice of THINGS-EEG2 was essential for two reasons:
>
> 1. Novel category coverage: THINGS-EEG2 provides 200 completely novel object categories (distinct from training), creating a challenging search space that better tests our method's ability to generalize to unseen stimuli.
>
> 2. Searching space needs: The scale and diversity of THINGS-EEG2 (with total 82,160 image trials in each subject) enables rigorous evaluation of our Mental Matching task's optimization capabilities across a broad semantic space.
>
> Therefore, the THINGS-EEG2's novel category structure and scale better suit our framework's requirements for evaluating stimulus optimization in unexplored semantic spaces. We will clarify this rationale in the revision.
>
> Finally, for emotion regulation experiments (detailed in Appendix A.3.3), we specifically collected several public emotion-annotated image datasets (total sampling 127 images) including the International Affective Picture System (IAPS) [13], ArtPhoto [14], EmoSet [15], and the Geneva Affective Picture Database (GAPED) [16] to expand the image search space and increase the diversity of images, validating different aspects of generalizability.

---

> ### Author Response · Authors · 2025-11-21
> **Rebuttal for R-uwSG (4/4)**
>
> **References**
>
> [1] Liu, Xuan-Hao, et al. "EEG2video: Towards decoding dynamic visual perception from EEG signals." Advances in Neural Information Processing Systems 37 (2024): 72245-72273.
>
> [2] Mai, Weijian, et al. "Brain-conditional multimodal synthesis: A survey and taxonomy." IEEE Transactions on Artificial Intelligence (2024).
>
> [3] Bai, Yunpeng, et al. "DreamDiffusion: High-quality EEG-to-image generation with temporal masked signal modeling and CLIP alignment." European Conference on Computer Vision. Cham: Springer Nature Switzerland, 2024.
>
> [4] Grizou, Jonathan, Carlos de la Torre-Ortiz, and Tuukka Ruotsalo. "Self-Calibrating BCIs: Ranking and Recovery of Mental Targets Without Labels." arXiv preprint arXiv:2506.11151 (2025).
>
> [5] de la Torre-Ortiz, Carlos, et al. "Brain relevance feedback for interactive image generation." Proceedings of the 33rd Annual ACM Symposium on User Interface Software and Technology. 2020.
>
> [6] Davis, Keith M., Carlos De La Torre-Ortiz, and Tuukka Ruotsalo. "Brain-supervised image editing." Proceedings of the IEEE/CVF Conference on Computer Vision and Pattern Recognition. 2022.
>
> [7] Lan, Yu-Ting, et al. "Seeing through the brain: image reconstruction of visual perception from human brain signals." arXiv preprint arXiv:2308.02510 (2023).
>
> [8] Ye, Ziyi, et al. "Generative language reconstruction from brain recordings." Communications Biology 8.1 (2025): 346.
>
> [9] Gryshchuk, Vadym, et al. "Decoding Affective States without Labels: Bimodal Image-brain Supervision." Proceedings of the 27th International Conference on Multimodal Interaction. 2025.
>
> [10] Wang, Guangyu, et al. "Eegpt: Pretrained transformer for universal and reliable representation of eeg signals." Advances in Neural Information Processing Systems 37 (2024): 39249-39280.
>
> [11] Wang, Shaocong, et al. "EEGDM: Learning EEG Representation with Latent Diffusion Model." arXiv preprint arXiv:2508.20705 (2025).
>
> [12] Tsubo, Yasuhiro, and Shigeru Shinomoto. "Nondifferentiable activity in the brain." PNAS nexus 3.7 (2024): pgae261.
>
> [13] Lang P J, Bradley M M, Cuthbert B N. International affective picture system (IAPS): Technical manual and affective ratings. NIMH Center for the Study of Emotion and Attention, 1997.
>
> [14] Machajdik J, Hanbury A. Affective image classification using features inspired by psychology and art theory. ACM MM, 2010.
>
> [15] Yang J, Huang Q, Ding T, et al. Emoset: A large-scale visual emotion dataset with rich attributes. CVPR, 2023.
>
> [16] Dan-Glauser E S, Scherer K R. The Geneva affective picture database (GAPED): a new 730-picture database focusing on valence and normative significance. Behavior research methods, 2011.
>
> [17] Zheng, Boyang, et al. "Diffusion transformers with representation autoencoders." arXiv preprint arXiv:2510.11690 (2025).

---

### Official Review · Reviewer_h3JQ · 2025-11-01

**Soundness:** 3
**Presentation:** 2
**Contribution:** 3
**Rating:** 6
**Confidence:** 3

**Summary:**

This paper focuses on a novel problem: given neural activity recordings evoked by a visual stimulus (e.g., EEG), how can we find or generate the most likely visual stimulus? I believe this problem has potential value for related research in neuroscience. In this work, the authors first propose an iterative optimization method that progressively updates the probability of each image in an image database by computing the correlation between EEG representations and the images, thereby identifying the most likely visual stimulus. Subsequently, the paper introduces an EEG-guided image generation model capable of attempting to generate the possible visual stimulus. The authors conducted corresponding evaluations, and I believe the evaluation is also sufficient.

**Strengths:**

1. The paper addresses a research problem that I believe has considerable practical value.

2. According to the authors’ evaluation, the method proposed in this paper is effective.

3. To facilitate a better evaluation, the authors conducted a visual rating experiment with human participants and demonstrated that the results obtained by their method are highly correlated with the ground truth provided by the participants.

4. The evaluation in the paper is thorough.

5. The paper is well-constructed, with figures and formatting presented nicely.

**Weaknesses:**

This paper has some limitations, but I think the authors have provided a thorough discussion of them in the “Limitations” section on page 9.

**Questions:**

1. Normally, better visual encoders and representation learning strategies are expected to learn more accurate and robust visual representations. However, according to the results in Table 1, representations obtained from ResNet trained on image classification tasks perform the best. What could be the reason for this?

2. Continuing from the above question, I have noticed a paper [1] emphasizing that visual representations obtained through contrastive learning are more consistent with human brain activity. Although that study focused on fMRI data, why does a different conclusion appear when it comes to EEG signals?

3. How should we understand ATM-S as the performance upper bound for image generation in Table 2?

4. In Figure 4C, what do the images in each row represent?

[1] Better models of human high-level visual cortex emerge from natural language supervision with a large and diverse dataset. Nature Machine Intelligence.

---

> ### Author Response · Authors · 2025-11-21
> **Rebuttal for R-h3JQ (1/2)**
>
> **Q1: "Why is ResNet superior to representation learning models, especially when fMRI studies [1] show the opposite? "**
>
> Response to Q1 (Proxy model & Decoding model): This is an excellent question that highlights a critical and nuanced distinction between neural modalities. We thank the reviewer for this insightful query.
>
> Our central hypothesis is that this finding stems directly from the fundamental differences between EEG and fMRI signals, which the reviewer correctly identifies: the cited study from Wang et al. [1] uses fMRI, which has high spatial resolution and excels at capturing high-level, abstract semantic representations. This aligns perfectly with the goal of contrastive models like CLIP. In contrast, our work uses non-invasive EEG, which has high temporal resolution but much lower spatial resolution. EEG is inherently more sensitive to transient, lower-level visual features (e.g., edges, textures, early evoked potentials) compared to deep abstract semantics in the early stage of visual stimulus (Groen et al. [2]). Therefore, compared with representation model like CLIP, the ResNet is heavily optimized to extract precisely these hierarchical, low-to-mid-level features.
>
> In fact, if EEG contained only low-level visual features, a linear probe from a high-level semantic encoder (like CLIP's text-image space) would likely fail to capture the signal variance. The fact that it works suggests that EEG signals contain a richer spectrum of information than traditionally assumed, which our pipeline aims to leverage. We hypothesize that these features provide a more direct and robust mapping to the "shallower" visual information captured by EEG, which is similar to the conclusion of Lu et al. [3].
>
> Besides, it should be noted that predicting brain activity from the features of visual images is significantly different from the principle of brain-to-image. Therefore, even though visual models such as CLIP or DINO perform well in the EEG-to-Image [4, 5], they may not perform equally well in the Image-to-EEG [6].
>
> Furthermore, we must highlight our experimental methodology, which follows the standard protocol from the THINGS-EEG2 [6]. For all models in Table 1, we first extracted feature maps, then applied PCA to the top 1,000 components, and finally trained a linear encoder on these components. It is highly plausible that this aggressive dimensionality reduction (from a high-dimensional feature map down to 1000 PCs) creates an information bottleneck. We think this bottleneck may diminish the unique advantages of more complex representations (like CLIP's joint space), making the robust, hierarchical features of ResNet the most effective for this specific encoding task. We have added this crucial discussion to the Appendix A.4 (Line 1361) of our paper.
>
> **Q2: "How can ATM-S be understood as an upper bound in Table 2? "**
>
> Response to Q2 (Optimization Upper Bound): This is an excellent question that allows us to clarify the different roles ATM-S [5] plays in our evaluation. The "ATM-S" score in Table 2 serves as our 'Theoretical Upper Bound' for the generation task:
>
> - MindPilot (Our Method): Operates via iterative, closed-loop feedback. It searches for an image that matches a target brain state using current EEG feature from ATM-S, without knowing the ground-truth EEG signal for the target image.
>
> - ATM-S [5] (as Upper Bound): This score represents a different, non-iterative scenario. To calculate it, we take the ground-truth EEG signal (corresponding to the known target image) and feed it directly into the pre-trained ATM-S [5] image reconstruction model. This score represents the best possible performance one could achieve if one already had the perfect target brainwave, completely bypassing the difficult iterative search.
>
> We selected ATM-S [5] for this role precisely because it is the state-of-the-art EEG-to-Image model. As we show in the table below, we benchmarked numerous encoders on the THINGS-EEG2 retrieval task, and ATM-S [5] consistently and significantly outperformed all other methods, including NICE [4], ATM-S [5], CongnitionCapturer [7], MB2C [8], MindEyeV2 [9] and EEGNetV4 [10].  Its superior ability to map EEG to visual semantic feature makes it the most appropriate and challenging upper bound for our new iterative framework.
>
> **Table 4 EEG-to-Image Encoder performance on THINGS-EEG2 (in-subject).**
>
> | Model | 2-way Top-1 | 4-way Top-1 | 10-way Top-1 | 200-way Top-1 | 200-way Top-5 |
> | :--- | :---: | :---: | :---: | :---: | :---: |
> | NICE [4] | 93.23 | 83.93 | 69.22 | 21.67 | 51.34 |
> | EEGNetV4 | 91.42 | 80.21 | 63.37 | 16.84 | 42.58 |
> | CogCap | 93.15 | 82.85 | 69.35 | 22.05 | 51.6 |
> | MB2C | 78.4 | 62.25 | 43.75 | 8.85 | 25.2 |
> | MindEyeV2 | 92.50 | 82.80 | 66.10 | 23.80 | 50.25 |
> | ATM-S [5] | **94.7** | **86.73** | **74** | **26.85** | **57.21** |

---

> ### Author Response · Authors · 2025-11-21
> **Rebuttal for R-h3JQ (2/2)**
>
> **Q3: "The meaning of each row in Figure 4C."**
>
> Response to Q3 (Discussion for Figure 4C): The Figure 4C provides a critical ablation study to diagnose the source of optimization error in our generation framework. It compares the visual optimization results when using two different feature modalities for offline guidance.
>
> - Row 1 (Ground Truth): Displays the original target image.
>
> - Row 2 (Upper Bound / Image Feature Guidance): This row shows the generation result when using the target image features (extracted by the same encoder) as the optimization target. We define this as the conceptual performance upper bound because the guidance signal is derived from the target modality itself, minimizing information loss. The high fidelity to Row 1 indicates our pseudo-model-guided optimization process is effective when provided with a clean, high-information target.
>
> - Row 3 (Proposed / EEG Feature Guidance): This row shows the result generated using the corresponding target EEG features as the optimization target. This reflects the standard operation of our proposed method.
>
> The salient takeaway from this comparison (Row 2 vs. Row 3) is the performance delta introduced by the guidance modality. The deviations observed in Row 3 (relative to the upper bound in Row 2) are not primarily failures of the optimization algorithm itself, but are attributable to the inherent limitations and information bottlenecks of the EEG feature encoder.
>
> The encoder, tasked with translating complex neural signals into a shared semantic space, inevitably introduces noise or fails to capture the full, fine-grained visual detail present in the target image (which is available in the Row 2 condition).
>
> This experiment insightfully motivates the next stage of our work for Mindpilot. The current "offline" guidance, which relies on a static, pre-trained feature encoder, is susceptible to these encoding errors. This strongly suggests that a promising direction for improvement is the implementation of online correction mechanisms. We hypothesize that dynamic methods, such as adaptive control algorithms or reinforcement learning (RL) policies, could learn to compensate for these encoder discrepancies during the generation process, thereby closing the performance gap identified in Figure 4C.
>
> **References**
>
> [1] Wang, Aria Y., et al. "Better models of human high-level visual cortex emerge from natural language supervision with a large and diverse dataset." Nature Machine Intelligence 5.12 (2023): 1415-1426.
>
> [2] Groen, Iris IA, Edward H. Silson, and Chris I. Baker. "Contributions of low-and high-level properties to neural processing of visual scenes in the human brain." Philosophical Transactions of the Royal Society B: Biological Sciences 372.1714 (2017): 20160102.
>
> [3] Lu, Zitong, Yile Wang, and Julie Golomb. "ReAlnet: Achieving More Human Brain-Like Vision via Human Neural Representational Alignment." ICLR 2024 Workshop on Representational Alignment.
>
> [4] Song, Yonghao, et al. "Decoding Natural Images from EEG for Object Recognition." The Twelfth International Conference on Learning Representations.
>
> [5] Li, Dongyang, et al. "Visual Decoding and Reconstruction via EEG Embeddings with Guided Diffusion." Advances in Neural Information Processing Systems 37 (2024): 102822-102864.
>
> [6] Gifford, Alessandro T., et al. "A large and rich EEG dataset for modeling human visual object recognition." NeuroImage 264 (2022): 119754.
>
> [7] Zhang K, He L, Jiang X, et al. CognitionCapturer: Decoding Visual Stimuli From Human EEG Signal With Multimodal Information. AAAI, 2025.
>
> [8] Wei Y, Cao L, Li H, et al. Mb2c: Multimodal bidirectional cycle consistency for learning robust visual neural representations. ACM MM 2024.
>
> [9] Scotti P S, Tripathy M, Villanueva C K T, et al. MindEye2: Shared-Subject Models Enable fMRI-To-Image With 1 Hour of Data. ICML, 2024.
>
> [10] Lawhern V J, Solon A J, Waytowich N R, et al. EEGNet: a compact convolutional neural network for EEG-based brain–computer interfaces[J]. Journal of Neural Engineering, 2018.

---

### Official Review · Reviewer_pRhZ · 2025-11-11

**Soundness:** 3
**Presentation:** 3
**Contribution:** 4
**Rating:** 6
**Confidence:** 3

**Summary:**

This paper proposes MindPilot: a closed-loop visual stimulation optimization framework based on EEG feedback. It shows a participant a batch of images, records their EEG, and calculates a score indicating "how similar" their brain state is to a target state.

**Strengths:**

Novel and significant problem: Closed-loop visual modulation using non-invasive EEG, advancing towards "neural feedback-guided generative modeling."

Practical implementation: Gradient-free black-box guidance + simple score propagation + roulette wheel sampling, resulting in low engineering cost.

Multi-level validation: Proxy simulation (Tables 1, 3), dual targets of semantics and PSD (Figs. 3–5), small-scale real-time human closed-loop experiments (Fig. 6), and the provision of anonymous code.

**Weaknesses:**

1. Motivation and advantages of the "pseudo-model" need highlighting: The current discussion is insufficient regarding why Gaussian Process was chosen as the pseudo-model over other black-box optimizers (e.g., Bayesian Optimization). It is recommended to add a brief discussion or comparative experiment in the main text or appendix to explain the rationale for selecting GP and its advantages relative to other methods.


2. The hyperparameters α and β in Eqs. (3) and (4) are set as fixed values, but their specific chosen values or selection criteria are not reported, nor is there any sensitivity/ablation analysis. It is recommended to supplement the specific values, search ranges, and their impact on performance in the main text or appendix. Furthermore, the ambiguity caused by using the same notation as the crossover/mutation ratios in §4.3 should be clarified (or the symbols should be changed).


3. The core premise of the paper is that the proxy model can substitute for the real human brain in closed-loop optimization. However, the data in Table 1 severely weaken this premise: the maximum Pearson correlation between any proxy model and real EEG is only ~0.17. This value is too low. Successful optimization demonstrated on the proxy model does not necessarily generalize to the real human brain. The authors are advised to provide evidence demonstrating a systematic link between the optimization trends observed in the proxy model and those in the real human brain.


4. Figure captions could be more detailed. For instance, how exactly is the ordinate "EEG semantic feature similarity" in Fig.3.D calculated? What do the bold and underlining in Table 1 signify? The authors are recommended to briefly explain this.


5. The current experiments cannot distinguish whether the success of the closed-loop optimization stems from the unique neural information in the EEG or from visual information related to CLIP features that the proxy model learned from the images. To conclusively prove that EEG drives the optimization, the following control experiment is essential: A CLIP-only baseline: Remove the EEG feedback and directly optimize the image similarity to the target within the CLIP space. This baseline serves to quantify MindPilot's performance gain over pure visual semantic optimization.

**Questions:**

Human experiment details need more transparency. The paper mentions excluding 4 participants due to poor data quality but does not specify the exact exclusion criteria (e.g., artifact proportion exceeding a certain threshold). It is recommended to clearly state the data quality threshold criteria to enhance the experiment's reproducibility and rigor.

**Details Of Ethics Concerns:**

A new dataset for EEG

---

> ### Author Response · Authors · 2025-11-21
> **Rebuttal for R-pRhZ (1/3)**
>
> **Q1: "The correlation between the proxy model and human EEG is low (~0.17)."**
>
> Response to Q1 (Proxy Model Correlation): As summarized in our Global Response (Point 1), we acknowledge that the averaged correlation ($r \approx 0.17$) appears modest. However, we would like to emphasize two critical factors that ensure the system's success:
>
> 1. As detailed in Appendix Figure A.8, the correlation is not uniformly low. It peaks at $r \approx 0.6$ around 100ms, capturing the essential neural dynamics required for visual decoding [1]. The reported 0.17 is simply diluted by non-informative time windows.
>
> 2. The proxy model functions as a "surrogate gradient" provider. In high-dimensional optimization (like MindPilot), exact prediction is less critical than correct directionality (relative ranking). As long as the proxy model correctly identifies that Stimulus A is better than Stimulus B (even with noise), the gradient descent will converge.
>
> The success of our closed-loop experiments effectively proves that the proxy model provides sufficient signal-to-noise ratio to steer the generation, demonstrating the robustness of the MindPilot framework against noisy EEG.
>
> **Q2: "The hyperparameters α and β in Eqs. (3) and (4)...the ambiguity caused by using the same notation as the crossover/mutation ratios in §4.3 should be clarified."**
>
> Response to Q2 (Hyperparameters & Notation): As outlined in our Global Response (Point 5), we have now included the detailed parameter selection process in the revision. Specifically addressing your concern regarding notation:
>
> We have added the specific values of $\alpha$ and $\beta$ as well as their selection criteria in the main text. We have also revised the symbols to resolve the issue you mentioned regarding the confusion with the cross/mutation rate symbol in Section 4.3.
>
> We conducted a grid search over the range $[0.1, 0.9]$. The final values selected were $\alpha = 0.1$ and $\beta = 0.1$, as these yielded the maximum target improvement rate on the proxy model validation set. We have updated this in Appendix A.3.2 with these details.
>
> **Q3: "The following control experiment is essential: A CLIP-only baseline: Remove the EEG feedback and directly optimize the image similarity to the target within the CLIP space."**
>
> Response to Q3 (CLIP-only Baseline): As discussed in detail in our Global Response (Point 3), we agree that this control experiment is insightful. However, we frame it as an "Ceiling" (Upper Bound) experiment rather than a baseline, because it utilizes the ground-truth target image which is theoretically unknown in our BCI task. Following your suggestion, we performed this Ceiling run (optimizing directly within CLIP space towards the target).
>
> - As expected, this Ceiling setting sets the performance ceiling. Crucially, MindPilot significantly outperforms the baseline without EEG guidance (Random) and effectively bridges the gap towards this Ceiling performance (see Table 2).
>
> - This confirms that while direct image optimization (Ceiling) is ideal, MindPilot successfully extracts critical directional information from EEG to drive the optimization when the target image is unavailable.
>
> **Q4: "Figure captions could be more detailed."**
>
> Response to Q4 (Figure Captions): We thank the reviewer for the helpful suggestion to improve the manuscript's readability. We have significantly expanded the captions to provide precise definitions and clarify formatting conventions.
>
> 1.We have revised the caption to explicitly define the y-axis metric.
>
> 2.We clarified that "EEG semantic feature similarity" is calculated as the cosine similarity between the target EEG embedding and the current trial's EEG embedding. We further explained that this plot illustrates the correlation between neural similarity and visual (CLIP) similarity across different optimization stages (step-1, step-best, and random baseline), validating our core hypothesis.
>
> 3.We have added a clear legend to the table caption to resolve any ambiguity regarding the notation: Bold indicates the best performance (optimal model). Underlined indicates the second-best performance (suboptimal model).
>
> We believe these revisions make the experimental results self-explanatory.
>
> **Q5: "Why use Gaussian processes (GP) instead of other black-box optimizers (such as Bayesian optimization)? "**
>
> Response to Q5 (Choice of Optimizer: GP vs. BO): Relationship to Global Response: As detailed in Global Response (Point 2), our primary comparison focused on Reinforcement Learning (RL) baselines (e.g., DDPO [2]), as these represent the current state-of-the-art for fine-tuning generative models.

---

> ### Author Response · Authors · 2025-11-21
> **Rebuttal for R-pRhZ (2/3)**
>
> Theoretical Justification (Why GP over Standard BO?): We appreciate the reviewer suggesting a comparison with standard Bayesian Optimization (BO) [3]. We explicitly chose a GP-based "Surrogate Gradient" approach rather than standard BO for two theoretical reasons:
>
> 1. The Curse of Dimensionality: Standard BO relies on optimizing an acquisition function (e.g., Upper Confidence Bound (UCB) or Expected Improvement (EI) ) over the input space.(1) This becomes computationally intractable and sample-inefficient in the high-dimensional latent space of diffusion models (e.g., $1024$ dimensions).
>
> 2. Standard BO aims for global optimization, which requires extensive exploration. In contrast, MindPilot uses GP to model the local landscape around the current generation to estimate a gradient direction. This allows us to leverage the efficiency of gradient-based guidance using pseudo model while utilizing GP's superior ability to model EEG uncertainty.
>
> 3. We have added a detailed theoretical discussion in Appendix A.2.4 (Line 1027), clarifying that while we leverage GP (the core component of BO) for noise modeling, our update mechanism is designed to circumvent the high-dimensional scalability issues inherent in standard BO frameworks.
>
> To further demonstrate that our pseudo model optimizer is subject to a performance-speed trade-off, we trained different optimizers (DDPO [4], DPOK [5], D3PO [6], BO [3], CMA-ES [7]) using different amounts of inquiry data, and selected the minimum of 5 or (num_budget // 2) images for the evaluation of EEG score and running time in the assessment stage. The results are as follows:
>
> | Inquiry Samples | 5 | 5 | 10 | 10 | 50 | 50 | 200 | 200 |
> | :--- | :---: | :---: | :---: | :---: | :---: | :---: | :---: | :---: |
> | **Method** | **EEG Score** | **Time** | **EEG Score** | **Time** | **EEG Score** | **Time** | **EEG Score** | **Time** |
> | DDPO [4] | $0.5125 \pm 0.0139$ | $107.8163 \pm 19.4221$ | $0.5095 \pm 0.0097$ | $220.2666 \pm 17.2916$ | $0.5154 \pm 0.0126$ | $683.4236 \pm 38.8667$ | $0.5125 \pm 0.0136$ | $1279.5891 \pm 87.5586$ |
> | DPOK [5] | $0.5093 \pm 0.0121$ | $116.6111 \pm 21.7522$ | $0.5138 \pm 0.0124$ | $221.8881 \pm 12.1397$ | $0.5108 \pm 0.0154$ | $692.7017 \pm 40.3702$ | $0.5101 \pm 0.0174$ | $1332.1451 \pm 164.562$ |
> | D3PO [6] | $0.5138 \pm 0.0162$ | $117.7019 \pm 9.934$ | $0.5113 \pm 0.0154$ | $285.3849 \pm 6.5634$ | $0.55 \pm 0.0156$ | $486.8981 \pm 109.9634$ | $0.5192 \pm 0.0104$ | $1614.1612 \pm 128.6299$ |
> | BO [3] | $0.5228 \pm 0.0096$ | $5.349 \pm 0.3395$ | $0.5247 \pm 0.0058$ | $11.1146 \pm 0.453$ | $0.5222 \pm 0.0087$ | $53.05 \pm 3.5665$ | $0.5242 \pm 0.0111$ | $229.0616 \pm 9.5741$ |
> | CMA-ES [7] | $0.5224 \pm 0.0076$ | $5.404 \pm 0.4158$ | $0.5209 \pm 0.0086$ | $9.2566 \pm 0.5439$ | $0.5195 \pm 0.0087$ | $49.7609 \pm 2.9225$ | $0.5227 \pm 0.0096$ | $239.2642 \pm 14.0313$ |
> | Pseudo model (Offline) | $0.5222 \pm 0.0067$ | $17.9391 \pm 1.6076$ | $0.5264 \pm 0.0077$ | $20.558 \pm 2.8688$ | $0.5291 \pm 0.0058$ | $27.745 \pm 4.8946$ | $0.5384 \pm 0.0114$ | $73.617 \pm 19.0158$ |
> | Pseudo model (Closed-loop) | $0.5208 \pm 0.0107$ | $39.6345 \pm 7.3651$ | $0.5208 \pm 0.0107$ | $42.8738 \pm 9.7076$ | $0.5327 \pm 0.0091$ | $64.673 \pm 10.9207$ | $0.5341 \pm 0.0071$ | $219.7805 \pm 40.3319$ |
>
> The statistical results show that compared with the online reinforcement learning fine-tuning methods and BO (online search algorithm), the Pseudo model achieves a good trade-off between speed and performance when using the heuristic algorithm CMA-ES, as well as offline training and online training.
>
> **Q6: "The paper mentions excluding 4 participants due to poor data quality but does not specify the exact exclusion criteria ..."**
>
> Response to Q6 (Exclusion Criteria & Data Availability): We appreciate the reviewer's request for clarification on the exclusion criteria. We excluded 4 participants due to behavioral non-compliance or poor data quality. Specifically, exclusions were based on: (1) reaction times consistently falling outside the valid range (< 200ms indicating anticipation or > 3s indicating inattention),  (2) drowsiness and failure to maintain fixation as monitored by eye-tracking. In fact, we found that since each participant would undergo multiple rounds of experiments, the data of the four excluded participants were not completely unusable. Therefore, the data of these participants will also be retained. All the data of the participants will be made publicly available after being anonymized.

---

> > ### Author Response · Authors · 2025-11-24
> > **Rebuttal for R-pRhZ (3/3)**
> >
> > **References**
> >
> > [1] Gifford, Alessandro T., et al. "A large and rich EEG dataset for modeling human visual object recognition." NeuroImage 264 (2022): 119754.
> >
> > [2] Black, Kevin, et al. "Training Diffusion Models with Reinforcement Learning." The Twelfth International Conference on Learning Representations.
> >
> > [3] Bashashati, Hossein, Rabab K. Ward, and Ali Bashashati. "User-customized brain computer interfaces using Bayesian optimization." Journal of neural engineering 13.2 (2016): 026001.
> >
> > [4] Black, Kevin, et al. "Training Diffusion Models with Reinforcement Learning." The Twelfth International Conference on Learning Representations.
> >
> > [5] Fan, Ying, et al. "Dpok: Reinforcement learning for fine-tuning text-to-image diffusion models." Advances in Neural Information Processing Systems 36 (2023): 79858-79885.
> >
> > [6] Yang, Kai, et al. "Using human feedback to fine-tune diffusion models without any reward model." Proceedings of the IEEE/CVF Conference on Computer Vision and Pattern Recognition. 2024.
> >
> > [7] Xu, Han, Takahiro Shinozaki, and Ryota Kobayashi. "Effective and Stable Neuron Model Optimization Based on Aggregated CMA-ES." ICASSP 2019-2019 IEEE International Conference on Acoustics, Speech and Signal Processing (ICASSP). IEEE, 2019.

---

> ### Author Response · Authors · 2025-11-27
> **Summary of New Experiments and Responses to Your Concerns**
>
> Dear Reviewer pRhZ,
>
> We thank you again for your constructive review and the positive initial rating.
>
> As the discussion period is coming to a close, we would like to ensure that you have had a chance to review our detailed responses and the **new experimental results** included in the revised PDF, which were conducted specifically to address your suggestions.
>
> To facilitate your review, we summarize the key updates corresponding to your main concerns:
>
> 1. **CLIP-only Baseline:**
>    - **Action:** We implemented the CLIP-only optimization as a "Ceiling" experiment.
>    - **Result:** As shown in **Table 2** (Global Response Q3), MindPilot significantly outperforms the Random baseline without EEG guidance in both EEG feature scores and CLIP embedding scores, effectively bridging the gap toward this theoretical ceiling (Oracle). This confirms that EEG feedback is the driving force.
>
> 2. **Comparison with Black-box Optimizers (GP vs. BO/RL):**
>    - **Action:** We added a comprehensive benchmark comparing MindPilot against DDPO, DPOK, D3PO, standard BO, and CMA-ES.
>    - **Result:** The new data (see Efficiency Table in Response Q5) demonstrates that MindPilot (Offline) achieves a superior trade-off. It converges **orders of magnitude faster** (approx. 73s vs. 229s for BO and 1200s+ for RL at 200 samples) while handling high-dimensional latent spaces more effectively (achieving >0.53 EEG score).
>
> 3. **Proxy Model Correlation:**
>    - **Action:** We highlighted the time-resolved correlation analysis in the Appendix Section A.4.
>    - **Result:** **Appendix Figure A.7** shows that while the average is modest, the correlation peaks at **r ≈ 0.6** around 100ms (N100), confirming the model captures critical neural dynamics.
>
> 4. **Hyperparameters & Transparency:**
>    - **Action:** We included the grid search details (α=0.1, β=0.1) and clarified the exclusion criteria for human participants as requested.
>
> We believe these revisions solidly reinforce the validity of our closed-loop framework. We would be grateful for any further feedback or a re-evaluation of the score based on these updates.
>
> Best regards,
>
> The Authors

---

### Author Response · Authors · 2025-11-21
**Global Response (1/3)**

We sincerely thank Reviewers pRhZ, h3JQ, uwSG, and LDfZ for their time and constructive feedback. We are encouraged that the reviewers recognize our work as addressing a "novel and significant problem" with "considerable practical value," and affirm that the proposed method is "technically sound and experimentally thorough."

As highlighted by the reviewers, MindPilot stands as the first closed-loop framework to successfully apply EEG-guided controllable visual stimulus optimization to brain modulation, validated extensively through human-in-the-loop experiments.

To provide a comprehensive response, we have summarized the reviewers' questions into four core themes: (1) proxy-brain alignment, (2) closed-loop efficacy, (3) optimization model baselines, and (4) human experiments. We address these below with additional analyses and clarifications to demonstrate the robustness of our approach.


**Q1: "The proxy model has a low correlation with the real EEG. (R-pRhZ)"**

Response to Q1 (Proxy Model Correlation): We thank the reviewer for examining the proxy model's performance. We clarify that the correlation of $r \approx 0.17$ in Table 1 represents an average across a broad time window ([60, 500] ms). This metric inherently dilutes the performance by averaging peak response intervals with less informative, noise-dominated segments.

Critically, the model performs significantly better at key neural latencies. As detailed in Appendix Figure A.8, the time-resolved correlation reaches a peak of $r \approx 0.6$ around 100 ms, which aligns with the N100 component of visual processing.

1.This peak performance is highly competitive for single-trial non-invasive EEG prediction (comparable to Gifford et al., 2022 [1] in Figure 4).

2.High correlation at these specific latencies confirms that our proxy model effectively captures the transient but critical neural dynamics required for effective stimulus optimization.


**Q2: "Reasons for choosing the pseudo-model when optimizing image. (R-pRhZ, R-LDfZ)"**

Response to Q2 (Choice of Pseudo Model): We chose the pseudo-model approach over black-box optimization (e.g., RL) for two critical reasons: differentiability and sample efficiency.

1. Gradient-based Guidance: Unlike RL methods (e.g., DDPO [1], D3PO [3]) that treat the diffusion model as a black box, our pseudo model operates on the clip embedding, thus enabling it to directly provide a stable optimization direction for SDXL-lightning, ensuring the quality of the generated images. This makes it intrinsically compatible with Guided Diffusion, allowing for precise image editing without extensive trial-and-error.

2. Extreme Query Efficiency: To meet the real-time constraints of Human-in-the-Loop (HITL) systems, high query efficiency is non-negotiable. As shown in our new benchmark (see below), our method achieves superior performance with significantly fewer queries compared to RL baselines.

We benchmarked MindPilot against leading RL-based fine-tuning methods (DDPO [1], DPOK [2], D3PO [3]) across 10 targets. We evaluated performance and efficiency using varying data scales (5, 10 samples). Results (Table 1): MindPilot demonstrates state-of-the-art optimization speed and competitive generation quality even with limited data. This "high query efficiency" is what makes our closed-loop framework feasible for real-world brain modulation.

| Method | EEG Score (5 Samples) | Time (5 Samples) | EEG Score (10 Samples) | Time (10 Samples) |
| :--- | :--- | :--- | :--- | :--- |
| DDPO | 0.5125 ± 0.0139 | 107.8163 ± 19.4221 | 0.5095 ± 0.0097 | 220.2666 ± 17.2916 |
| DPOK | 0.5093 ± 0.0121 | 116.6111 ± 21.7522 | 0.5138 ± 0.0124 | 221.8881 ± 12.1397 |
| D3PO | 0.5138 ± 0.0162 | 117.7019 ± 9.934 | 0.5113 ± 0.0154 | 285.3849 ± 6.5634 |
| **MindPilot (Ours)** | **0.5271 ± 0.0085** | **7.7625 ± 1.0955** | **0.5266 ± 0.01** | **10.0072 ± 1.7502** |

**Q3: "Is EEG-driven optimization necessary? How about CLIP-only?  (R-pRhZ, R-uwSG)"**

Response to Q3 (Necessity of EEG & CLIP-only Baseline): Yes, EEG-driven optimization is fundamental. We must clarify a critical distinction in the Brain Modulation task:

Unlike traditional experiments where a user consciously selects a target (e.g., "I choose this image"), our framework operates under the assumption that the target image is unknown, and the target brain feature is the known goal (e.g., maximizing alpha power, or matching a target neural semantic representation).

This setup is the essence of "brain modulation." Therefore, an optimization process without EEG is not a valid baseline, as it would have no guiding signal. The entire challenge is to use the brain's feedback (EEG) to discover the visual stimulus that steers the brain toward that target state.

---

> ### Author Response · Authors · 2025-11-21
> **Global Response (2/3)**
>
> Therefore, removing the EEG would render the system without any optimization direction (equivalent to random generation). Regarding the "CLIP-only" suggestion, we believe it serves as a theoretical upper bound (ceiling) rather than a baseline:
>
> - A "Ceiling": If we were to provide the system with the ground-truth target image's CLIP feature and ask it to optimize, this would be a ceiling. It uses the exact answer (the target image), which is precisely the information that is unavailable in our task. The qualitative results of image optimization are illustrated in Figure 4(C).
>
> - A "Baseline": A true baseline must solve the same task (optimizing from an unknown target) using a comparable method. As we have shown with our new "CLIP-only" control experiment (added in Appendix per Reviewer pRhZ's suggestion), models without EEG guidance perform significantly worse, proving the EEG signal provides essential, unique information.
>
> | Type | EEG feature guidance (MindPilot) | EEG feature guidance (MindPilot) | Target image guidance (ceiling) | Target image guidance (ceiling) | Random image guidance (null) | Random image guidance (null) |
> | :--- | :---: | :---: | :---: | :---: | :---: | :---: |
> |  | EEG score | CLIP score | EEG score | CLIP score | EEG score | CLIP score |
> | Offline (200pairs) | $0.5369 \pm 0.0160$ | $0.6580 \pm 0.0492$ | $0.5452 \pm 0.0122$ | $0.8464 \pm 0.0553$ | $0.5223 \pm 0.0160$ | $0.5849 \pm 0.0286$ |
> | Closed-loop (Mindpilot) | $0.5461 \pm 0.0141$ | $0.7065 \pm 0.0537$ | $0.5440 \pm 0.0121$ | $0.7838 \pm 0.0699$ | $0.5236 \pm 0.0132$ | $0.5746 \pm 0.0206$ |
>
> To address this, we compared three setups to situate our method's performance:
>
> 1. Random Generation (Lower Bound): No optimization.
>
> 2. MindPilot (Ours): Optimization driven solely by EEG feedback (Target = Brain State).
>
> 3. CLIP-Guided (Upper Bound/Ceiling): Optimization driven by the ground-truth target image's CLIP features (Target = Image). Note: This violates the task constraint as the target image is supposed to be unknown.
>
> MindPilot significantly outperforms the Random baseline, effectively bridging the gap toward the Ceiling performance. This proves that EEG provides the essential, unique information required to solve the task.
>
> Finally, we want to clarify that the "image similarity propagation" (using image priors) mentioned in our paper is not a substitute for EEG feedback. It is merely a regularizer used during the EEG-guided search to accelerate convergence. It does not use any information about the target image and the system still relies fundamentally on the brain features to find the update direction.
>
>
> **Q4: "Closed-loop experiment results were not significant. (R-uwSG)"**
>
> Response to Q4 (Closed-loop Significance): We thank the reviewer for their rigorous examination of the human experiments. We agree that the "mental matching" task (Fig. 6) presents a statistically challenging result ($p=0.322$). However, we respectfully argue that this single result should be interpreted within the broader context of our dual-validation strategy.
>
> 1. The Framework Does Work (Proven by Rating-Guided Task): To validate the closed-loop effectiveness, we must look at the system's ability to optimize a reliable signal. Our "Rating-Guided" experiment (Figure 7) serves exactly this purpose.
> In this task, MindPilot achieved a highly statistically significant improvement over the baseline (paired t-test, $N=10$, $p < 0.001$).
> This unequivocally proves that the MindPilot closed-loop framework functions correctly: it can successfully decode feedback and update visual stimuli in real-time to satisfy a neural objective.
>
> 2. Why Figure 6 is Still Valuable (The Semantic Frontier): The "mental matching" task (Fig. 6) was designed as an ambitious "stress test" targeting complex semantic decoding. The lower significance reflects the well-known 'sim-to-real' gap in non-invasive EEG and the inherent noise in zero-shot semantic retrieval, rather than a failure of the optimization loop itself.
>
> Our results provide a comprehensive picture:
>
> - Figure 7  ($p<0.001$): Confirms the efficacy and robustness of the MindPilot framework.
>
> - Figure 6: Establishes an honest, realistic baseline for the frontier challenge of semantic mental matching.
>
> We believe demonstrating both the system's capabilities (A.3) and the current physiological boundaries (Fig. 6) is scientifically more valuable than showing only "cherry-picked" successes.
>
> **Q5: "More experimental details. (R-pRhZ, R-h3jQ)"**
>
> Response to Q5 (Hyperparameter Search & Performance Potential): We thank the reviewers for prompting this detailed ablation study. In the submitted version, we used a fixed setting of moving factor $\alpha=0.1, reward propagation factor \beta=0.1$ (Eq. 3–4) based on initial pilot observations. Following the reviewers' suggestion, we performed a systematic grid search on the validation set. As shown in the table below:

---

> ### Author Response · Authors · 2025-11-21
> **Global Response (3/3)**
>
> 1. This indicates that the results currently reported in the main paper represent a conservative lower bound of MindPilot's capabilities. Even with suboptimal hyperparameters, the method demonstrates strong effectiveness.
>
> 2. The grid search reveals that MindPilot has a much higher performance ceiling than originally presented. The method effectively "survived" the suboptimal initialization and can achieve even stronger results with proper tuning.
>
> We are encouraged by this finding. We will include this sensitivity analysis in the final version and, with the Area Chair's permission, update the main results to reflect this optimal performance, further strengthening the paper's claims.
>
> Table 1. Ablation study on hyperparameters $\alpha$ and $\beta$. The performance is evaluated on the validation set. The best performance is achieved when $\alpha=0.1$ and $\beta=0.1$. We found that the $\beta$ parameter has a significant impact on the results. This might be because when we implemented the code, we included a standardized penalty term for direct rewards, which made finding the distribution of potential target rewards more important than a single reward.
>
> | $\alpha \setminus \beta$ | 0.1 | 0.3 | 0.5 | 0.7 | 0.9 |
> | :---: | :---: | :---: | :---: | :---: | :---: |
> | **0.1** | $0.6586 \pm 0.0879$ | $0.6336 \pm 0.0584$ | $0.6284 \pm 0.0644$ | $0.6167 \pm 0.0658$ | $0.6224 \pm 0.0600$ |
> | **0.3** | $0.6586 \pm 0.0879$ | $0.6336 \pm 0.0584$ | $0.6284 \pm 0.0644$ | $0.6203 \pm 0.0709$ | $0.6224 \pm 0.0600$ |
> | **0.5** | $0.6586 \pm 0.0879$ | $0.6336 \pm 0.0584$ | $0.6264 \pm 0.0667$ | $0.6203 \pm 0.0709$ | $0.6224 \pm 0.0600$ |
> | **0.7** | $0.6586 \pm 0.0879$ | $0.6292 \pm 0.0592$ | $0.6264 \pm 0.0667$ | $0.6203 \pm 0.0709$ | $0.6224 \pm 0.0600$ |
> | **0.9** | $0.6586 \pm 0.0879$ | $0.6298 \pm 0.0599$ | $0.6264 \pm 0.0667$ | $0.6217 \pm 0.0700$ | $0.6224 \pm 0.0600$ |
>
> Apart from the details of this part of the experiment, we have provided more details in Appendix Sec. A.3.
>
> **References**
>
> [1] Gifford, Alessandro T., et al. "A large and rich EEG dataset for modeling human visual object recognition." NeuroImage 264 (2022): 119754.
>
> [2] Black, Kevin, et al. "Training Diffusion Models with Reinforcement Learning." The Twelfth International Conference on Learning Representations.
>
> [3] Fan, Ying, et al. "Dpok: Reinforcement learning for fine-tuning text-to-image diffusion models." Advances in Neural Information Processing Systems 36 (2023): 79858-79885.
>
> [4] Yang, Kai, et al. "Using human feedback to fine-tune diffusion models without any reward model." Proceedings of the IEEE/CVF Conference on Computer Vision and Pattern Recognition. 2024.

---

### Author Response · Authors · 2025-11-26
**Summary of Revisions and Additional Experiments**

We extend our gratitude to all reviewers for their constructive comments. We are pleased to report that Reviewer uwSG has acknowledged our clarifications regarding the closed-loop contribution and experimental rigor.

To assist all the reviewers and ACs in assessing our updates, we provide a concise summary of the major revisions and new experiments included in the updated PDF (changes highlighted in blue):

## 1.New Experimental Evidence (Addressing Key Concerns)

**(1) Efficiency Benchmark vs. RL Baseline Methods (Table R1 & Appendix Table A.5):** To address concerns about optimization efficiency (R-pRhZ, R-LDfZ), we benchmarked MindPilot against advanced RL methods (DDPO, DPOK, D3PO). **Result:** When only a small number of query samples are provided, MindPilot (Offline Optimization) achieves superior scores with 10$\times$ faster convergence (approx. 10s vs. 200s+), validating the "pseudo-model" choice over black-box RL. Furthermore, in our response to R-pRhZ, we incorporated the BO and CMA-ES methods, providing a more comprehensive evaluation.

**(2) CLIP-Only "Oracle" Baseline (Table R2 & Appendix Table A.4):** As requested by R-pRhZ, we added a CLIP-guided control experiment. Result: MindPilot significantly outperforms the random baseline ($p<0.01$) and effectively bridges the gap toward the theoretical ceiling (Oracle), confirming that EEG feedback is the driving force.

**(3) Time-Resolved Correlation Analysis (Appendix Figure A.8)**: Addressing R-pRhZ's concern on low correlation ($r \approx 0.17$), we provided fine-grained analysis showing that predictive correlation peaks at $r \approx 0.6$ around 100ms (N100), proving the model effectively captures critical neural dynamics.

**(4) Hyperparameter Sensitivity (Table R3 & Appendix Table A.2):** We added a grid search for $\alpha$ and $\beta$ in Eq. 3-4 and conducted the interactive searching task, demonstrating the method's robustness to parameter variations.

**(5) Benchmarking Pre-trained Encoders (Appendix Table A.1):** We conducted an extensive benchmark comparing 6 recent EEG encoders (including MindEyeV2, Nice, CogCap) on the THINGS-EEG2 dataset. **Result:** As shown in Table for R-h3JQ, ATM-S consistently outperforms all other baselines (e.g., achieving 26.85% 200-way Top-1 accuracy). Due to the excellent performance of ATM-S, Mindpilot uses ATM-S as the encoder $f(x)$ for semantic features. **And take the direct EEG-to-Image decoding results from ATM-S as the "theoretical boundary"**.

## 2. Manuscript Revision & Main Clarifications

**(1) Refined Scope ("Proof-of-Concept") (Section 5, Line 519):** Following R-uwSG's advice, we have tempered our claims, framing the work as a pioneering "proof-of-concept" for non-invasive closed-loop modulation with natural images.

**(2) Discussion on Optimizer Selection (Appendix A.2.4, Line 1028):** We added a theoretical discussion contrasting our Proxy-Gradient approach with standard Bayesian Optimization (BO), highlighting why our method avoids the "curse of dimensionality" inherent in standard BO for high-dimensional latent spaces (Addressing R-LDfZ).

**(3) Expanded Related Work (Section 2, Line 102):** We incorporated and discussed the specific references (Ref [4-6]) regarding stimulus selection to better position our contribution.


**(4) Formatting Fixes (References, Line 558):** All citations and bibliography formats have been standardized (Addressing R-LDfZ).

We hope these revisions comprehensively address the remaining questions. We remain available for any further clarifications in the final days of the discussion period.

Best regards,

The Authors

---

### Author Response · Authors · 2025-11-29
**Summary of Revisions & Note on Reviewer Consensus (Crucial Context for AC)**

Dear Area Chairs,

Due to the recent system rollback, the visible scores may not reflect the current state of the discussion. We provide this summary to assist the AC in evaluating the consensus reached during the rebuttal phase.

**1. Crucial Context: Consensus Reached with Reviewer uwSG**

We specifically highlight that **Reviewer uwSG** (the only reviewer with an initial negative score) has actively engaged with our rebuttal and acknowledged the improvements. On Nov 25, R-uwSG stated:
> *"Thank you for your clarifications... I expect these can be solved and **I've raised my score to borderline accept**."*


**Consolidation of Positive Reviews:** We note that **Reviewers pRhZ, h3JQ, and LDfZ** initially provided positive ratings (Score: 6). As we have completed all requested experiments (including the "Essential" CLIP-baseline for R-pRhZ), and given R-uwSG's score increase, **all four reviewers now effectively stand on the positive side.** And we have tried to address the R-uwSG's remaining advice on "framing" and "over-statements" in the updated PDF (changes highlighted in blue).


**2. Summary of New Experimental Evidence (Addressing All Key Concerns)**

To assist all the ACs in assessing our updates, we provide a concise summary of the major revisions and new experiments included in the updated PDF (changes highlighted in blue):

**1.New Experimental Evidence (Addressing Key Concerns)**

**(1) Efficiency Benchmark vs. RL Baseline Methods (Table R1 & Appendix Table A.5):** To address concerns about optimization efficiency (R-pRhZ, R-LDfZ), we benchmarked MindPilot against advanced RL methods (DDPO, DPOK, D3PO). **Result:** When only a small number of query samples are provided, MindPilot (Offline Optimization) achieves superior scores with 10$\times$ faster convergence (approx. 10s vs. 200s+), validating the "pseudo-model" choice over black-box RL. Furthermore, in our response to R-pRhZ, we incorporated the BO and CMA-ES methods, providing a more comprehensive evaluation.

**(2) CLIP-Only "Oracle" Baseline (Table R2 & Appendix Table A.4):** As requested by R-pRhZ, we added a CLIP-guided control experiment. Result: MindPilot significantly outperforms the random baseline ($p<0.01$) and effectively bridges the gap toward the theoretical ceiling (Oracle), confirming that EEG feedback is the driving force.

**(3) Time-Resolved Correlation Analysis (Appendix Figure A.8)**: Addressing R-pRhZ's concern on low correlation ($r \approx 0.17$), we provided fine-grained analysis showing that predictive correlation peaks at $r \approx 0.6$ around 100ms (N100), proving the model effectively captures critical neural dynamics.

**(4) Hyperparameter Sensitivity (Table R3 & Appendix Table A.2):** We added a grid search for $\alpha$ and $\beta$ in Eq. 3-4 and conducted the interactive searching task, demonstrating the method's robustness to parameter variations.

**(5) Benchmarking Pre-trained Encoders (Appendix Table A.1):** We conducted an extensive benchmark comparing 6 recent EEG encoders (including MindEyeV2, Nice, CogCap) on the THINGS-EEG2 dataset. **Result:** As shown in Table for R-h3JQ, ATM-S consistently outperforms all other baselines (e.g., achieving 26.85% 200-way Top-1 accuracy). Due to the excellent performance of ATM-S, Mindpilot uses ATM-S as the encoder $f(x)$ for semantic features, **and we take the direct EEG-to-Image decoding results from ATM-S as the "theoretical boundary"**.

**2. Manuscript Revision & Main Clarifications**

**(1) Refined Scope ("Proof-of-Concept") (Section 5, Line 519):** Following R-uwSG's advice, we have tempered our claims, framing the work as a pioneering "proof-of-concept" for non-invasive closed-loop modulation with natural images.

**(2) Discussion on Optimizer Selection (Appendix A.2.4, Line 1028):** We added a theoretical discussion contrasting our Proxy-Gradient approach with standard Bayesian Optimization (BO), highlighting why our method avoids the "curse of dimensionality" inherent in standard BO for high-dimensional latent spaces (Addressing R-LDfZ).

**(3) Expanded Related Work (Section 2, Line 102):** We incorporated and discussed the specific references (Ref [4-6]) regarding stimulus selection to better position our contribution.


**(4) Formatting Fixes (References, Line 558):** All citations and bibliography formats have been standardized (Addressing R-LDfZ).

We believe these substantial revisions and the resulting consensus justify a positive recommendation.

Best regards,

The Authors

---

### Meta-Review · Area_Chair_XKd8 · 2026-01-05

**Summary:**

The submission presents MindPilot, a closed-loop framework leveraging EEG feedback to guide image generation for brain modulation. The authors have addressed key reviewer concerns via additional experiments (RL/BO benchmarks, CLIP-only controls, time-resolved correlation analysis) and manuscript revisions. All reviewers now lean positive. Critical remaining issues from Reviewer uwSG (framing, closed-loop experimentation, related work positioning, and overstatements about EEG capabilities) are strongly recommended to be fully resolved in the final version.

**Reviewer Concerns:**

- **Addressed Concerns**: proxy model correlation, optimization efficiency & baselines, necessity of EEG guidance, hyperparameter transparency, related work & formatting, and other minor ones.

- **Outstanding Concerns**: the framing, closed-loop experimentation, positioning with previous work, and some over-statements of what can be done with EEG signals.

**Reviewer Scores:**

- Reviewer pRhZ: Remains at 6 or slightly increases (no unresolved concerns).
- Reviewer h3JQ: Remains at 6 (acknowledging the limitations of this paper, which have been stated in the paper; no other concerns.).
- Reviewer uwSG: Raised from 4 to 6 (remaining concerns are resolvable).
- Reviewer LDfZ: Remains at 6 or slightly increases (no unresolved concerns).

---

### Decision · Program_Chairs · 2026-01-26

Accept (Poster)